# ProOPF: Benchmarking and Improving LLMs for Professional-Grade Power Systems Optimization Modeling

**Chao Shen** [*1] **Zihan Guo** [*2] **Xu Wan** [*1] **Zhenghao Yang** [2] **Yifan Zhang** [3] **Wenqi Huang** [4] **Jie Song** [2] **Zongyan Zhang** [2] **Mingyang Sun** [2]

## Abstract

Growing renewable penetration introduces substantial uncertainty into power system operations, necessitating frequent adaptation of dispatch objectives and constraints and challenging expertise-intensive, near-real-time modeling workflows. Large Language Models (LLMs) provide a promising avenue for automating this process by translating natural-language (NL) operational requirements into executable optimization models via semantic reasoning and code synthesis. Yet existing LLM datasets and benchmarks for optimization modeling primarily target coarse-grained cross-domain generalization, offering limited, rigorous evaluation in power-system settings, particularly for Optimal Power Flow (OPF). We therefore introduce **ProOPF-D** and **ProOPF-B**, a dataset and benchmark for professional-grade OPF modeling: ProOPF-D contains 12K instances pairing NL requests with parameter adjustments and structural extensions to a canonical OPF, together with executable implementations; ProOPF-B provides 121 expert-annotated test cases with ground-truth code, enabling end-to-end evaluation under both concrete and abstract OPF modeling regimes. Our code, dataset, and benchmark are publicly available at this GitHub repository.

## 1. Introduction

Power system optimization modeling has become increas-

*Equal contribution [1]College of Control Science and Engineering, Zhejiang University [2]School of Advanced Manufacturing and Robotics, Peking University [3]School of Electrical Engineering, Xi'an Jiaotong University [4]China Southern Power Grid Novel Electric Power System (BEIJING) Research Institute Co., Ltd.. Correspondence to: Mingyang Sun <smy@pku.edu.cn>, Zongyan Zhang <zongyanzhang@pku.edu.cn>.

*Proceedings of the 43rd International Conference on Machine Learning*, Seoul, South Korea. PMLR 306, 2026. Copyright 2026 by the author(s).

ingly complex as growing renewable penetration and expanding system scale amplify the diversity of operating conditions and operational requirements (Shen et al., 2025a; Biagioni et al., 2020; Pan et al., 2020; Wan et al., 2023). In this context, optimal power flow (OPF) and its variants serve as a foundational modeling framework that underpins a wide range of operational decision-making tasks in power systems (Abdi et al., 2017; Zuo et al., 2025). Since its introduction by Carpentier in 1962 (Babiker et al., 2025), OPF has been widely deployed to optimize generation and network operation under physical power flow laws and operational constraints, delivering substantial economic benefits (Schecter & O'Neill, 2013). In real-world operations, diverse operating conditions are typically reflected through parameter variations in OPF instances, such as changes in loads, or network limits (Pan et al., 2020; Rivera et al., 2026). Beyond parametric changes, the rising frequency of extreme scenarios (e.g., severe weather events or sudden outages) demands adaptive real-time dispatch models capable of dynamic structural modifications, such as injecting ad-hoc security constraints or probabilistic formulations (Wang et al., 2023; Zuo et al., 2025). While power system experts can manually revise these OPF models to ensure physical consistency, the process is prohibitively expertise-intensive (Jabr, 2013; Gao et al., 2022). Crucially, the stringent time limits of real-time dispatch leave operators with no room to manually rewrite and debug complex implementation code when unexpected anomalies occur (Jia et al., 2025; Jabr, 2013). This temporal bottleneck underscores the critical need for an automated and agile modeling paradigm.

Recent advances in large language models (LLMs) offer a promising paradigm to bridge this gap. Specifically, LLMs can automate the translation of expert intent into executable OPF formulations: operators provide high-level operational requirements and scenario descriptions via natural language instructions, while the LLM handles the tedious semantic-to-code mapping and structural formulation adjustments (Jia et al., 2025). To systematically evaluate LLMs' proficiency in optimization modeling, early benchmark efforts have largely focused on the broader operations research domain (Xiao et al., 2025), yielding general-purpose datasets such as NL4Opt (Ramamonjison et al., 2023), MAMO (Huang

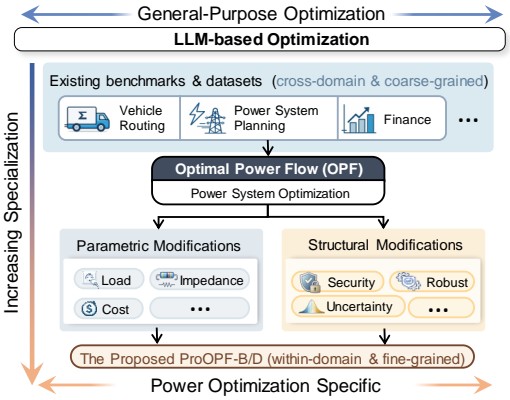

(a) From cross-domain to within-domain generalization

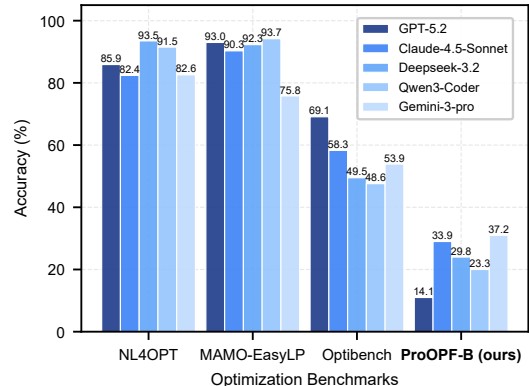

(b) Comparison of existing benchmarks and ProOPF-B

*Figure 1.* From cross-domain to within-domain generalization in LLM-based optimization. Existing works emphasize coarse-grained generalization across heterogeneous optimization tasks, whereas ProOPF-B/D targets fine-grained, within-domain generalization in OPF through parametric and structural formulation modifications.

et al., 2024), and Optibench (Yang et al., 2025b). However, these existing benchmarks predominantly evaluate coarse-grained, cross-domain generalization (e.g., from vehicle routing (Xiao et al., 2023) to job-shop scheduling (Yang et al., 2025b)) for non-expert users, as shown in Figure 1. They fall short of assessing the fine-grained, expert-level competence—specifically, the capacity for rigorous parameter reasoning and structural reformulation—required for professional-grade power system operation.

Complementing these benchmarks, a range of datasets and data synthesis pipelines has been developed to support LLM training for optimization modeling (Huang et al., 2025; Jiang et al., 2025; Yang et al., 2025b). Existing synthesis pipelines adopt two complementary approaches: either rewriting NL problem descriptions and inferring corresponding optimization models (Huang et al., 2025), or operating directly on structured optimization models and reverse-generating aligned NL descriptions (Huang et al., 2025; Yang et al., 2025b). However, these pipelines exhibit three limitations: 1) they focus only on mathematical solvability without explicitly enforcing physical validity; 2) insufficient fine-grained coverage of realistic OPF operating scenarios; and 3) they mainly supervise LLMs to generate complete optimization formulations, which is unnecessary for OPF modeling and prone to introducing physical and modeling inconsistencies.

To bridge these gaps, we introduce **ProOPF-D** (Dataset) and **ProOPF-B** (Benchmark)[1] to systematically improve and evaluate LLM competence in professional-grade power system optimization modeling. 1) **ProOPF-D** comprises 12K samples organized into four difficulty levels, defined by the form of parameter specification (explicit versus infer-

---

[1]ProOPF-D and ProOPF-B are available: ProOPF-B/D repository

ence) and the required degree of structural extension beyond a base OPF formulation. ProOPF-D designs a modification-based representation in which each instance specifies the differential components (parameter changes and structural extensions) rather than the full OPF formulations. In addition, expert-curated constraints restrict the feasible spaces of parameter modifications and structural extensions to enforce physical consistency with power system laws. 2) **ProOPF-B** comprises 121 benchmark instances spanning all four difficulty levels of ProOPF-D. Each instance is expert-annotated with representative OPF modeling tasks from the literature, facilitating systematic evaluation across OPF modeling complexities. As shown in Figure 1b, while state-of-the-art LLMs can achieve over 90% accuracy on existing benchmarks, their performance drops to below 30% on ProOPF-B, highlighting the need for fine-grained, professional-grade evaluation frameworks to accurately assess domain-specific modeling competence.

The main contributions of this work are:

1. We introduce ProOPF-D/B, the first dataset & benchmark designed to systematically evaluate LLM competence in specialized power system optimization modeling, shifting LLM-based optimization to large-scale, complex power domain specific modeling.

2. For ProOPF-D, we introduce a novel multi-level dataset construction pipeline that systematically progresses from explicit to inferred parameters and from parametric changes to structural extensions. Each instance is represented as a modification to a canonical OPF formulation and constrained within physically feasible spaces, with aligned NL descriptions, OPF modeling specifications, and executable implementations.

3. For ProOPF-B, we construct a benchmark from expert-selected, peer-reviewed literature and aligned its difficulty

tiers with those of ProOPF-D. In particular, a level-specific evaluation protocol is proposed that distinguishes concrete and abstract OPF-based modeling, enabling rigorous end-to-end assessment of executable correctness.

## 2. Related Work

**Benchmarks for Optimization Modeling.** A range of benchmarks have been developed to evaluate optimization modeling from NL using LLMs. Early work such as NL4Opt (Ramamonjison et al., 2023) focuses on translating NL descriptions into mathematical formulations, without considering solver execution. Subsequent benchmarks, including ComplexOR (Xiao et al., 2023), NLP4LP (AhmadiTeshnizi et al., 2023), and Mamo (Huang et al., 2024), provide paired NL descriptions and optimization solutions, but remain largely confined to linear or mixed-integer linear programs. More recent benchmarks, such as IndustryOR (Huang et al., 2025) and OptiBench (Yang et al., 2025b), extend coverage to nonlinear optimization problems derived from industrial scenarios.

However, prior benchmarks largely emphasize breadth across heterogeneous optimization domains, which limits fine-grained evaluation on complex, power-domain specific tasks such as OPF with tightly coupled physical constraints. A substantial scale gap further complicates assessment: even small OPF instances (e.g., IEEE 9-bus) typically have >20 decision variables and >60 constraints, while IndustryOR averages fewer than 15 variables and constraints per instance (Xiao et al., 2025).

**Datasets Synthesis for LLM-based Optimization**

High-quality training data for optimization modeling remains scarce, motivating data synthesis techniques to construct scalable datasets (Xiao et al., 2025). Existing synthesis pipelines broadly fall into problem-centric and model-centric approaches. Problem-centric methods expand datasets by rewriting NL descriptions from seed problems and inferring corresponding optimization models (e.g., OR-Instruct (Huang et al., 2025), LLMOPT (Jiang et al., 2025)), but they struggle to systematically scale instance complexity, as mapping increasingly elaborate NL to valid and solvable models can be fragile. By contrast, model-centric methods operate directly on structured optimization models and reverse-generate aligned NL descriptions (e.g., ReSocratic (Yang et al., 2025b), MILP-Evolve (Li et al., 2024)), offering controllable instance generation with solvability and semantic consistency.

Across paradigms, samples are often generated by perturbing seed problems without explicitly enforcing domain physics (e.g., power flow constraints), so physical feasibility may be violated even when the formulation is mathematically well-posed. Moreover, most pipelines supervise LLMs to regenerate complete models; for OPF, instances largely share a common backbone and differ mainly via scenario-dependent parameter updates and structured extensions, making end-to-end regeneration redundant and error-prone.

## 3. Power System Optimization Modeling

### 3.1. Foundation: Optimal Power Flow Problem

The OPF problem seeks optimal generator dispatch and network operating conditions by minimizing generation cost under power flow equations and network operating constraints. Given a power system represented as a graph with bus set $\mathcal{N}$, branch set $\mathcal{E}$, and generator set $\mathcal{G} \subseteq \mathcal{N}$, the OPF problem can be formulated as:

$$\min_{\boldsymbol{x}} \quad \boldsymbol{f}(\boldsymbol{x}) \tag{1}$$

$$\text{s.t.} \quad \boldsymbol{g}(\boldsymbol{x}) = 0, \ \boldsymbol{h}(\boldsymbol{x}) \leq 0, \tag{2}$$

$$\boldsymbol{x} \in [\, \underline{\boldsymbol{x}}, \ \overline{\boldsymbol{x}} \,]. \tag{3}$$

where $\boldsymbol{x} = [\boldsymbol{V}, \boldsymbol{\theta}, \boldsymbol{P}_G, \boldsymbol{Q}_G]$ denotes the decision variables, including the voltage magnitudes $\boldsymbol{V} = \{V_i\}_{i \in \mathcal{N}}$ and phase angles $\boldsymbol{\theta} = \{\theta_i\}_{i \in \mathcal{N}}$ at all buses, as well as the active and reactive power outputs of generators, $\boldsymbol{P}_G = \{P_{G,g}\}_{g \in \mathcal{G}}$ and $\boldsymbol{Q}_G = \{Q_{G,g}\}_{g \in \mathcal{G}}$. The objective function $f(\boldsymbol{x})$ represents the total generation cost and is defined as

$$f(\boldsymbol{x}) = \sum_{g \in \mathcal{G}} \left( a_g P_{G,g}^2 + b_g P_{G,g} + c_g \right), \tag{4}$$

where $a_g$, $b_g$, and $c_g$ denote the cost coefficients of $g$-th generator. The equality constraints $\boldsymbol{g}(\boldsymbol{x})$ are given by the real and reactive power balance equations at each bus $i \in \mathcal{N}$:

$$
\begin{aligned}
P_{G,i} - P_{D,i} &= V_i \sum_{j \in \mathcal{N}} V_j \left( G_{ij} \cos \theta_{ij} + B_{ij} \sin \theta_{ij} \right), \\
Q_{G,i} - Q_{D,i} &= V_i \sum_{j \in \mathcal{N}} V_j \left( G_{ij} \sin \theta_{ij} - B_{ij} \cos \theta_{ij} \right),
\end{aligned}
\tag{5}
$$

Here, $P_{G,i}$ and $Q_{G,i}$ denote the active and reactive power injections at bus $i$ (set to zero if no generator is connected), $P_{D,i}$ and $Q_{D,i}$ are the corresponding power demands, $G_{ij}$ and $B_{ij}$ denote the conductance and susceptance between buses $i$ and $j$, and $\theta_{ij} = \theta_i - \theta_j$ is the voltage phase angle difference. The inequality constraints $\boldsymbol{h}(\boldsymbol{x})$ enforce branch thermal limits by bounding the apparent power flows at both ends of each transmission line. For each branch $(i, j) \in \mathcal{E}$, the apparent power magnitudes at the from and to ends are constrained by

$$|S_{f,ij}| = |V_i I_{f,ij}^*| \leq \overline{S}_{ij}, \quad |S_{t,ij}| = |V_j I_{t,ij}^*| \leq \overline{S}_{ij}, \tag{6}$$

where $|S_{f,ij}|$ and $|S_{t,ij}|$ are the apparent power at the from and to ends, $V_i$ and $V_j$ are the complex bus voltages, $I_{f,ij}$ and $I_{t,ij}$ are the complex branch currents, $(\cdot)^*$ denotes the

complex conjugate, and $\overline{S}_{ij}$ is the thermal apparent power limit of branch $(i, j)$. The variable limits in (3) impose bounds on bus voltage magnitudes and phase angles, as well as on generator active and reactive power outputs:

$$
\begin{aligned}
&V_i \in [\underline{V}_i, \overline{V}_i],\ \theta_i \in [\underline{\theta}_i, \overline{\theta}_i],\ \forall i \in \mathcal{N}; \\
&P_{G,g} \in [\underline{P}_{G,g}, \overline{P}_{G,g}],\ Q_{G,g} \in [\underline{Q}_{G,g}, \overline{Q}_{G,g}],\ \forall g \in \mathcal{G}.
\end{aligned} \tag{7}
$$

where $\underline{(\cdot)}$ and $\overline{(\cdot)}$ denote the lower and upper bounds.

## 3.2. Base Model Modification

The OPF formulation in (4)–(7) produces a large-scale, tightly coupled nonlinear program, with $\mathcal{O}(|\mathcal{N}| + |\mathcal{G}|)$ decision variables and $\mathcal{O}(|\mathcal{N}| + |\mathcal{E}| + |\mathcal{G}|)$ constraints. At this scale, annotating complete models is costly, and generating complete formulations while preserving global structural validity (indexing, dimensions, and network couplings) is brittle. We therefore leverage the invariant physical backbone shared across OPF instances and supervise only what varies—parameter updates and structured extensions—by representing each instance as a modification of a canonical base model. To formalize this, we introduce a base OPF problem $\mathcal{Q}_0$ corresponding to (4)–(7), parameterized by base system parameters $\boldsymbol{\pi}_{\text{sys}}$ with the following structure:

$$
\begin{aligned}
\boldsymbol{\pi}_{\text{sys}} \triangleq \{&(a_g, b_g, c_g)_{g \in \mathcal{G}},\ (G_{ij}, B_{ij}, \overline{S}_{ij})_{(i,j) \in \mathcal{E}}, \\
&(\underline{\boldsymbol{x}}, \overline{\boldsymbol{x}}),\ (P_{D,i}, Q_{D,i})_{i \in \mathcal{N}}\}.
\end{aligned} \tag{8}
$$

We parameterize each target OPF model $\mathcal{Q}$ as a modification of the canonical base model $\mathcal{Q}_0$, specified by parameter modifications $\Delta\boldsymbol{\pi}$ and a structural modification operator $s$:

$$
\begin{aligned}
\mathcal{Q}(\Delta\boldsymbol{\pi}, s \mid \boldsymbol{\pi}_{\text{sys}}) &\triangleq \text{Modify}_s(\mathcal{Q}_0(\boldsymbol{\pi}_{\text{sys}} + \Delta\boldsymbol{\pi})), \\
\Delta\boldsymbol{\pi} &\in \Omega_{\boldsymbol{\pi}},\ s \in \Omega_s \cup \{\varnothing\},
\end{aligned} \tag{9}
$$

where $\Omega_{\boldsymbol{\pi}}$ and $\Omega_s$ are expert-curated sets of admissible parameter modifications and structural modification operators, respectively.

# 4. ProOPF-D: Expert-Curated Dataset via Modification-Based OPF Representation

## 4.1. Structured Sample Representation

Each sample $z \in$ ProOPF-D is represented as a triple

$$
z = \{\mathcal{P},\ \mathcal{M},\ \mathcal{I}\}, \tag{10}
$$

where $\mathcal{P}$ denotes an NL description of OPF modeling and solving requirements, $\mathcal{M}$ denotes a target OPF model specification, and $\mathcal{I}$ denotes an executable implementation of $\mathcal{M}$. Following (9), the target OPF model $\mathcal{Q}$ is fully specified by parameter modifications $\Delta\boldsymbol{\pi}$, structural modification operator $s$, and base system parameters $\boldsymbol{\pi}_{\text{sys}}$. Accordingly, $\mathcal{M}$ is represented as

$$
\mathcal{M} = \{\omega,\ \Delta\boldsymbol{\pi},\ s,\ \mathcal{R}\}, \tag{11}
$$

where $\omega \in \Omega_{\text{sys}}$ identifies a reference power system and retrieves the corresponding base parameters $\boldsymbol{\pi}_{\text{sys}}(\omega)$. The complete list of systems in $\Omega_{\text{sys}}$ is provided in subsection A.1. Parametric modifications $\Delta\boldsymbol{\pi}$ are organized as a finite set of parameter patches $\{\delta_k\}_{k=1}^{K}$ where each patch $\delta_k$ is a structured tuple encoding the component type, target parameter identifier, modification operation, value, and other fields. Structural modifications $s$ are organized as a triple $s = \{s_p, s_c, s_o\}$ where $s_p$ denotes the problem type (e.g., DCOPF), $s_c$ specifies constraint extensions (e.g., security constraints), and $s_o$ encodes objective function modifications (e.g., renewable curtailment penalties). Each component may contain multiple fields specifying the modification type and formulation details. The resolution specification $\mathcal{R}$ encodes solver configuration parameters including solver selection (e.g., MIPS), termination criteria (e.g., violation tolerance), and output options. The space $\Omega_{\mathcal{R}}$ denotes the set of all admissible solver requirement. Detailed specifications $\mathcal{M}$ are provided in Appendix A.

## 4.2. Multi-Level Difficulty Taxonomy and Synthesis

ProOPF-D is categorized into four levels based on the degree of expert knowledge required to translate an NL request into an executable OPF formulation. In particular, we consider two orthogonal dimensions: whether parameter modifications $\Delta\boldsymbol{\pi}$ are explicitly specified in $\mathcal{P}$ or should be inferred from scenario-dependent operational description, and whether a structural modification $s \in \mathcal{S}$ is required beyond the base OPF model. The Cartesian product of these two binary dimensions yields four difficulty levels that systematically cover the modeling action space $\Omega_{\boldsymbol{\pi}} \times (\mathcal{S} \cup \{\varnothing\})$, spanning a progressive spectrum from basic modification to expert-level model adaptation. Data synthesis follows a model-centric paradigm: $\mathcal{M}$ is first instantiated from the level-specific sampling space $\Omega_{\mathcal{M}}^{\mathcal{L}_i}$ for level $i$, then $\mathcal{P}$ and $\mathcal{I}$ are derived through deterministic or LLM-guided generation conditioned on $\mathcal{M}$. Complete examples and prompts used for LLM-based generation of $\mathcal{P}$ and $\mathcal{I}$ for all four levels are provided in Appendix D. The synthesis procedures for each level are shown in Figure 2 and formalized as pseudocode algorithms (Algorithm 1-4) in Appendix C. We elaborate on each level below.

### Level 1 ($\mathcal{L}_1$): Explicit Parameter Specification

**Task Definition.** $\mathcal{L}_1$ covers cases in which parameter modifications $\Delta\boldsymbol{\pi}$ are explicitly specified in $\mathcal{P}$ without any structural modifications ($s = \varnothing$). For example, the input description "*Increase active loads at Bus 7 by 10%*" explicitly specifies $\Delta\boldsymbol{\pi}$ and enables direct model instantiation. Such explicit parameter directives commonly arise in control-room operations and what-if studies, where measurements, forecasts, or operating limits are already quantified and can be issued as numeric updates. $\mathcal{L}_1$ evaluates the model's abil-

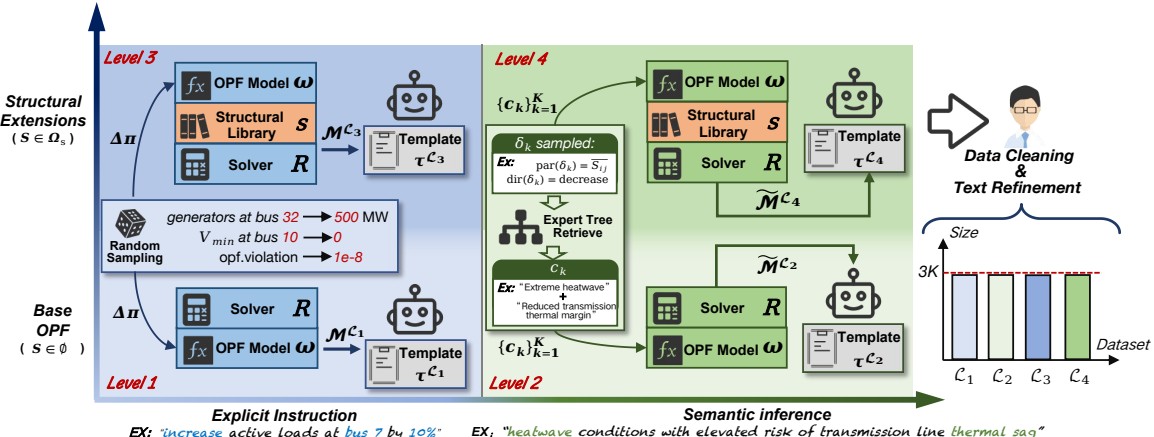

*Figure 2.* The ProOPF-D dataset construction pipeline. $\mathcal{L}_1$ generates samples by directly instantiating OPF models from explicitly specified parameter patches. $\mathcal{L}_2$ synthesizes scenario-driven samples by mapping qualitative operational descriptions to parameter modification directions using expert-curated scenario trees. $\mathcal{L}_3$ extends the base OPF formulation with expert-designed structural variants combined with explicit parameter updates. $\mathcal{L}_4$ integrates semantic parameter inference and structural extensions, representing the most challenging expert-level modeling setting. Finally, the synthetic data undergoes Data Cleaning and Text Refinement, resulting in a total dataset of 12k samples evenly distributed across four levels (3k each).

ity to parse explicit numerical instructions and accurately map them to corresponding OPF parameter configurations.

**Data Synthesis.** For each sample $z \in \mathcal{L}_1$, we construct the data instance via a two-stage generative process. First, an OPF model specification is uniformly sampled as $\mathcal{M}^{\mathcal{L}_1} \sim \mathcal{U}\left(\Omega_{\mathcal{M}}^{\mathcal{L}_1}\right)$, where $\Omega_{\mathcal{M}}^{\mathcal{L}_1} := \Omega_{\text{sys}} \times \Omega_{\boldsymbol{\pi}} \times \{\varnothing\} \times \Omega_{\mathcal{R}}$ denotes the sampling space over system templates, explicit parameter patches, empty structural modifications, and solver requirements.

Conditioned on the sampled model specification $\mathcal{M}^{\mathcal{L}_1}$ and a pre-defined instruction specification $\boldsymbol{\tau}$, an NL request $\mathcal{P}$ and an executable implementation $\mathcal{I}$ are generated by the LLM according to

$$(\mathcal{P}, \mathcal{I}) \sim p_{\text{LLM}}\left(\cdot \mid \mathcal{M}^{\mathcal{L}_1}, \boldsymbol{\tau}^{\mathcal{L}_1}\right), \quad (12)$$

where $\boldsymbol{\tau}^{\mathcal{L}_1}$ defines structured prompting rules that guide the LLM in translating the sampled parameter patches and solver requirements into coherent NL descriptions and executable MATPOWER code. The complete sample $z$ is then formed by combining $\mathcal{M}^{\mathcal{L}_1}$, $\mathcal{P}$, and $\mathcal{I}$.

**Level 2 ($\mathcal{L}_2$): Semantic Parameter Inference**

**Task Definition.** In $\mathcal{L}_2$, parameter modifications $\Delta\boldsymbol{\pi}$ are implicitly conveyed through operational scenario descriptions, requiring models to infer affected parameter types and qualitative variation trends without concrete numerical values. For instance, the description "*heatwave conditions with elevated risk of transmission line thermal sag*" semantically implies thermal constraint tightening, which translates to decreased apparent power limits $\overline{S}_{ij}$. More generally, $\mathcal{L}_2$ maps scenario-level semantic cues to qualitative parameter adjustments (e.g., increase, decrease, no change), typical of event-driven operating conditions (e.g., heatwaves, storms, contingencies) where only qualitative implications for limits and demands are available.

**Data Synthesis.** For each sample $z \in \mathcal{L}_2$, a model specification $\mathcal{M}^{\mathcal{L}_2}$ is sampled uniformly from

$$\Omega_{\mathcal{M}}^{\mathcal{L}_2} := \Omega_{\text{sys}} \times \Omega_{\boldsymbol{\pi}}^{\text{dir}} \times \{\varnothing\} \times \Omega_{\mathcal{R}}. \quad (13)$$

Compared to $\mathcal{L}_1$, the space $\Omega_{\boldsymbol{\pi}}^{\text{dir}}$ consists of parameter patches $\delta_k$ without concrete values, each characterized by a parameter identifier $\text{par}(\delta_k)$ and a qualitative modification indicator $\text{dir}(\delta_k) \in \{\text{INCREASE}, \text{DECREASE}, \text{SETZERO}\}$. However, directly prompting an LLM with $\mathcal{M}^{\mathcal{L}_2}$ to generate $\mathcal{P}$ is problematic for two reasons. First, exposing parameter identifiers and modification directions in the prompt may cause the generated $\mathcal{P}$ to explicitly enumerate these parameters and their trends (e.g., "decrease thermal limits $\overline{S}_{ij}$"), which leaks the inference targets that $\mathcal{L}_2$ is designed to evaluate. Second, unconstrained generation without domain-specific guidance may yield semantically misaligned scenario descriptions due to limited coverage of power system operational semantics.

To address this, we introduce a collection of expert-curated scenario trees $\mathcal{T} = \{T_m\}_{m=1}^{M}$ (see Figure 3), hierarchically organizing operational semantics from event-level nodes (E-level, e.g., extreme heatwave) to mechanism-level nodes (M-level, e.g., reduced transmission thermal margin), with leaf nodes annotated by parameter-direction pairs. Given $\Delta\boldsymbol{\pi} = \{\delta_k\}_{k=1}^{K} \in \Omega_{\boldsymbol{\pi}}^{\text{dir}}$, we retrieve the corresponding leaf nodes:

$$\mathcal{V}_k = \text{Retrieve}(\mathcal{T} \mid \text{par}(\delta_k), \text{dir}(\delta_k)), \quad (14)$$

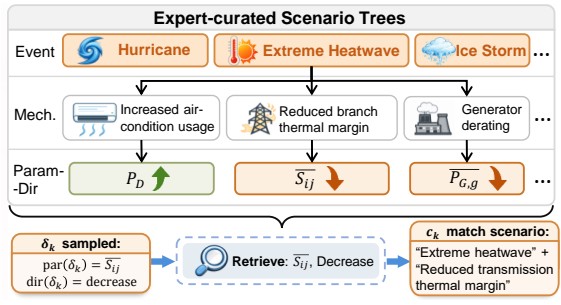

*Figure 3.* Expert-curated scenario trees for $\mathcal{L}_2$ synthesis. Top: Hierarchical structure from event-level (E-level) through mechanism-level (M-level) nodes to leaf nodes encoding parameter types and modification trends. Bottom: Retrieval process where $\mathrm{Retrieve}(\mathcal{T} \mid \mathrm{par}(\delta_k), \mathrm{dir}(\delta_k))$ matches parameter patch $\delta_k$ to leaf nodes, and the root-to-leaf path forms scenario fragment $c_k$.

where $\mathcal{V}_k$ consists of all leaf nodes in $\mathcal{T}$ whose associated parameter and direction annotations match $\mathrm{par}(\delta_k)$ and $\mathrm{dir}(\delta_k)$, respectively. Let $\mathcal{N}_k$ denote the set of root-to-leaf paths induced by $\mathcal{V}_k$. A scenario fragment is sampled uniformly as $c_k \sim \mathcal{U}(\mathcal{N}_k)$, where each $c_k$ is a concatenation of E-level and M-level nodes along a root-to-leaf path. The resulting fragment set $\mathbf{c} = \{c_k\}_{k=1}^K$ replaces $\Delta\boldsymbol{\pi}$ to form the intermediate specification $\widetilde{\mathcal{M}}^{\mathcal{L}_2}$:

$$\widetilde{\mathcal{M}}^{\mathcal{L}_2} = \{\omega, \{c_k\}_{k=1}^K, \boldsymbol{s}, \mathcal{R}\} \tag{15}$$

This substitution prevents the leakage of inference targets by removing direct $\Delta\boldsymbol{\pi}$ exposure. Meanwhile, the scenario fragments $\{c_k\}_{k=1}^K$ facilitate diverse and semantically grounded scenario descriptions aligned with power system operational knowledge. Given $\widetilde{\mathcal{M}}^{\mathcal{L}_2}$ and instructions $\boldsymbol{\tau}^{\mathcal{L}_2}$, the LLM generates the NL description $\mathcal{P}$ and executable implementation $\mathcal{I}$ jointly:

$$(\mathcal{P}, \mathcal{I}) \sim p_{\mathrm{LLM}}\left(\cdot \mid \widetilde{\mathcal{M}}^{\mathcal{L}_2}, \boldsymbol{\tau}^{\mathcal{L}_2}\right), \tag{16}$$

where $\boldsymbol{\tau}^{\mathcal{L}_2}$ directs the LLM to aggregate scenario fragments $\{c_k\}_{k=1}^K$ into a coherent operational narrative in $\mathcal{P}$ and to express $\mathcal{I}$ via placeholder-based parameter assignments that preserve the specified modification directions.

**Level 3 ($\mathcal{L}_3$): Structural Extensions with Explicit Parameters**

**Task Definition.** $\mathcal{L}_3$ focuses on the challenging scenario where a model simultaneously accommodates structural modifications $s \in \Omega_s$ that extend the base OPF formulation and explicit parameter modifications $\Delta\boldsymbol{\pi}$ specified in $\mathcal{P}$. Structural extensions introduce additional controllable units (e.g., unit commitment variables), operational constraints (e.g., N-1 security constraints), or objective functions (e.g., environmental emission costs). For instance, the input "*enforce N-1 security constraints with active loads at bus 7 increased by 10%*" requires introducing a structural extension

(security constraints) while explicitly specifying parameter modifications (load adjustments).

**Data Synthesis.** For each sample $z \in \mathcal{L}_3$, a model specification $\mathcal{M}^{\mathcal{L}_3}$ is sampled uniformly from

$$\Omega_{\mathcal{M}}^{\mathcal{L}_3} := \Omega_{\mathrm{sys}} \times \Omega_{\boldsymbol{\pi}} \times \Omega_s \times \Omega_{\mathcal{R}}, \tag{17}$$

Unlike $\mathcal{L}_1$ and $\mathcal{L}_2$, $\mathcal{L}_3$ employs a non-empty structural modification space $\Omega_s$ instead of $\{\varnothing\}$. We construct $\Omega_s$ through expert curation of established OPF variants from peer-reviewed literature and industry practices (see Appendix B for complete references), ensuring physical validity and engineering soundness. The space $\Omega_s$ comprises three orthogonal dimensions: decision variable extensions $\mathcal{S}_p$ (7 variants), objective function extensions $\mathcal{S}_o$ (15 variants), and constraint extensions $\mathcal{S}_c$ (9 variants). Four domain experts provided design rationales and implementation code for each variant. Complete specifications, including names, mathematical formulations, and design rationales, are provided in Appendix B.

For each sample, a structural modification $s = (s_p, s_o, s_c)$ is constructed by sampling $s_p \sim \mathcal{U}(\mathcal{S}_p)$ to establish the base formulation, then sampling $s_o \subseteq \mathcal{S}_o$ and $s_c \subseteq \mathcal{S}_c$ from their power sets, subject to cardinality bounds $|s_o| \leq K_o$ and $|s_c| \leq K_c$ for computational tractability. This yields $\Omega_s = \mathcal{S}_p \times 2^{\mathcal{S}_o} \times 2^{\mathcal{S}_c}$, where $2^{\mathcal{S}_o}$ and $2^{\mathcal{S}_c}$ denote the power sets of $\mathcal{S}_o$ and $\mathcal{S}_c$, respectively. The remaining components ($\omega, \Delta\boldsymbol{\pi}, \mathcal{R}$) are sampled as in $\mathcal{L}_1$, and combined with $s$ to form $\mathcal{M}^{\mathcal{L}_3}$. Given $\mathcal{M}^{\mathcal{L}_3}$ and instructions $\boldsymbol{\tau}^{\mathcal{L}_3}$, the LLM generates the NL description $\mathcal{P}$ and executable implementation $\mathcal{I}$:

$$(\mathcal{P}, \mathcal{I}) \sim p_{\mathrm{LLM}}\left(\cdot \mid \mathcal{M}^{\mathcal{L}_3}, \boldsymbol{\tau}^{\mathcal{L}_3}\right), \tag{18}$$

where $\boldsymbol{\tau}^{\mathcal{L}_3}$ extends $\boldsymbol{\tau}^{\mathcal{L}_1}$ by incorporating the variant name, the design rationale for generating $\mathcal{P}$, and the implementation code $\mathcal{I}$ for each structural component in $s$.

**Level 4 ($\mathcal{L}_4$): Structural Extensions with Semantic Parameters**

**Task Definition.** $\mathcal{L}_4$ targets scenarios where models simultaneously infer parameter modifications $\Delta\boldsymbol{\pi}$ from scenario-level semantics and accommodate structural modifications $s \in \Omega_s$ that extend the base OPF formulation. For instance, the input "*formulate an optimal transmission switching problem under heatwave conditions with elevated transmission line thermal risk*" requires inferring parameter modifications (e.g., decreased thermal limits) from the scenario description while introducing structural extensions (e.g., optimal transmission switching).

**Data Synthesis.** For each sample $z \in \mathcal{L}_4$, a model specification $\mathcal{M}^{\mathcal{L}_4}$ is sampled uniformly from

$$\Omega_{\mathcal{M}}^{\mathcal{L}_4} := \Omega_{\mathrm{sys}} \times \Omega_{\boldsymbol{\pi}}^{\mathrm{dir}} \times \Omega_s \times \Omega_{\mathcal{R}}, \tag{19}$$

combining the parameter space $\Omega_{\boldsymbol{\pi}}^{\mathrm{dir}}$ from $\mathcal{L}_2$ with the structural modification space $\Omega_s$ from $\mathcal{L}_3$. Structural modification $s = (s_p, s_o, s_c)$ is constructed as in $\mathcal{L}_3$, with relaxed cardinality bounds $|s_o| \leq K_o'$ and $|s_c| \leq K_c'$ where $K_o' > K_o$ and $K_c' > K_c$. Parameter modifications $\Delta\boldsymbol{\pi} \in \Omega_{\boldsymbol{\pi}}^{\mathrm{dir}}$ are converted into scenario fragments $\{c_k\}_{k=1}^K$ via the scenario tree retrieval mechanism from $\mathcal{L}_2$, replacing $\Delta\boldsymbol{\pi}$ in $\mathcal{M}^{\mathcal{L}_4}$ to form $\widetilde{\mathcal{M}}^{\mathcal{L}_4}$. Given $\widetilde{\mathcal{M}}^{\mathcal{L}_4}$ and instructions $\boldsymbol{\tau}^{\mathcal{L}_4}$, the LLM generates $\mathcal{P}$ and $\mathcal{I}$:

$$(\mathcal{P}, \mathcal{I}) \sim p_{\mathrm{LLM}}\left( \cdot \mid \widetilde{\mathcal{M}}^{\mathcal{L}_4}, \boldsymbol{\tau}^{\mathcal{L}_4} \right), \qquad (20)$$

where $\boldsymbol{\tau}^{\mathcal{L}_4}$ combines $\boldsymbol{\tau}^{\mathcal{L}_2}$ (for scenario fragment aggregation) and $\boldsymbol{\tau}^{\mathcal{L}_3}$ (for structural variant integration).

**Data Cleaning and Text Refinement.**

We post-process synthesized samples to enforce semantic validity and improve linguistic diversity. We first apply compatibility filtering using an expert-curated rule table $\mathcal{C}$ to remove semantically invalid pairs of structural edits and parameter patches, e.g., DCOPF omits reactive power and voltage-magnitude constraints, so reactive-power or voltage-bound updates are inapplicable. We discard $z$ if $(s, \Delta\boldsymbol{\pi}) \in \mathcal{C}$. Second, to reduce lexical redundancy among samples with the same model specification $\mathcal{M}$, we diversify the NL prompt $\mathcal{P}_i$ within each equivalence class $\mathcal{Z}_{\mathcal{M}} = \{z_i \mid \mathcal{M}_i = \mathcal{M}\}$ (when $|\mathcal{Z}_{\mathcal{M}}| > 1$) via LLM-based paraphrasing:

$$\mathcal{P}_i' \sim p_{\mathrm{LLM}}(\cdot \mid \mathcal{P}_i, \boldsymbol{\tau}_{\mathrm{diversify}}), \quad \forall z_i \in \mathcal{Z}_{\mathcal{M}}, \qquad (21)$$

where $\boldsymbol{\tau}_{\mathrm{diversify}}$ enforces semantic equivalence while varying surface form. The full cleaning and diversification pipeline is given in Algorithm 5 (see Section C.5).

# 5. ProOPF-B: Benchmark for LLM Performance on OPF Modeling Tasks

ProOPF-B is an expert-annotated benchmark for assessing LLMs' OPF modeling capability across four difficulty levels (Section 4.2). We recruited four power system experts to curate representative OPF modeling tasks from peer-reviewed literature and to provide aligned reference implementations using a MATPOWER-based toolchain. ProOPF-B comprises 121 test cases spanning all levels (Level 1: 36; Level 2: 30; Level 3: 29; Level 4: 26) and is evaluated via two end-to-end pipelines, distinguished by whether $\mathcal{P}$ specifies concrete parameter instantiations: *concrete* OPF modeling (Levels 1/3) and *abstract* OPF modeling (Levels 2/4), as illustrated in Figure 5. All benchmark instances were audited to ensure no overlap with ProOPF-D. See Appendix E for representative examples and evaluation prompts.

**Concrete OPF Modeling Evaluation (Levels 1/3).** For Levels 1 and 3, each test sample $z$ in ProOPF-B consists of

three components:

$$z = \{\mathcal{P}, \mathcal{I}, f^*\} \qquad (22)$$

where $f^*$ denotes the optimal objective value obtained by executing $\mathcal{I}$ via the MATLAB engine. During evaluation, the LLM receives $\mathcal{P}$ and generates an implementation $\widehat{\mathcal{I}}$, which is executed to obtain $\widehat{f^*}$. A test case is considered correct if and only if $|\widehat{f^*} - f^*| \leq \epsilon$ for a tolerance threshold $\epsilon > 0$ (a small positive number). The complete evaluation workflow is formalized in Algorithm 6.

**Abstract OPF Modeling Evaluation (Levels 2/4).** For Levels 2 and 4, each test sample $z$ in ProOPF-B consists of four components:

$$z = \{\mathcal{P}, \mathcal{I}, \boldsymbol{\pi}, f^*(\boldsymbol{\pi})\} \qquad (23)$$

where $\boldsymbol{\pi}$ denotes a predefined parameter instantiation consistent with the modification trends inferred from $\mathcal{P}$, and $f^*(\boldsymbol{\pi})$ denotes the optimal objective value obtained by executing $\mathcal{I}$ with $\boldsymbol{\pi}$ as input. For evaluation, the LLM receives only $\mathcal{P}$ and generates an implementation $\widehat{\mathcal{I}}$, which is executed with $\boldsymbol{\pi}$ to obtain $\widehat{f^*}(\boldsymbol{\pi})$. A test case is considered correct if and only if $|\widehat{f^*}(\boldsymbol{\pi}) - f^*(\boldsymbol{\pi})| \leq \epsilon$. The complete evaluation workflow is formalized in Algorithm 7.

# 6. Experiment and Analysis

## 6.1. Baselines and Settings

**Baselines and Evaluation Setting.** We evaluate models on ProOPF-B across four difficulty levels using objective value accuracy computed from the generated implementations, as detailed in section 5 and Appendix F. As baselines, we consider the **GPT-family** (Achiam et al., 2023), **Claude-family** (Anthropic, 2024), **DeepSeek-family** (Liu et al., 2024), and **Qwen3-family** (Yang et al., 2025a) models in both zero-shot and few-shot settings. For the supervised fine-tuning (SFT) setting, we use **Qwen3-30B-A3B** as the pre-trained model and finetune it on ProOPF-D.

**Fine-tuning Setting.** We fine-tuned Qwen3-30B-A3B-Instruct on ProOPF-D via the VERL framework (Sheng et al., 2025) using $8 \times$ A100 (80GB) GPUs. The training employs a learning rate of $1 \times 10^{-4}$, a global batch size of 8, and a maximum sequence length of 8192 tokens.

## 6.2. Baselines Evaluation on ProOPF-B

Table 1 presents the performance of six state-of-the-art LLMs on ProOPF-B under few-shot and zero-shot settings.

**From Explicit to Semantic.** Model performance exhibits a sharp negative correlation with task abstraction. As shown in Table 1, leading models like Gemini 3.0 Pro and DeepSeek V3.2 demonstrate strong proficiency in explicit

*Table 1.* Model Performance(%) on Four Levels in ProOPF-B

| Setting | Model | Concrete Modeling | | Abstract Modeling | | Average |
|---|---|---|---|---|---|---|
| | | Level 1 | Level 3 | Level 2 | Level 4 | |
| *Few-shot* | GPT-5.2 | 33.33 | 10.34 | 6.67 | 0.00 | 14.05 |
| | Claude 4.5 Sonnet | 77.78 | **37.93** | 6.67 | 0.00 | 33.88 |
| | DeepSeek V3.2 | **94.44** | 6.90 | 0.00 | 0.00 | 29.75 |
| | Gemini 3.0 Pro | **94.44** | 31.03 | 6.67 | 0.00 | **37.19** |
| | Qwen3-Coder | 80.56 | 0.00 | 0.00 | 0.00 | 23.97 |
| | Qwen3-30B-A3B | 50.00 | 0.00 | 0.00 | 0.00 | 14.88 |
| *Zero-shot* | GPT-5.2 | 25.00 | 5.56 | 6.67 | 0.00 | 10.42 |
| | Claude 4.5 Sonnet | 66.67 | 5.56 | 6.67 | 0.00 | 22.82 |
| | DeepSeek V3.2 | **88.89** | 0.00 | 0.00 | 0.00 | 26.45 |
| | Gemini 3.0 Pro | 77.78 | **13.89** | 0.00 | 0.00 | 26.47 |
| | Qwen3-Coder | 13.89 | 5.56 | 0.00 | 0.00 | 5.46 |
| | Qwen3-30B-A3B | 0.00 | 0.00 | 0.00 | 0.00 | 0.00 |
| *SFT (ProOPF-D)* | Qwen3-30B-A3B (few-shot) | 75.00 | 20.70 | **33.30** | **11.54** | 35.53 |
| | Qwen3-30B-A3B (zero-shot) | 63.90 | 10.30 | **23.30** | **7.69** | **27.26** |

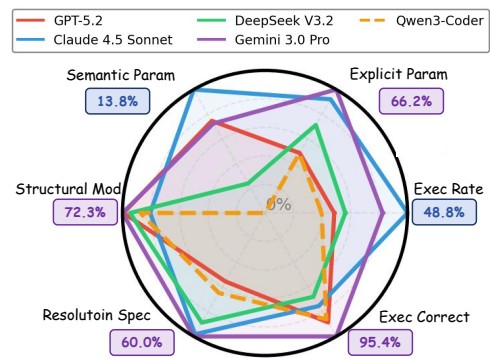

*Figure 4.* Six-Dimensional Capability Radar Chart. Each axis represents a fundamental competency in OPF modeling (see Appendix G for detailed definitions). All values are expressed as percentages.

parametric modifications, achieving **94.44%** accuracy on Level 1. However, the introduction of semantic ambiguity in Level 2 triggers a catastrophic collapse, with accuracy rates plummeting near zero. Crucially, cause the compounding complexity of structural modifications coupled with semantic parameter inference renders the task intractable for Level 4, all models result in **0.00%** success rate.

**From Few-shot to Zero-shot.** The dependency on in-context learning is highly uneven across task types for current models. For Level 1, all models experience performance declines ranging **from 5% to 18%**. More critically, the removal of few-shot examples triggers a collapse in structural modification capabilities, with relative accuracy plummeting by **50% to over 85%** across leading baselines, indicating that current models depend heavily on in-context scaffolding to navigate topological constraints.

**Diagnostic Failure Analysis.** Here we decompose capability limitations using the six-dimensional framework, isolating distinct failure modes:

1. *The Implementation Gap:* Most models exhibit a strong conceptual understanding of structural changes, achieving *Structural Modification Identification* scores between **60%–72%**. However, it fails to translate this into working code. It indicates that the bottleneck lies not in recognizing *what to change*, but in a deficiency of specific *resolution specification* and API knowledge required to implement these changes without errors.

2. *The Non-Executable Expert:* A distinct pathology is observed in models like GPT-5.2, characterized by a high *Executable Correctness Rate* with a critically low *Executable Rate*. It implies that the model generates logically sound OPF formulations that are rendered useless by trivial syntax errors or invalid solver config-

urations, highlighting a fragility in basic coding proficiency despite high-level domain logic.

3. *The Semantic Barrier:* The universal failure on Levels 2 and 4 is directly attributable to deficits in *Semantic Parameter Inference*, where scores hover near zero across all baselines. Unlike structural tasks where failure occurs at the implementation stage, here the failure is foundational: models are unable to map qualitative descriptors (e.g., "heavy load") to quantitative domain standards without explicit numerical grounding.

### 6.3. Supervised Fine-tuning on ProOPF-D

To further validate the effectiveness of ProOPF-D, we fine-tune Qwen3-30B-A3B using the dataset. As shown in Table 1, SFT yields substantial improvement. The most profound improvement occurs at Level 2, where few-shot accuracy surges **from 0.00% to 33.30%**. Notably, on Level-4, the most challenging setting that combines semantic inference with structural reformulation which is completely unsolved by all baseline models, the fine-tuned model attains **11.54%** accuracy. Under zero-shot evaluation, the fine-tuned model also achieves at least a **7.69%** gain across all four levels, indicating that ProOPF-D provides effective, structure-preserving supervision in OPF tasks. We also verify the portability of ProOPF-D supervision on Llama-3.1-8B, where SFT similarly lifts performance from 0.00% to nontrivial accuracy across all four levels (detailed in Appendix H.3).

### 6.4. Data Scaling Analysis

To investigate the impact of training data size on model performance, we conduct a data scaling analysis. We fine-tune the model using subsets of ProOPF-D containing 4K, 8K, and the full 12K training samples, maintaining an equal

*Table 2.* Performance comparison (%) of models fine-tuned with varying amounts of training data across different difficulty levels.

| Training Data | Concrete Modeling | | Abstract Modeling | |
|---|---|---|---|---|
| | Level 1 | Level 3 | Level 2 | Level 4 |
| 4K | 22.20 | 10.30 | 0.00 | 0.00 |
| 8K | 50.00 | 10.30 | 16.67 | 0.00 |
| 12K | 75.00 | 33.30 | 20.70 | 11.54 |

distribution across the four difficulty levels. The evaluation results on ProOPF-B are summarized in Table 2.

These results show a clear positive scaling trend across all difficulty levels as the training data size increases. Notably, the performance gains on Levels 1 and 2 begin to taper off when scaling from 8K to 12K samples. In contrast, the performance on the more complex structural tasks (Levels 3 and 4) improves markedly in this regime, with Level 4 accuracy increasing from 0.00% to 11.54%. These findings suggest that while smaller datasets may suffice for simpler parametric modifications, further expanding the dataset, particularly with Level-3 and Level-4 instances, is likely to yield continued improvements for complex OPF modeling tasks.

### 6.5. OOD Generalization Analysis on ProOPF-D/B

To thoroughly evaluate the robustness of the fine-tuned model, we conduct out-of-distribution (OOD) generalization analyses across three dimensions. For a complete assessment, the detailed results on unseen grid topologies and novel operational scenarios are presented in Appendix H.1. In this section, we compare the performance of a model trained exclusively on simpler tasks (Levels 1–3) against one trained on the full dataset (Levels 1–4). As shown in Table 3, omitting Level-4 training data severely degrades few-shot performance on Level-4 tasks from 11.54% to 3.84%. This performance drop indicates that while partial transfer from simpler tasks occurs, explicit supervision on combined structural and parametric modifications remains crucial for mastering the most difficult OPF scenarios.

*Table 3.* Compositional generalization performance (%) on Level 4 when trained on different data splits.

| Training Split | Zero-shot | Few-shot |
|---|---|---|
| Levels 1–4 (Full) | 7.69 | 11.54 |
| Levels 1–3 Only | 3.84 | 3.84 |

### 6.6. Tool-Augmented OPF Modeling Analysis

To assess whether stronger tool and prompt support can mitigate the performance degradation observed on abstract modeling tasks (Levels 2 and 4), we conduct additional experiments by augmenting two state-of-the-art LLMs (GPT-5.2 and Claude 4.5 Sonnet) with solver-oriented reference information. Specifically, we enrich their prompts with: (1) explicit instructions for modifying objectives and constraints within the MATPOWER framework; (2) reference implementations of representative OPF variants; and (3) guidelines for translating qualitative scenario descriptions into quantitative parameter changes.

*Table 4.* Performance comparison (%) of models on Levels 2 and 4 with original and enhanced (tool-augmented) prompts.

| Model | Original | | Enhanced | |
|---|---|---|---|---|
| | Level 2 | Level 4 | Level 2 | Level 4 |
| GPT-5.2 | 6.67 | 0.00 | 26.67 | 26.92 |
| Claude 4.5 Sonnet | 6.67 | 0.00 | 23.33 | 23.08 |

The results, presented in Table 4, demonstrate that providing substantial solver-oriented support yields significant improvements. Both models exhibit a nearly fourfold increase in accuracy on Level 2 and successfully solve over 20% of the previously intractable Level-4 instances. These findings indicate that while current LLMs lack intrinsic domain knowledge for complex OPF formulations, their reasoning capabilities can be effectively unlocked through targeted tool augmentation and context enrichment, highlighting a promising direction for future research in tool-augmented OPF modeling.

## 7. Conclusion and Future Work

We introduce ProOPF-D and ProOPF-B, the first dataset and benchmark for evaluating LLMs on professional-grade OPF modeling. ProOPF-D contains 12K instances across four difficulty levels, covering parametric updates and structural extensions of a canonical OPF. ProOPF-B provides 121 expert-annotated test cases for end-to-end evaluation. Results show that while LLMs handle explicit parameter modifications, they struggle with semantic parameter inference and structural modeling, highlighting the necessity of domain-aware training and supervision.

Despite these contributions, several limitations highlight avenues for future research. First, ProOPF currently focuses on transmission-level optimization modeling; extending the benchmark to distribution networks and microgrids is a crucial next step. Second, while execution-based evaluation ensures functional correctness, finer-grained validation of intermediate steps would yield deeper insights into LLM reasoning. Finally, expanding the dataset for complex structural tasks (Levels 3 and 4) will further advance LLM capabilities in power system optimization.

## Impact Statement

This paper presents work whose goal is to advance the field of Machine Learning. There are many potential societal consequences of our work, none which we feel must be specifically highlighted here.

## Acknowledgements

This work was supported in part by the Smart Grid-National Science and Technology Major Project (2025ZD0803600/2025ZD0803604), the National Natural Science Foundation of China under Grant 72571007 and Grant 72595830/72595831, and the Beijing Nova Program (No. 20250484850).

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

# A. Details of Model Specification in ProOPF-D

## A.1. Base System Pool $\Omega_{sys}$

The base system pool $\Omega_{sys}$ comprises 63 base power systems that serve as reference configurations for synthesizing ProOPF-D samples. These systems are curated from three primary sources: (1) the MATPOWER test case library (Zimmerman et al., 2010), which provides standardized power system models widely used in research and education; (2) IEEE PES Test Feeder Archive (IEEE Dataport), which hosts peer-reviewed test systems representing diverse network topologies and operating characteristics; and (3) academic publications in power systems journals and conferences, which document real-world system configurations and benchmark problems (Shen et al., 2025a;c;b; Zuo et al., 2025). Each system $\omega \in \Omega_{sys}$ is associated with base parameters $\boldsymbol{\pi}_{sys}(\omega)$ encoding network topology, generator cost coefficients, branch impedances, load profiles, and operational limits. The systems span a wide range of scales, from small-scale test networks (e.g., 9-bus systems) to large-scale transmission networks (e.g., 300+ bus systems), enabling comprehensive evaluation across varying problem complexities.

A summary of the base system pool, including network topology statistics and system descriptions, is provided in Table 5. For each system, the table reports the number of buses $|\mathcal{N}|$, branches $|\mathcal{E}|$, and generators $|\mathcal{G}|$, along with brief descriptions of system characteristics and source information.

*Table 5.* Base system pool $\Omega_{sys}$ summary statistics.

| System Name $\omega$ | $|\mathcal{N}|$ | $|\mathcal{E}|$ | $|\mathcal{G}|$ | Description |
|---|---|---|---|---|
| case4_dist | 4 | 3 | 2 | 4-bus example radial distribution system case |
| case4gs | 4 | 4 | 2 | 4-bus example case from Grainger & Stevenson |
| case5 | 5 | 6 | 5 | modified 5-bus PJM example case from Rui Bo |
| case6ww | 6 | 11 | 3 | 6-bus example case from Wood & Wollenberg |
| case9 | 9 | 9 | 3 | 9-bus example case from Chow |
| case10ba | 10 | 9 | 1 | 10-bus radial distribution system from Baghzouz and Ertem |
| case12da | 12 | 11 | 1 | 12-bus radial distribution system from Das, Nagi, and Kothari |
| case14 | 14 | 20 | 5 | IEEE 14-bus case |
| case15da | 15 | 14 | 1 | 15-bus radial distribution system from Das, Kothari, and Kalam |
| case15nbr | 15 | 14 | 1 | 15-bus radial distribution system from Battu, Abhyankar, Senroy |
| case16am | 15 | 14 | 1 | 16-bus radial distribution system from Das, Kothari, and Kalam |
| case16ci | 16 | 16 | 3 | 16-bus distribution system from Civanlar, Grainger, Yin, and Lee |
| case17me | 17 | 16 | 1 | 17-bus radial distribution system from Mendoza, Morales, Lopez, et. al. |
| case18 | 18 | 17 | 1 | 18-bus radial distribution system from Grady, Samotyj and Noyola |
| case18nbr | 18 | 17 | 1 | 18-bus radial distribution system from Battu, Abhyankar, Senroy |
| case22 | 22 | 21 | 1 | 22-bus radial distribution system from Raju, Murthy and Ravindra |
| case24_ieee_rts | 24 | 38 | 33 | IEEE RTS 24-bus case |
| case28da | 28 | 27 | 1 | 28-bus radial distribution system from Das, Nagi, and Kothari |
| case30 | 30 | 41 | 6 | 30-bus case, based on IEEE 30-bus case |
| case_ieee30 | 30 | 41 | 6 | IEEE 30-bus case |
| case33bw | 33 | 37 | 1 | 33-bus radial distribution system from Baran and Wu |
| case33mg | 33 | 37 | 1 | 33-bus radial distribution system from Kashem, Ganapathy, Jasmon and Buhari |
| case34sa | 34 | 33 | 1 | 34-bus radial distribution system from Salama and Chikhani |
| case38si | 38 | 37 | 1 | 38-bus radial distribution system from Singh and Misra |
| case39 | 39 | 46 | 10 | 39-bus New England case |
| case51ga | 51 | 50 | 1 | 51-bus radial distribution system from Gampa and Das |
| case51he | 51 | 50 | 1 | 51-bus radial distribution system from Hengsritawat, Tayjasanant and Nimpitiwan |
| case57 | 57 | 80 | 7 | IEEE 57-bus case |
| case69 | 69 | 68 | 1 | 69-bus radial distribution system from Baran and Wu |
| case70da | 70 | 76 | 2 | 70-bus distribution system from Das |
| case_RTS_GMLC | 73 | 120 | 158 | 96-machine, 73-bus Reliability Test System |
| case74ds | 74 | 73 | 1 | 74-bus radial distribution system from Myint and Naing |
| case85 | 85 | 84 | 1 | 85-bus radial distribution system from Das, Kothari and Kalam |
| case89pegase | 89 | 210 | 12 | 89-bus portion of European transmission system from PEGASE project |
| case94pi | 94 | 93 | 1 | 94-bus radial distribution system from Pires, Antunes and Martins |
| case118 | 118 | 186 | 54 | IEEE 118-bus case |
| case118zh | 118 | 132 | 1 | 118-bus radial distribution system from Zhang, Fu and Zhang |

| System Name | $|\mathcal{N}|$ | $|\mathcal{E}|$ | $|\mathcal{G}|$ | Description |
|---|---|---|---|---|
| | | | | **Table 5 – continued from previous page** |
| case136ma | 136 | 156 | 1 | 136-bus radial distribution system from Mantovani, Casari and Romero |
| case141 | 141 | 140 | 1 | 141-bus radial distribution system from Khodr, Olsina, De Jesus and Yusta |
| case145 | 145 | 453 | 50 | IEEE 145-bus case, 50 generator dynamic test case |
| case_ACTIVSg200 | 200 | 245 | 49 | 200-bus Illinois synthetic model |
| case300 | 300 | 411 | 69 | IEEE 300-bus case |
| case_ACTIVSg500 | 500 | 597 | 90 | 500-bus South Carolina synthetic model |
| case1354pegase | 1354 | 1991 | 260 | 1354-bus portion of European transmission system from PEGASE project |
| case1888rte | 1888 | 2531 | 298 | 1888-bus snapshot of VHV French transmission system from RTE |
| case1951rte | 1951 | 2596 | 392 | 1951-bus snapshot of VHV French transmission system from RTE |
| case_ACTIVSg2000 | 2000 | 3206 | 544 | 2000-bus Texas synthetic model |
| case2383wp | 2383 | 2896 | 327 | Polish system - winter 1999-2000 peak |
| case2736sp | 2736 | 3504 | 420 | Polish system - summer 2004 peak |
| case2737sop | 2737 | 3506 | 399 | Polish system - summer 2004 off-peak |
| case2746wop | 2746 | 3514 | 514 | Polish system - winter 2003-04 off-peak |
| case2746wp | 2746 | 3514 | 520 | Polish system - winter 2003-04 evening peak |
| case2848rte | 2848 | 3776 | 548 | 2848-bus snapshot of VHV French transmission system from RTE |
| case2868rte | 2868 | 3808 | 600 | 2868-bus snapshot of VHV French transmission system from RTE |
| case2869pegase | 2869 | 4582 | 510 | 2869-bus portion of European transmission system from PEGASE project |
| case3012wp | 3012 | 3572 | 502 | Polish system - winter 2007-08 evening peak |
| case3120sp | 3120 | 3693 | 505 | Polish system - summer 2008 morning peak |
| case3375wp | 3374 | 4161 | 596 | Polish system plus - winter 2007-08 evening peak |
| case6468rte | 6468 | 9000 | 1296 | 6468-bus snapshot of VHV and HV French transmission system from RTE |
| case6470rte | 6470 | 9005 | 1331 | 6470-bus snapshot of VHV and HV French transmission system from RTE |
| case6495rte | 6495 | 9019 | 1373 | 6495-bus snapshot of VHV and HV French transmission system from RTE |
| case6515rte | 6515 | 9037 | 1389 | 6515-bus snapshot of VHV and HV French transmission system from RTE |
| case9241pegase | 9241 | 16049 | 1445 | 9241-bus portion of European transmission system from PEGASE project |

## A.2. Parametric Modifications $\Delta\pi$

Parametric modifications $\Delta\pi$ are organized as a finite set of parameter patches $\{\delta_k\}_{k=1}^K$, where each patch $\delta_k$ encodes a single parameter modification operation. Each patch $\delta_k$ is a structured tuple that specifies which component parameter is being modified, how it is modified, and the modification value or direction. The patch structure enables compact representation of multiple parameter changes while preserving the relationship between components, their parameters, and modification semantics.

Each parameter patch $\delta_k$ contains several key attributes that collectively define a parameter modification. The **component** attribute identifies the power system component type (bus, generator, or branch) to which the modification applies. For bus and generator components, the **bus_id** attribute specifies the target bus identifier. For branch components, the **fbus** and **tbus** attributes specify the from-bus and to-bus identifiers, respectively. The **target_parameter** attribute identifies the specific parameter being modified (e.g., active power demand, generator capacity limit, branch reactance). The modification operation is specified through either an **operation** attribute (for Level 1 samples with explicit values) or a **direction** attribute (for Level 2 samples with semantic constraints). When an explicit operation is used, the **value** attribute provides the numerical modification value. Common operations include Scale (multiplicative scaling) and Set (absolute value assignment), while common directions include Increase, Decrease, and Set zero.

The complete specification of patch attributes, their data types, allowed values, and usage examples are provided in Table 6. Complete sample examples demonstrating parameter patch usage across all difficulty levels are provided in Appendix D. This structured representation enables systematic sampling of parameter modifications while ensuring physical consistency and semantic validity across different difficulty levels.

*Table 6.* Parameter patch $\delta_k$ attribute specifications.

| Attribute | Description | Example |
|---|---|---|
| `component` | Component type identifier | `"bus"`, `"gen"`, `"branch"` |
| `bus_id` | Bus identifier (for bus/gen components) | 1 (bus 1), 32 (bus 32) |
| `fbus` | From-bus identifier (for branch components) | 2 (from bus 2) |
| `tbus` | To-bus identifier (for branch components) | 25 (to bus 25) |
| `target_parameter` | Parameter identifier being modified | `"PD"` (active power demand), `"PMAX"` (max generator output), `"BR_X"` (branch reactance), `"VMIN"` (min voltage) |
| `operation` | Modification operation type (Level 1& 3) | `"Scale"` (multiplicative), `"Set"` (absolute) |
| `direction` | Modification direction constraint (Level 2&4) | `"Increase"`, `"Decrease"`, `"Set zero"` |
| `value` | Numerical modification value (Level 1&3) | 1.5 (scale factor), 500 (absolute value in MW) |

## A.3. Structural Modifications $s$

Structural modifications $s$ are organized as a triple $s = \{s_p, s_c, s_o\}$ that encodes modifications to the problem formulation structure. Each component corresponds to a specific aspect of structural variation: $s_p$ specifies the problem type (decision variable extensions), $s_c$ specifies constraint extensions, and $s_o$ specifies objective function modifications.

The **problem** component $s_p$ corresponds to decision variable extensions that fundamentally alter the problem structure by introducing new decision variables or reformulating the problem formulation. Common problem types include `"ACOPF"` (canonical AC optimal power flow), `"DCOPF"` (DC approximation), `"ED"` (economic dispatch), `"UC"` (unit commitment), and `"OTS"` (optimal transmission switching). The problem type establishes the base formulation structure, upon which objective and constraint extensions can be applied.

The **objective_modification** component $s_o$ encodes modifications to the objective function. Each objective modification is a structured object containing fields such as `op` (operation type, e.g., `"add"` to add a new term), `name` (identifier for the modification, e.g., `"angle_difference_penalty"`), `form` (mathematical formulation), and additional parameters (e.g., `beta` for penalty coefficients). Multiple objective modifications can be combined to form composite objective functions.

The **constraint_modification** component $s_c$ specifies constraint extensions that add new constraints to the base problem formulation. Similar to objective modifications, constraint modifications are structured objects containing fields such as `op` (operation type), `name` (constraint identifier, e.g., `"N-1_security"`), `form` (constraint formulation), and associated parameters. Constraint modifications enable the incorporation of operational requirements such as security constraints, voltage stability limits, and renewable integration constraints.

The complete specification of structural modification components, their fields, data types, and usage examples are provided in Table 7. Complete sample examples demonstrating structural modification usage across all difficulty levels are provided in Appendix D. The complete catalog of available structural variants in the seed set $\mathcal{S}_{\text{seed}}$ is detailed in Appendix B.

*Table 7.* Structural modification $s$ component and field specifications.

| Component/Field | Description | Example |
|---|---|---|
| `problem` ($s_p$) | Problem type identifier | `"ACOPF"`, `"DCOPF"`, `"ED"`, `"UC"`, `"OTS"` |
| `objective_modification` ($s_o$) | Objective function modification object | `{"op": "add", "name": "angle_difference_penalty", ...}` |
| `op` | Operation type for objective modification | `"add"` (add new term), `"replace"` (replace objective) |
| `name` | Identifier for the objective modification | `"angle_difference_penalty"`, `"renewable_curtailment"` |

**Table 7 – continued from previous page**

| Component/Field | Description | Example |
|---|---|---|
| `form` | Mathematical formulation of the objective term | `"beta * ||E*Va||_2^2"` |
| `beta` | Penalty coefficient parameter | `10` (penalty coefficient) |
| `constraint_modification` $(s_c)$ | Constraint extension object | `{"op": "add", "name": "N-1_security", ...}` |
| `op` | Operation type for constraint modification | `"add"` (add new constraint) |
| `name` | Identifier for the constraint modification | `"N-1_security"`, `"voltage_stability"` |
| `form` | Mathematical formulation of the constraint | `"L_i(V,theta) <= L^max"` |

## A.4. Resolution Specification $\mathcal{R}$

The resolution specification $\mathcal{R}$ encodes solver configuration parameters that control how the OPF model is solved and how results are reported. The specification contains three main categories of parameters: solver selection, convergence criteria, and output configuration. Complete specifications of resolution parameters, their descriptions, and usage examples are provided in Table 8. Complete sample examples demonstrating resolution specification usage across all difficulty levels are provided in Appendix D.

*Table 8.* Resolution specification $\mathcal{R}$ parameter categories and field specifications.

| Category/Field | Description | Example |
|---|---|---|
| `solver` | Optimization solver identifier | `"MIPS"`, `"IPOPT"`, `"KNITRO"` |
| `opf_violation` | Maximum acceptable constraint violation tolerance (convergence criterion) | `1e-8` (tight tolerance), `1e-6` (loose tolerance) |
| `output` | Output configuration specifying precision and verbosity | Numerical precision settings, verbosity levels |

## B. Seed Set of OPF Structural Variants

This appendix presents the curated seed set $\mathcal{S}_{\text{seed}}$ of OPF structural variants used for synthesizing Level 3 and Level 4 data in ProOPF-D. The seed set defines the structural modification space for generating OPF problem variants beyond the canonical AC-OPF formulation. The variants included in $\mathcal{S}_{\text{seed}}$ are drawn from established formulations reported in peer-reviewed power systems literature and operational practices adopted in industry. Each variant represents a well-documented extension or reformulation that addresses specific operational requirements, computational constraints, or modeling objectives encountered in real-world power system planning and operation. Detailed references for each variant, including corresponding academic publications and industrial case studies, are provided in the supplementary material: OPF variants reference list.

The structural modification space is organized into three orthogonal dimensions: **decision variable extensions** $\mathcal{S}_p$ (8 variants including the base AC-OPF formulation), **objective function extensions** $\mathcal{S}_O$ (15 variants), and **constraint extensions** $\mathcal{S}_c$ (9 variants). Decision variable extensions define the fundamental problem formulation, encompassing variants that introduce new decision variables or reformulate the problem structure (e.g., economic dispatch, DCOPF, unit commitment, optimal transmission switching). These variants often correspond to distinct power system optimization problems that share structural similarities with OPF. Objective function extensions and constraint extensions represent modular components that can be combined to extend a base problem formulation. The complete structural modification space is given by the Cartesian product $\mathcal{S}_p \times 2^{\mathcal{S}_O} \times 2^{\mathcal{S}_c}$, where $2^{\mathcal{S}_O}$ and $2^{\mathcal{S}_c}$ denote the power sets (allowing combinations of multiple extensions) of objective function and constraint extensions, respectively.

As described in subsection 4.2, each Level 3 or Level 4 sample is constructed by first sampling a decision variable extension $s_p \sim \mathcal{U}(\mathcal{S}_p)$ to establish the base problem formulation, then sampling compatible objective function and constraint extensions from $\mathcal{S}_O$ and $\mathcal{S}_c$ (potentially as combinations), and finally combining the resulting structural modification $s$ with parameter modifications $\Delta\pi$ sampled from $\Omega_\pi$.

### B.1. Decision Variable Extensions $\mathcal{S}_p$

This category comprises 8 problem formulations, including the canonical AC-OPF (ID 1) as the base formulation studied in this work, along with 7 structural variants that define alternative problem formulations by introducing new decision variables or reformulating the problem structure. These variants capture distinct power system optimization problems that share structural similarities with OPF, including economic dispatch that neglects network constraints, DC approximation for computational tractability, temporal coupling through multi-period optimization and unit commitment, topology control via transmission switching, and load curtailment for feasibility recovery. Each variant $s_p \in \mathcal{S}_p$ establishes the base problem structure, upon which objective function and constraint extensions can be applied. Table 9 present the complete problem formulation and key features for each formulation.

*Table 9.* Decision variable variants in OPF problems.

| ID | Problem Type | Formulation | Descriptions |
|---|---|---|---|
| 1 | AC OPF | $$\min_{P_g,Q_g,V,\theta} \sum_{g\in\mathcal{G}} C_g(P_g)$$ subject to: $$P_g - P_d = \sum_k V_i V_k Y_{ik} \cos(\theta_i - \theta_k - \alpha_{ik}), \quad \forall i$$ $$Q_g - Q_d = \sum_k V_i V_k Y_{ik} \sin(\theta_i - \theta_k - \alpha_{ik}), \quad \forall i$$ $$P_g^{\min} \le P_g \le P_g^{\max}, \quad Q_g^{\min} \le Q_g \le Q_g^{\max}, \quad \forall g$$ $$V^{\min} \le V \le V^{\max}, \quad |S_{ik}| \le S_{ik}^{\max}, \quad \forall i,k$$ | Canonical OPF formulation studied in this work; optimizes generator dispatch $(P_g, Q_g)$ and bus voltages $(V, \theta)$ subject to nonlinear AC power flow equations and operational limits; serves as the base problem formulation upon which objective function and constraint extensions are applied |
| 2 | DC OPF | $$\min_{P_g,\theta} \sum_{g\in\mathcal{G}} C_g(P_g)$$ subject to: $$\sum_{g\in\mathcal{G}_i} P_g - P_{d,i} = \sum_{k\in\mathcal{N}_i} B_{ik}(\theta_i - \theta_k), \quad \forall i \in \mathcal{N}$$ $$|B_{ik}(\theta_i - \theta_k)| \le F_{ik}^{\max}, \quad \forall(i,k) \in \mathcal{L}$$ $$P_g^{\min} \le P_g \le P_g^{\max}, \quad \forall g$$ $$\theta_{\text{ref}} = 0$$ | Simplifies AC-OPF by eliminating voltage magnitude and reactive power variables; linearizes power flow using small-angle approximation; widely used in large-scale system operations, market clearing, and real-time dispatch due to computational efficiency |

**Table 9 – continued from previous page**

| ID | Problem Type | Formulation | Descriptions |
|---|---|---|---|
| 3 | Multi-period OPF | $\min\limits_{P_g, Q_g, V, \theta} \sum\limits_{t=1}^{T} \sum\limits_{g \in \mathcal{G}} C_g(P_{gt})$ 

 subject to: 
 $P_{gt} - P_{dt} = \sum_k V_{it} V_{kt} Y_{ik} \cos(\theta_{it} - \theta_{kt} - \alpha_{ik}), \quad \forall i, t$ 
 $Q_{gt} - Q_{dt} = \sum_k V_{it} V_{kt} Y_{ik} \sin(\theta_{it} - \theta_{kt} - \alpha_{ik}), \quad \forall i, t$ 
 $P_g^{\min} \leq P_{gt} \leq P_g^{\max}, \quad Q_g^{\min} \leq Q_{gt} \leq Q_g^{\max}, \quad \forall g, t$ 
 $V^{\min} \leq V_{it} \leq V^{\max}, \quad |S_{ikt}| \leq S_{ik}^{\max}, \quad \forall i, k, t$ 
 $P_{gt} - P_{g,t-1} \leq R_g^{\text{up}}, \quad P_{g,t-1} - P_{gt} \leq R_g^{\text{dn}}, \quad \forall g, t \geq 2$ | Extends single-period OPF over planning horizon $T$ by coupling consecutive periods via generator ramping constraints; optimizes generation schedule considering temporal variations in load and renewable generation; units remain committed throughout horizon |
| 4 | Security-constrained unit commitment | $\min\limits_{u, P_g, Q_g, V, \theta} \sum\limits_{t=1}^{T} \sum\limits_{g \in \mathcal{G}} [C_g(P_{gt}) u_{gt} + SC_g(u_{gt}, u_{g,t-1})]$ 

 where $SC_g = S_g^{\text{up}}(1 - u_{g,t-1}) u_{gt} + S_g^{\text{dn}} u_{g,t-1}(1 - u_{gt})$ 
 subject to: 
 AC power flow equations $\forall t$ 
 $u_{gt} P_g^{\min} \leq P_{gt} \leq u_{gt} P_g^{\max}, \quad u_{gt} Q_g^{\min} \leq Q_{gt} \leq u_{gt} Q_g^{\max}, \quad \forall g, t$ 
 $V^{\min} \leq V_{it} \leq V^{\max}, \quad |S_{ikt}| \leq S_{ik}^{\max}, \quad \forall i, k, t$ 
 $P_{gt} - P_{g,t-1} \leq R_g^{\text{up}} u_{gt}, \quad P_{g,t-1} - P_{gt} \leq R_g^{\text{dn}} u_{g,t-1}, \quad \forall g, t$ 
 $\sum_{\tau = t - T_g^{\text{up}} + 1}^{t} (1 - u_{g\tau}) = 0, \quad \sum_{\tau = t - T_g^{\text{dn}} + 1}^{t} u_{g\tau} = 0, \quad \forall g, t$ 
 $u_{gt} \in \{0, 1\}, \quad \forall g, t$ | Integrates unit commitment with multi-period dispatch optimization; determines optimal on/off schedule and power output for each generator over time horizon; enforces inter-temporal constraints including minimum up/down times and startup-dependent ramping limits |
| 5 | Economic dispatch | $\min\limits_{P_g} \sum\limits_{g \in \mathcal{G}} C_g(P_g)$ 

 subject to: 
 $\sum_{g \in \mathcal{G}} P_g = \sum_{d \in \mathcal{D}} P_d$ 
 $P_g^{\min} \leq P_g \leq P_g^{\max}, \quad \forall g$ | Simplest formulation that determines generator active power outputs to satisfy total demand at minimum cost; neglects network topology, power flow constraints, and reactive power; provides fast dispatch decisions assuming transmission capacity is sufficient; serves as initial solution for more detailed OPF models |
| 6 | Optimal transmission switching | $\min\limits_{P_g, \theta, z} \sum\limits_{g \in \mathcal{G}} C_g(P_g)$ 

 subject to: 
 $P_g - P_d = \sum_k B_{ik} z_{ik}(\theta_i - \theta_k), \quad \forall i$ 
 $-z_{ik} P_{ik}^{\max} \leq B_{ik}(\theta_i - \theta_k) \leq z_{ik} P_{ik}^{\max}, \quad \forall (i, k) \in \mathcal{L}$ 
 $P_g^{\min} \leq P_g \leq P_g^{\max}, \quad \forall g$ 
 $z_{ik} \in \{0, 1\}, \quad \forall (i, k) \in \mathcal{L}$ | Introduces binary variables for line switching status to enable network topology control; uses DC power flow approximation (linearized) for computational tractability; allows strategic line outages to relieve congestion and reduce generation cost; results in mixed-integer linear program (MILP) |
| 7 | Load shedding | $\min\limits_{P_g, Q_g, V, \theta, L^{\text{shed}}} \sum\limits_{g \in \mathcal{G}} C_g(P_g) + \sum\limits_{d \in \mathcal{D}} W_d L_d^{\text{shed}}$ 

 subject to: 
 $P_g - P_d + L_d^{\text{shed}} = \sum_k V_i V_k Y_{ik} \cos(\theta_i - \theta_k - \alpha_{ik}), \quad \forall i$ 
 $Q_g - Q_d = \sum_k V_i V_k Y_{ik} \sin(\theta_i - \theta_k - \alpha_{ik}), \quad \forall i$ 
 $P_g^{\min} \leq P_g \leq P_g^{\max}, \quad Q_g^{\min} \leq Q_g \leq Q_g^{\max}, \quad V^{\min} \leq V \leq V^{\max}$ 
 $0 \leq L_d^{\text{shed}} \leq L_d, \quad \forall d$ | Introduces continuous load shedding variables $L_d^{\text{shed}}$ to ensure problem feasibility when generation capacity is insufficient or network constraints are too restrictive; assigns high penalty cost $W_d$ to represent severe economic and social impact of unserved energy |

## B.2. Objective Function Extensions $\mathcal{S}_O$

This category comprises 15 structural variants that modify or extend the objective function of the base problem formulation. These extensions reflect diverse operational goals encountered in modern power system operation, including economic dispatch with multiple cost components, multi-objective optimization balancing economic and environmental criteria, and security-oriented objectives that improve system robustness. Multiple objective function extensions can be combined to form composite objectives (e.g., minimizing

generation cost while penalizing voltage deviations and line overloads). Table 10 presents the mathematical formulation and design rationale for each variant.

*Table 10.* Objective function variants in OPF problems.

| ID | Objective Name | Formulation | Descriptions |
|---|---|---|---|
| 1 | Real power redispatch cost (RPR) | $RPR = \sum_{i \in N_G} \left( c_{it}^u P_{G,it}^u + c_{it}^d P_{G,it}^d \right)$ | Minimize magnitude and cost of real-power adjustments between reference schedule and redispatched solution |
| 2 | Reactive power generation cost (RGC) | $RGC = \sum_{i \in N_G} \left( a_Q Q_{G,i}^2 + b_Q Q_{G,i} + c_Q \right)$ | Minimize total cost of providing reactive power support from generators |
| 3 | Real power procurement cost (RPC) | $RPC = \sum_{i=1}^{N_{G_p}} P_{pi} \lambda$ | Minimize total payment for purchasing active power from external sources at price $\lambda$ |
| 4 | Reactive power procurement cost (QPC) | $QPC = \sum_{n \in N_B} \sum_{m \in \Omega_n} V_n^2 \frac{Y_{Ch,nm}}{2}$ | Minimize reactive power procurement from shunt capacitors to support bus voltage magnitudes |
| 5 | Real power reserve cost (RRC) | $RRC = \sum_{i \in I_{RU}} f_{u,i}(R_{u,i}) + \sum_{i \in I_{SP}} f_{s,i}(R_{s,i})$ | Minimize capacity payment for multiple types of active power reserves (up, down, spinning, non-spinning) |
| 6 | Reactive reserve insufficiency cost (RRI) | $RRI = -\sum_{i \in N_G} w_i(Q_{G,i}^{\max} - Q_{G,i})$ | Minimize negative reactive power reserve margin (equivalently maximize reserve) to enhance voltage security |
| 7 | System congestion cost (SCC) | $SCC = \frac{1}{N_T} \sum_{l=1}^{N_T} \frac{S_l}{S_l^{\max}}$ | Minimize average utilization level of transmission elements to enhance loadability and reduce congestion |
| 8 | Voltage stability margin cost (VSC) | $VSC = \max(L_i), \quad L_j = \left\| 1 - \sum F_{ji} \frac{V_i}{V_j} \right\|$ | Minimize maximum L-index to increase distance to voltage-collapse points and improve stability margin |
| 9 | Active power loss cost (PLC) | $PLC = \sum_{i=1}^{N_G} P_{Gi} - \sum_{j=1}^{N_L} P_{Lj}$ | Minimize total active power losses in transmission network to improve system efficiency |
| 10 | Environmental emission cost (EEC) | $EEC = \sum_{i,t} \left( \alpha_i P_{G,it}^2 + \beta_i P_{G,it} + \gamma_i + \zeta_i e^{\lambda_i P_{G,it}} \right)$ | Minimize total pollutant emissions ($NO_x$, $SO_2$, $CO_2$) using quadratic-plus-exponential emission functions |
| 11 | Negative transfer capability cost (NTC) | $NTC = -\lambda \sum_{i \in S_L} b_{Pi}$ | Minimize negative transfer capability (equivalently maximize load transfer capacity) to estimate available transfer capability |
| 12 | Negative social welfare cost (NSW) | $NSW = \sum_{j \in G} C_j(x_j) - \sum_{i \in C} B_i(x_i)$ | Minimize generation cost minus consumer utility (equivalently maximize social welfare) to achieve economic efficiency |
| 13 | Voltage deviation penalty cost (VDP) | $VDP = C_{\text{gen}}(P_G) + \beta \sum_{i=1}^{N} (V_i - 1.0)^2$ | Minimize generation cost plus penalty on squared voltage deviations from nominal value (1.0 p.u.) |

**Table 10 – continued from previous page**

| ID | Objective Name | Formulation | Descriptions |
|---|---|---|---|
| 14 | Angle difference penalty cost (ADP) | $ADP = C_{\text{gen}}(P_G) + \beta \sum_{(i,j) \in \mathcal{E}} (\theta_i - \theta_j)^2$ | Minimize generation cost plus penalty on excessive voltage angle differences across transmission lines |
| 15 | Line overload penalty cost (LOP) | $LOP = \sum_g C_g(P_g) - \lambda \sum_\ell s_\ell$ | Minimize generation cost with economic penalty for line thermal limit violations via slack variables $s_\ell$ |

## B.3. Constraint Extensions $\mathcal{S}_c$

This category comprises 9 structural variants that augment the constraint set of the base problem formulation to address security, reliability, uncertainty, and operational flexibility requirements. These extensions reflect critical concerns in modern power system operation, including N-1 security criteria for contingency resilience, probabilistic and robust formulations for uncertainty handling, frequency stability constraints for low-inertia systems, and various reserve requirement formulations for ancillary services. Multiple constraint extensions can be combined to enforce multiple operational requirements simultaneously (e.g., N-1 security constraints together with reserve requirements and frequency stability constraints). Table 11 presents the mathematical formulation and design rationale for each constraint variant.

*Table 11.* Constraint variants in OPF problems.

| ID | Constraint Type | Formulation | Descriptions |
|---|---|---|---|
| 1 | Chance constraint | $\mathbb{P}\{h(x,\xi) \leq 0\} \geq 1 - \varepsilon$ | Limit violation probability under uncertain generation/load; balance economy and risk level for high renewable penetration |
| 2 | Robust constraint | $h(x,\xi) \leq 0, \forall \xi \in \mathcal{U}$ | Maintain feasibility for all uncertainty realizations in bounded uncertainty set; provide worst-case security guarantee |
| 3 | Frequency stability constraint | $\Delta f_{\min} \geq \Delta f^{\text{req}}, \quad \text{RoCoF} \leq \text{RoCoF}^{\max}$ | Limit frequency nadir and rate of change of frequency; reflect inertia and primary frequency response requirements |
| 4 | Reserve limit constraint | $-P_{R,i}^D \leq \Delta P_{G,i} \leq P_{R,i}^U, \quad -Q_{R,i}^D \leq \Delta Q_{G,i} \leq Q_{R,i}^U$ | Explicitly limit dispatch adjustment capability to pre-scheduled reserve ranges for real and reactive power |
| 5 | Regulation up reserve constraint | $R_j^{RU} - \sum_{i \in I_{RU} \cap Z_j} RU_i \leq 0$ | Require sufficient upward regulation reserve capacity within each zone to handle load variations and forecast errors |
| 6 | Spinning reserve constraint | $R_j^{RU} + R_j^{SP} - \sum_{i \in I_{RU} \cap Z_j} RU_i - \sum_{i \in I_{SP} \cap Z_j} SP_i \leq 0$ | Ensure adequate spinning reserve for large disturbances such as generator outages beyond regulation needs |
| 7 | N-1 security constraint (DC) | $|B_f^{(k)} \theta^{(k)}| \leq F^{\max}, \forall k \in \mathcal{C}$ | Ensure no line overload after any single line outage; implement preventive security-constrained dispatch |
| 8 | N-1 security constraint (AC) | $g^{(k)}(x) = 0, \quad h^{(k)}(x) \leq 0, \quad \forall k \in \mathcal{C}$ | Enforce voltage magnitude and thermal limits after any single contingency in full AC power flow model |

**Table 11 – continued from previous page**

| ID | Constraint Type | Formulation | Descriptions |
|----|-----------------|-------------|--------------|
| 9 | Voltage stability constraint | $L_i(V,\theta) \leq L^{\mathrm{max}}, \quad \forall i \in N_B$ | Prevent voltage instability by constraining voltage stability indices such as L-index below critical threshold |

## C. Data Synthesis Pseudocode

This appendix provides detailed pseudocode algorithms for the data synthesis process of each difficulty level in ProOPF-D. These algorithms formalize the generative procedures described in subsection 4.2.

### C.1. Level 1: Explicit Parameter Specification

---

**Algorithm 1** Data Synthesis for Level 1 ($\mathcal{L}_1$)

---

**Require:** Base system pool $\Omega_{\text{sys}}$, parameter modification space $\Omega_{\boldsymbol{\pi}}$, solver requirement space $\Omega_{\mathcal{R}}$, instruction specification $\boldsymbol{\tau}^{\mathcal{L}_1}$, LLM model $p_{\text{LLM}}$, number of samples $N$

**Ensure:** Sample set $\mathcal{Z} = \{z_i\}_{i=1}^N$ where each $z_i = \{\mathcal{M}_i^{\mathcal{L}_1}, \mathcal{P}_i, \mathcal{I}_i\}$

1: Initialize sample set: $\mathcal{Z} \leftarrow \emptyset$
2: **for** $i = 1$ to $N$ **do**
3:   Sample base system: $\omega \sim \mathcal{U}(\Omega_{\text{sys}})$
4:   Sample parameter modifications: $\Delta\boldsymbol{\pi} = \{\delta_k\}_{k=1}^K \sim \mathcal{U}(\Omega_{\boldsymbol{\pi}})$
5:   Set structural modification: $s \leftarrow \varnothing$
6:   Sample solver requirements: $\mathcal{R} \sim \mathcal{U}(\Omega_{\mathcal{R}})$
7:   Construct model specification: $\mathcal{M}^{\mathcal{L}_1} \leftarrow \{\omega, \Delta\boldsymbol{\pi}, s, \mathcal{R}\}$
8:   Generate NL description and implementation: $(\mathcal{P}, \mathcal{I}) \sim p_{\text{LLM}}(\cdot \mid \mathcal{M}^{\mathcal{L}_1}, \boldsymbol{\tau}^{\mathcal{L}_1})$
9:   Form complete sample: $z \leftarrow \{\mathcal{M}^{\mathcal{L}_1}, \mathcal{P}, \mathcal{I}\}$
10:   Add to sample set: $\mathcal{Z} \leftarrow \mathcal{Z} \cup \{z\}$
11: **end for**
12: **return** $\mathcal{Z}$

---

### C.2. Level 2: Semantic Parameter Inference

---

**Algorithm 2** Data Synthesis for Level 2 ($\mathcal{L}_2$)

---

**Require:** Base system pool $\Omega_{\text{sys}}$, directional parameter space $\Omega_{\boldsymbol{\pi}}^{\text{dir}}$, solver requirement space $\Omega_{\mathcal{R}}$, scenario tree collection $\mathcal{T} = \{T_m\}_{m=1}^M$, instruction specification $\boldsymbol{\tau}^{\mathcal{L}_2}$, LLM model $p_{\text{LLM}}$, number of samples $N$

**Ensure:** Sample set $\mathcal{Z} = \{z_i\}_{i=1}^N$ where each $z_i = \{\mathcal{M}_i^{\mathcal{L}_2}, \mathcal{P}_i, \mathcal{I}_i\}$

1: Initialize sample set: $\mathcal{Z} \leftarrow \emptyset$
2: **for** $i = 1$ to $N$ **do**
3:   Sample base system: $\omega \sim \mathcal{U}(\Omega_{\text{sys}})$
4:   Sample directional parameter modifications: $\Delta\boldsymbol{\pi} = \{\delta_k\}_{k=1}^K \sim \mathcal{U}(\Omega_{\boldsymbol{\pi}}^{\text{dir}})$
5:   Set structural modification: $s \leftarrow \varnothing$
6:   Sample solver requirements: $\mathcal{R} \sim \mathcal{U}(\Omega_{\mathcal{R}})$
7:   Initialize scenario fragment set: $\mathbf{c} \leftarrow \emptyset$
8:   **for** $k = 1$ to $K$ **do**
9:    Extract parameter identifier: $p_k \leftarrow \text{par}(\delta_k)$
10:    Extract modification direction: $d_k \leftarrow \text{dir}(\delta_k)$
11:    Retrieve matching leaf nodes: $\mathcal{V}_k \leftarrow \text{Retrieve}(\mathcal{T} \mid p_k, d_k)$
12:    Extract root-to-node paths: $\mathcal{N}_k \leftarrow \{\text{paths from root to } v \mid v \in \mathcal{V}_k\}$
13:    Sample scenario fragment: $c_k \sim \mathcal{U}(\mathcal{N}_k)$
14:    Add to fragment set: $\mathbf{c} \leftarrow \mathbf{c} \cup \{c_k\}$
15:   **end for**
16:   Construct intermediate specification: $\widetilde{\mathcal{M}}^{\mathcal{L}_2} \leftarrow \{\omega, \mathbf{c}, s, \mathcal{R}\}$
17:   Generate NL description and implementation: $(\mathcal{P}, \mathcal{I}) \sim p_{\text{LLM}}(\cdot \mid \widetilde{\mathcal{M}}^{\mathcal{L}_2}, \boldsymbol{\tau}^{\mathcal{L}_2})$
18:   Reconstruct original model specification: $\mathcal{M}^{\mathcal{L}_2} \leftarrow \{\omega, \Delta\boldsymbol{\pi}, s, \mathcal{R}\}$
19:   Form complete sample: $z \leftarrow \{\mathcal{M}^{\mathcal{L}_2}, \mathcal{P}, \mathcal{I}\}$
20:   Add to sample set: $\mathcal{Z} \leftarrow \mathcal{Z} \cup \{z\}$
21: **end for**
22: **return** $\mathcal{Z}$

---

## C.3. Level 3: Structural Extensions with Explicit Parameters

---

**Algorithm 3** Data Synthesis for Level 3 ($\mathcal{L}_3$)

---

**Require:** Base system pool $\Omega_{\text{sys}}$, parameter modification space $\Omega_{\boldsymbol{\pi}}$, structural modification space $\Omega_s = \mathcal{S}_p \times 2^{\mathcal{S}_o} \times 2^{\mathcal{S}_c}$, solver requirement space $\Omega_{\mathcal{R}}$, cardinality bounds $K_o, K_c$, instruction specification $\boldsymbol{\tau}^{\mathcal{L}_3}$, LLM model $p_{\text{LLM}}$, number of samples $N$

**Ensure:** Sample set $\mathcal{Z} = \{z_i\}_{i=1}^N$ where each $z_i = \{\mathcal{M}_i^{\mathcal{L}_3}, \mathcal{P}_i, \mathcal{I}_i\}$

 1: Initialize sample set: $\mathcal{Z} \leftarrow \emptyset$
 2: **for** $i = 1$ to $N$ **do**
 3:     Sample base system: $\omega \sim \mathcal{U}(\Omega_{\text{sys}})$
 4:     Sample parameter modifications: $\Delta\boldsymbol{\pi} = \{\delta_k\}_{k=1}^K \sim \mathcal{U}(\Omega_{\boldsymbol{\pi}})$
 5:     Sample problem type: $s_p \sim \mathcal{U}(\mathcal{S}_p)$
 6:     Sample objective extensions: $s_o \subseteq \mathcal{S}_o$ with $|s_o| \leq K_o$
 7:     Sample constraint extensions: $s_c \subseteq \mathcal{S}_c$ with $|s_c| \leq K_c$
 8:     Construct structural modification: $s \leftarrow (s_p, s_o, s_c)$
 9:     Sample solver requirements: $\mathcal{R} \sim \mathcal{U}(\Omega_{\mathcal{R}})$
10:     Construct model specification: $\mathcal{M}^{\mathcal{L}_3} \leftarrow \{\omega, \Delta\boldsymbol{\pi}, s, \mathcal{R}\}$
11:     Generate NL description and implementation: $(\mathcal{P}, \mathcal{I}) \sim p_{\text{LLM}}(\cdot \mid \mathcal{M}^{\mathcal{L}_3}, \boldsymbol{\tau}^{\mathcal{L}_3})$
12:     Form complete sample: $z \leftarrow \{\mathcal{M}^{\mathcal{L}_3}, \mathcal{P}, \mathcal{I}\}$
13:     Add to sample set: $\mathcal{Z} \leftarrow \mathcal{Z} \cup \{z\}$
14: **end for**
15: **return** $\mathcal{Z}$

---

## C.4. Level 4: Structural Extensions with Semantic Parameters

---

**Algorithm 4** Data Synthesis for Level 4 ($\mathcal{L}_4$)

---

**Require:** Base system pool $\Omega_{\text{sys}}$, directional parameter space $\Omega_{\boldsymbol{\pi}}^{\text{dir}}$, structural modification space $\Omega_s = \mathcal{S}_p \times 2^{\mathcal{S}_o} \times 2^{\mathcal{S}_c}$, solver requirement space $\Omega_{\mathcal{R}}$, relaxed cardinality bounds $K_o', K_c'$, scenario tree collection $\mathcal{T} = \{T_m\}_{m=1}^M$, instruction specification $\boldsymbol{\tau}^{\mathcal{L}_4}$, LLM model $p_{\text{LLM}}$, number of samples $N$

**Ensure:** Sample set $\mathcal{Z} = \{z_i\}_{i=1}^N$ where each $z_i = \{\mathcal{M}_i^{\mathcal{L}_4}, \mathcal{P}_i, \mathcal{I}_i\}$

 1: Initialize sample set: $\mathcal{Z} \leftarrow \emptyset$
 2: **for** $i = 1$ to $N$ **do**
 3:     Sample base system: $\omega \sim \mathcal{U}(\Omega_{\text{sys}})$
 4:     Sample directional parameter modifications: $\Delta\boldsymbol{\pi} = \{\delta_k\}_{k=1}^K \sim \mathcal{U}(\Omega_{\boldsymbol{\pi}}^{\text{dir}})$
 5:     Sample problem type: $s_p \sim \mathcal{U}(\mathcal{S}_p)$
 6:     Sample objective extensions: $s_o \subseteq \mathcal{S}_o$ with $|s_o| \leq K_o'$
 7:     Sample constraint extensions: $s_c \subseteq \mathcal{S}_c$ with $|s_c| \leq K_c'$
 8:     Construct structural modification: $s \leftarrow (s_p, s_o, s_c)$
 9:     Sample solver requirements: $\mathcal{R} \sim \mathcal{U}(\Omega_{\mathcal{R}})$
10:     Initialize scenario fragment set: $\mathbf{c} \leftarrow \emptyset$
11:     **for** $k = 1$ to $K$ **do**
12:         Extract parameter identifier: $p_k \leftarrow \text{par}(\delta_k)$
13:         Extract modification direction: $d_k \leftarrow \text{dir}(\delta_k)$
14:         Retrieve matching leaf nodes: $\mathcal{V}_k \leftarrow \text{Retrieve}(\mathcal{T} \mid p_k, d_k)$
15:         Extract root-to-node paths: $\mathcal{N}_k \leftarrow \{\text{paths from root to } v \mid v \in \mathcal{V}_k\}$
16:         Sample scenario fragment: $c_k \sim \mathcal{U}(\mathcal{N}_k)$
17:         Add to fragment set: $\mathbf{c} \leftarrow \mathbf{c} \cup \{c_k\}$
18:     **end for**
19:     Construct intermediate specification: $\widetilde{\mathcal{M}}^{\mathcal{L}_4} \leftarrow \{\omega, \mathbf{c}, s, \mathcal{R}\}$
20:     Generate NL description and implementation: $(\mathcal{P}, \mathcal{I}) \sim p_{\text{LLM}}(\cdot \mid \widetilde{\mathcal{M}}^{\mathcal{L}_4}, \boldsymbol{\tau}^{\mathcal{L}_4})$
21:     Reconstruct original model specification: $\mathcal{M}^{\mathcal{L}_4} \leftarrow \{\omega, \Delta\boldsymbol{\pi}, s, \mathcal{R}\}$
22:     Form complete sample: $z \leftarrow \{\mathcal{M}^{\mathcal{L}_4}, \mathcal{P}, \mathcal{I}\}$
23:     Add to sample set: $\mathcal{Z} \leftarrow \mathcal{Z} \cup \{z\}$
24: **end for**
25: **return** $\mathcal{Z}$

---

## C.5. Data Cleaning and Text Diversification

This subsection provides the pseudocode algorithm for the data cleaning and text diversification process described in Section 4.2.

---

**Algorithm 5** Data Cleaning and Text Diversification

---

**Require:** Raw sample set $\mathcal{Z}_{\text{raw}} = \{z_i\}_{i=1}^N$, compatibility rule table $\mathcal{C}$, diversification instruction $\boldsymbol{\tau}_{\text{diversify}}$, LLM model $p_{\text{LLM}}$
**Ensure:** Cleaned and diversified sample set $\mathcal{Z}_{\text{clean}}$
  1: {Stage 1: Compatibility filtering}
  2: Initialize filtered sample set: $\mathcal{Z}_{\text{filtered}} \leftarrow \emptyset$
  3: **for** $z_i = \{\mathcal{M}_i, \mathcal{P}_i, \mathcal{I}_i\} \in \mathcal{Z}_{\text{raw}}$ **do**
  4:     Extract structural modification: $s \leftarrow \mathcal{M}_i.s$
  5:     Extract parameter modifications: $\Delta\boldsymbol{\pi}_i \leftarrow \mathcal{M}_i.\Delta\boldsymbol{\pi}$
  6:     **if** $(s, \Delta\boldsymbol{\pi}_i) \notin \mathcal{C}$ **then**
  7:         $\mathcal{Z}_{\text{filtered}} \leftarrow \mathcal{Z}_{\text{filtered}} \cup \{z_i\}$
  8:     **end if**
  9: **end for**
10: {Stage 2: Text diversification for samples with identical $\mathcal{M}$}
11: Initialize cleaned set: $\mathcal{Z}_{\text{clean}} \leftarrow \emptyset$
12: **for** each unique model specification $\mathcal{M}$ in $\mathcal{Z}_{\text{filtered}}$ **do**
13:     Construct equivalence class: $\mathcal{Z}_{\mathcal{M}} \leftarrow \{z_i \in \mathcal{Z}_{\text{filtered}} \mid \mathcal{M}_i = \mathcal{M}\}$
14:     **if** $|\mathcal{Z}_{\mathcal{M}}| = 1$ **then**
15:         $\mathcal{Z}_{\text{clean}} \leftarrow \mathcal{Z}_{\text{clean}} \cup \mathcal{Z}_{\mathcal{M}}$
16:     **else**
17:         **for** $z_i = \{\mathcal{M}_i, \mathcal{P}_i, \mathcal{I}_i\} \in \mathcal{Z}_{\mathcal{M}}$ **do**
18:             Generate diversified text: $\mathcal{P}_i' \sim p_{\text{LLM}}(\cdot \mid \mathcal{P}_i, \boldsymbol{\tau}_{\text{diversify}})$
19:             Form diversified sample: $z_i' \leftarrow \{\mathcal{M}_i, \mathcal{P}_i', \mathcal{I}_i\}$
20:             $\mathcal{Z}_{\text{clean}} \leftarrow \mathcal{Z}_{\text{clean}} \cup \{z_i'\}$
21:         **end for**
22:     **end if**
23: **end for**
24: **return** $\mathcal{Z}_{\text{clean}}$

---

## D. Sample Examples and Generation Prompts by Difficulty Level for ProOPF-D

### D.1. Level 1: Explicit Parameter Synthesis

#### Part 1: Natural Language ($\mathcal{P}$)

The NL description provides explicit instructions with specific numerical values for parameter modifications.

> Perform AC optimal power flow (ACOPF) on the IEEE 39-bus system (case39). Scale the active power demand at bus 1 by a factor of 1.5. Set the maximum active power output of all generators at bus 32 to 500 MW. Relax the minimum voltage magnitude constraint at bus 10 by setting VMIN to 0. Set the branch reactance for the line between bus 2 and bus 25 to 0.01. Set the OPF violation tolerance (opf.violation) to 1e-8, and write the corresponding MATPOWER code.

#### Part 2: Model Specification ($\mathcal{M}$)

The structured specification defines parameter modifications with explicit operations and values.

```
{
  "base_system": "case39",
  "parameter_modifications": [
    {"component": "bus", "bus_id": 1, "target_parameter": "PD", "operation": "Scale",
        "value": 1.5},
    {"component": "gen", "bus_id": 32, "target_parameter": "PMAX", "operation": "Set"
        , "value": 500},
    {"component": "bus", "bus_id": 10, "target_parameter": "VMIN", "operation": "Set"
        , "value": 0},
    {"component": "branch", "fbus": 2, "tbus": 25, "target_parameter": "BR_X",
        "operation": "Set", "value": 0.01}
  ],
  "structural_modification": {},
  "solving_requirements": {
    "opf_violation": 1e-8
  }
}
```

#### Part 3: MATPOWER Code

The generated code implements the specified modifications and solves the OPF problem.

```
1   define_constants;
2   mpc = loadcase('case39');
3
4   bus1_idx = find(mpc.bus(:, BUS_I) == 1);
5   mpc.bus(bus1_idx, PD) = mpc.bus(bus1_idx, PD) * 1.5;
6
7   gen32_idx = find(mpc.gen(:, GEN_BUS) == 32);
8   mpc.gen(gen32_idx, PMAX) = 500;
9
10  bus10_idx = find(mpc.bus(:, BUS_I) == 10);
11  mpc.bus(bus10_idx, VMIN) = 0;
12
13  branch_idx = find( ...
14      (mpc.branch(:, F_BUS) == 2  & mpc.branch(:, T_BUS) == 25) | ...
15      (mpc.branch(:, F_BUS) == 25 & mpc.branch(:, T_BUS) == 2) );
16  mpc.branch(branch_idx, BR_X) = 0.01;
17
18  mpopt = mpoption('opf.violation', 1e-8);
19  results = runopf(mpc, mpopt);
20  printpf(results);
```

#### Part 4: Synthesis Instructions ($\tau^{\mathcal{L}_1}$)

The synthesis instruction $\tau^{\mathcal{L}_1}$ guides the LLM to generate aligned natural language descriptions $\mathcal{P}$ and executable MATPOWER implementations $\mathcal{I}$ from a given model specification $\mathcal{M}^{\mathcal{L}_1}$. The prompt structure follows a few-shot in-context learning paradigm with explicit formatting requirements.

**Task:** You are an expert power systems engineer specializing in optimal power flow (OPF) modeling. Your task is to generate a natural language description and executable MATPOWER code for an OPF problem based on the provided model specification. The natural language description should be clear and professional, suitable for control room operators, while the MATPOWER code must be syntactically correct and executable.

**Input Format:** A JSON object containing base_system, parameter_modifications (each with component, bus_id or fbus/tbus, target_parameter, operation, and value), structural_modification (empty for Level 1), and solving_requirements.

**Output Requirements:**

1. **Natural Language Description:** Write a clear, professional instruction that explicitly states all parameter modifications with their numerical values. Use imperative language (e.g., "Scale...", "Set...") and include the base system, all modifications, and solver requirements.

2. **MATPOWER Code:** Generate executable MATLAB code that loads the base system using `loadcase()`, applies all parameter modifications using appropriate MATPOWER indexing, configures solver options via `mpoption()`, executes OPF via `runopf()`.

**Example:**
**Input Model Specification:**

```
{
   "base_system": "case39",
   "parameter_modifications": [
    {"component": "bus", "bus_id": 1, "target_parameter": "PD",
     "operation": "Scale", "value": 1.5},
    {"component": "gen", "bus_id": 32, "target_parameter": "PMAX",
     "operation": "Set", "value": 500},
    {"component": "bus", "bus_id": 10, "target_parameter": "VMIN",
     "operation": "Set", "value": 0},
    {"component": "branch", "fbus": 2, "tbus": 25,
     "target_parameter": "BR_X", "operation": "Set", "value": 0.01}
   ],
   "structural_modification": {},
   "solving_requirements": {"opf_violation": 1e-8}
}
```

**Expected Natural Language Output:** Perform AC optimal power flow (ACOPF) on the IEEE 39-bus system (case39). Scale the active power demand at bus 1 by a factor of 1.5. Set the maximum active power output of all generators at bus 32 to 500 MW. Relax the minimum voltage magnitude constraint at bus 10 by setting VMIN to 0. Set the branch reactance (BR_X) for the line between bus 2 and bus 25 to 0.01. Set the OPF violation tolerance (opf.violation) to 1e-8, and write the corresponding MATPOWER code.

**Expected MATPOWER Code Output:**

```matlab
define_constants;
mpc = loadcase('case39');

bus1_idx = find(mpc.bus(:, BUS_I) == 1);
mpc.bus(bus1_idx, PD) = mpc.bus(bus1_idx, PD) * 1.5;

gen32_idx = find(mpc.gen(:, GEN_BUS) == 32);
mpc.gen(gen32_idx, PMAX) = 500;

bus10_idx = find(mpc.bus(:, BUS_I) == 10);
mpc.bus(bus10_idx, VMIN) = 0;

branch_idx = find( ...
    (mpc.branch(:, F_BUS) == 2  & mpc.branch(:, T_BUS) == 25) | ...
    (mpc.branch(:, F_BUS) == 25 & mpc.branch(:, T_BUS) == 2) );
mpc.branch(branch_idx, BR_X) = 0.01;

mpopt = mpoption('opf.violation', 1e-8);
results = runopf(mpc, mpopt);
```

## D.2. Level 2: Semantic Parameter Inference

### Part 1: Natural Language ($\mathcal{P}$)

The NL description uses scenario-based narratives to describe parameter changes, requiring inference of modification directions rather than explicit values.

A regional grid is modeled using the IEEE 39-bus system (case39). During an extreme summer heatwave, widespread air-conditioning usage sharply increases the electrical demand around bus 1. Meanwhile, high ambient temperature forces the generator(s) connected to bus 32 to operate in a derated mode, limiting their deliverable output. During late-night hours, bus 10, which primarily serves industrial loads from an industrial park, enters a scheduled shutdown period with all production facilities ceasing operations. In addition, a fault-triggered outage of local compensation equipment on the corridor between bus 2 and bus 25 degrades power transfer capability along that path. Set opf.violation to 1e-8, use IPOPT as the ACOPF solver, and generate the corresponding MATPOWER code. All scenario-driven parameters are assigned using placeholder values of the form object_parameter_id, which represent the post-modification values consistent with the specified scenario directions.

### Part 2: Model Specification ($\mathcal{M}$)

The structured specification uses direction constraints (Increase/Decrease) instead of explicit values, requiring semantic understanding of the scenario.

```
{
  "base_system": "case39",
  "parameter_modifications":  [
    {"component": "bus", "bus_id": 1, "target_parameter": "PD", "direction": "
        Increase"},
    {"component": "gen", "bus_id": 32, "target_parameter": "PMAX", "direction": "
        Decrease"},
    {"component": "bus", "bus_id": 10, "target_parameter": "PD", "direction": "Set
        zero"},
    {"component": "branch", "fbus": 2, "tbus": 25, "target_parameter": "BR_X",
        "direction": "Increase"}
  ],
  "structural_modification":  {},
  "solving_requirements": {
    "opf_violation": 1e-8,
    "solver": "IPOPT"
  }
}
```

### Part 3: MATPOWER Code

The generated code is structured as a function that accepts placeholder variables as parameters and includes assertion checks to validate direction constraints.

```
1   function results = opf_case39(bus_PD_1, gen_PMAX_32, bus_PD_10, branch_BR_X_2_25)
2       define_constants;
3       mpc = loadcase('case39');
4
5       bus1_idx = find(mpc.bus(:, BUS_I) == 1);
6       pd0_bus1 = mpc.bus(bus1_idx, PD);
7       assert(bus_PD_1 >= pd0_bus1);
8       mpc.bus(bus1_idx, PD) = bus_PD_1;
9
10      gen32_idx = find(mpc.gen(:, GEN_BUS) == 32);
11      pmax0_gen32 = mpc.gen(gen32_idx, PMAX);
12      assert(all(gen_PMAX_32 <= pmax0_gen32));
13      mpc.gen(gen32_idx, PMAX) = gen_PMAX_32;
14
15      bus10_idx = find(mpc.bus(:, BUS_I) == 10);
16      assert(bus_PD_10 == 0);
17      mpc.bus(bus10_idx, PD) = bus_PD_10;
18
19      branch_idx = find( ...
20          (mpc.branch(:, F_BUS) == 2  & mpc.branch(:, T_BUS) == 25) | ...
21          (mpc.branch(:, F_BUS) == 25 & mpc.branch(:, T_BUS) == 2) );
```

```
22        x0_2_25 = mpc.branch(branch_idx, BR_X);
23        assert(all(branch_BR_X_2_25 >= x0_2_25));
24        mpc.branch(branch_idx, BR_X) = branch_BR_X_2_25;
25
26        mpopt = mpoption('opf.violation', 1e-8, 'opf.ac.solver', 'IPOPT');
27        results = runopf(mpc, mpopt);
28        printpf(results);
29    end
```

## Part 4: Synthesis Instructions ($\tau^{\mathcal{L}_2}$)

The synthesis instruction $\tau^{\mathcal{L}_2}$ guides the LLM to generate scenario-based natural language descriptions $\mathcal{P}$ and parameterized MATPOWER implementations $\mathcal{I}$ from scenario fragments $\{c_k\}_{k=1}^K$ that semantically encode parameter modification directions without explicit numerical values. The prompt structure emphasizes narrative composition and function design with placeholder-based parameterization.

**Task:** You are an expert power systems engineer specializing in optimal power flow (OPF) modeling. Your task is to generate a scenario-based natural language description and a parameterized MATPOWER function for an OPF problem based on the provided scenario fragments. The natural language description should compose these fragments into a coherent operational narrative without explicitly mentioning parameter names or modification directions. The MATPOWER code must be structured as a function that accepts placeholder variables and includes assertion checks to validate modification direction constraints.

**Input Format:** A JSON object containing base_system, scenario_fragments (a list of scenario descriptions that semantically encode parameter modification directions), structural_modification (empty for Level 2), and solving_requirements.

**Output Requirements:**

1. **Natural Language Description:** Compose the scenario fragments into a coherent, professional narrative that describes operational conditions and their implications on the power system. The description should:

   - Integrate all scenario fragments into a unified operational scenario
   - Use natural, scenario-based language (e.g., "During an extreme summer heatwave...", "Meanwhile, high ambient temperature forces...")
   - Avoid explicitly stating parameter names or modification directions
   - Include a statement that scenario-driven parameters use placeholder values of the form object_parameter_id
   - Include solver requirements and request MATPOWER code generation

2. **MATPOWER Code:** Generate a MATLAB function that:

   - Accepts placeholder variables as function parameters (naming convention: object_parameter_id, e.g., bus_PD_1, gen_PMAX_32)
   - Loads the base system using `loadcase()`
   - For each parameter modification, retrieves the original value, adds an assertion to validate the modification direction (e.g., `assert(bus_PD_1 >= pd0_bus1)` for Increase, `assert(all(gen_PMAX_32 <= pmax0_gen32))` for Decrease, `assert(bus_PD_10 == 0)` for Set zero)
   - Applies the placeholder parameter values to the model
   - Configures solver options via `mpoption()` including solver specification if provided
   - Executes OPF via `runopf()`

**Example:**
**Input Scenario Fragments:**

```
{
  "base_system": "case39",
  "scenario_fragments": [
    "Extreme summer heatwave + Increased electrical demand",
    "High ambient temperature + Generator derating",
    "Late-night hours + Industrial park shutdown",
    "Fault-triggered outage + Degraded power transfer capability"
  ],
  "structural_modification": {},
  "solving_requirements": {
    "opf_violation": 1e-8,
    "solver": "IPOPT"
  }
}
```

**Expected Natural Language Output:** A regional grid is modeled using the IEEE 39-bus system (case39). During an extreme summer heatwave, widespread air-conditioning usage sharply increases the electrical demand around bus 1. Meanwhile, high ambient temperature forces the generator(s) connected to bus 32 to operate in a derated mode, limiting their deliverable output. During late-night hours, bus 10, which primarily serves industrial loads from an industrial park, enters a scheduled shutdown period with all production facilities ceasing operations. In addition, a fault-triggered outage of local compensation equipment on the corridor between bus 2 and bus 25 degrades power transfer capability along that path. Set opf.violation to 1e-8, use IPOPT as the ACOPF solver, and generate the corresponding MATPOWER code. All scenario-driven parameters are assigned using placeholder values of the form object_parameter_id, which represent the post-modification values consistent with the specified scenario directions.

**Expected MATPOWER Code Output:**

```
1   function results = opf_case39(bus_PD_1, gen_PMAX_32, bus_PD_10, branch_BR_X_2_25)
2       define_constants;
3       mpc = loadcase('case39');
4
5       bus1_idx = find(mpc.bus(:, BUS_I) == 1);
6       pd0_bus1 = mpc.bus(bus1_idx, PD);
7       assert(bus_PD_1 >= pd0_bus1);
8       mpc.bus(bus1_idx, PD) = bus_PD_1;
9
10      gen32_idx = find(mpc.gen(:, GEN_BUS) == 32);
11      pmax0_gen32 = mpc.gen(gen32_idx, PMAX);
12      assert(all(gen_PMAX_32 <= pmax0_gen32));
13      mpc.gen(gen32_idx, PMAX) = gen_PMAX_32;
14
15      bus10_idx = find(mpc.bus(:, BUS_I) == 10);
16      assert(bus_PD_10 == 0);
17      mpc.bus(bus10_idx, PD) = bus_PD_10;
18
19      branch_idx = find( ...
20          (mpc.branch(:, F_BUS) == 2  & mpc.branch(:, T_BUS) == 25) | ...
21          (mpc.branch(:, F_BUS) == 25 & mpc.branch(:, T_BUS) == 2) );
22      x0_2_25 = mpc.branch(branch_idx, BR_X);
23      assert(all(branch_BR_X_2_25 >= x0_2_25));
24      mpc.branch(branch_idx, BR_X) = branch_BR_X_2_25;
25
26      mpopt = mpoption('opf.violation', 1e-8, 'opf.ac.solver', 'IPOPT');
27      results = runopf(mpc, mpopt);
28      printf(results);
29  end
```

## D.3. Level 3: Structural Extensions with Explicit Parameters

### Part 1: Natural Language ($\mathcal{P}$)

The NL description includes structural modifications (problem type change and custom objective function) along with explicit parameter specifications.

Formulate a DC optimal power flow (DCOPF) problem for the IEEE 39-bus system (case39). In addition to the default generation cost in the base case, add a quadratic penalty on phase-angle differences across all in-service transmission lines to discourage excessive angle separation (penalty weight beta = 10). Scale the active power demand at bus 1 by 1.5, set the maximum active power output of the generator(s) connected to bus 32 to 500 MW, and set the branch reactance of the line between bus 2 and bus 25 to 0.01. Set opf.violation to 1e-8, and write the corresponding MATPOWER code.

### Part 2: Model Specification ($\mathcal{M}$)

The structured specification includes both explicit parameter modifications and structural modifications (problem type and objective function extension).

```
    {
      "base_system": "case39",
      "parameter_modifications": [
        {"component": "bus", "bus_id": 1, "target_parameter": "PD", "operation": "Scale",
            "value": 1.5},
```

```
                {"component": "gen", "bus_id": 32, "target_parameter": "PMAX", "operation": "Set"
                    , "value": 500},
                {"component": "branch", "fbus": 2, "tbus": 25, "target_parameter": "BR_X",
                    "operation": "Set", "value": 0.01}
            ],
            "structural_modification": {
              "problem": "DCOPF",
              "objective_modification": {
                "op": "add",
                "name": "angle_difference_penalty",
                "beta": 10,
                "form": "beta * ||E*Va||_2^2"
              }
            },
            "solving_requirements": {
              "opf_violation": 1e-8
            }
          }
```

## Part 3: MATPOWER Code

The generated code implements structural modifications using advanced OPF setup with custom objective function.

```matlab
1    define_constants;
2    mpc = loadcase('case39');
3
4    bus1_idx = find(mpc.bus(:, BUS_I) == 1);
5    mpc.bus(bus1_idx, PD) = mpc.bus(bus1_idx, PD) * 1.5;
6
7    gen32_idx = find(mpc.gen(:, GEN_BUS) == 32);
8    mpc.gen(gen32_idx, PMAX) = 500;
9
10   branch_idx_2_25 = find( ...
11       (mpc.branch(:, F_BUS) == 2  & mpc.branch(:, T_BUS) == 25) | ...
12       (mpc.branch(:, F_BUS) == 25 & mpc.branch(:, T_BUS) == 2) );
13   mpc.branch(branch_idx_2_25, BR_X) = 0.01;
14
15   mpopt = mpoption('opf.violation', 1e-8, 'model', 'DC');
16
17   beta = 10;
18
19   om = opf_setup(mpc, mpopt);
20
21   nb = size(mpc.bus, 1);
22   br_on = find(mpc.branch(:, BR_STATUS) == 1);
23   nl = length(br_on);
24
25   f = mpc.branch(br_on, F_BUS);
26   t = mpc.branch(br_on, T_BUS);
27
28   E = sparse(1:nl, f,  1, nl, nb) + sparse(1:nl, t, -1, nl, nb);
29
30   Q_va = 2 * beta * (E' * E);
31   om = om.add_quad_cost('ang_diff_pen', Q_va, [], 0, {'Va'});
32
33   idx_i1 = om.var.idx.i1;
34   idx_iN = om.var.idx.iN;
35
36   [x, fval, eflag, ~, ~] = om.solve();
37
38   Va_solution = x(idx_i1.Va:idx_iN.Va);
39   Pg_solution = x(idx_i1.Pg:idx_iN.Pg);
40
41   results = mpc;
42   results.bus(:, VA) = Va_solution * 180/pi;
```

```
43    results.gen(:, PG) = Pg_solution;
44    results.f = fval;
45    results.success = (eflag > 0);
46
47    printpf(results);
```

**Part 4: Synthesis Instructions ($\tau^{\mathcal{L}_3}$)**

The synthesis instruction $\tau^{\mathcal{L}_3}$ guides the LLM to generate natural language descriptions $\mathcal{P}$ and executable MATPOWER implementations $\mathcal{I}$ that incorporate structural modifications (e.g., problem type changes, custom objective functions, additional constraints) along with explicit parameter modifications. The prompt structure emphasizes understanding structural extension design principles and their MATPOWER implementation patterns.

---

**Task:** You are an expert power systems engineer specializing in optimal power flow (OPF) modeling with advanced structural extensions. Your task is to generate a natural language description and executable MATPOWER code for an OPF problem that includes both structural modifications (problem type changes, objective function extensions, or constraint additions) and explicit parameter modifications with numerical values.

**Input Format:** A JSON object containing the following components:

- **base_system:** The base power system specification (e.g., MATPOWER case file identifier or system data structure).

- **parameter_modifications:** Explicit parameter modifications with specific operations (e.g., set, scale, offset) and numerical values applied to system parameters (e.g., generator costs, line impedances, load values).

- **structural_modification:** A specification of structural changes to the OPF problem formulation, which may include:

  - **Problem Type Changes:** Modifications to the fundamental problem formulation that alter the decision variable space or reformulate the problem structure, such as switching between different OPF variants (e.g., `"ACOPF"`, `"DCOPF"`, `"ED"`, `"UC"`, `"OTS"`).
  - **Objective Function Extensions:** Additions or modifications to the objective function beyond the standard cost minimization, such as quadratic penalties on specific decision variables, linear terms, or custom cost functions that incorporate additional system considerations.
  - **Constraint Modifications:** Additions, modifications, or relaxations of constraints beyond the standard OPF constraints, including custom equality or inequality constraints that capture additional operational requirements or system characteristics.

  The structural_modification component should specify the type of structural change, the mathematical form of the modification (e.g., quadratic penalty terms, linear constraints), the decision variables involved, and any relevant parameters or coefficients.

- **solving_requirements:** Solver configuration and solution requirements (e.g., solver type, tolerance settings, output format specifications).

**Structural Modification Design Principles:**

- **Problem Type Changes:** Structural modifications that alter the fundamental mathematical model (e.g., linearization assumptions, variable elimination, constraint relaxation) require corresponding changes in the optimization framework configuration and variable space definition.

- **Objective Function Extensions:** Custom objective terms should be mathematically well-defined and compatible with the underlying optimization framework, maintaining convexity properties when required and ensuring proper integration with existing cost functions.

- **Constraint Extensions:** Additional constraints must be formulated in a manner consistent with the optimization model structure, properly indexing decision variables and maintaining feasibility of the solution space.

- **Implementation Considerations:** The implementation should follow the optimization framework's standard patterns for model construction, variable definition, constraint addition, and solution extraction, ensuring compatibility with the framework's internal mechanisms.

**Output Requirements:**

1. **Natural Language Description:** Write a clear, professional instruction that:

   - Explicitly states the structural modification (e.g., "Formulate a DC optimal power flow...", "add a quadratic penalty on...")
   - Includes all explicit parameter modifications with their numerical values
   - Specifies solver requirements and requests MATPOWER code generation

---

2. **MATPOWER Code:** Generate executable MATLAB code that:

- Loads the base system and applies explicit parameter modifications
- Configures solver options including model type if structural modification specifies a problem type change
- Uses `opf_setup()` to create an optimization model object
- Implements structural modifications:
  - For objective extensions: Construct the quadratic cost matrix $Q$ based on the specified form, then use `om.add_quad_cost()` to add the term
  - For constraint extensions: Use appropriate constraint addition methods
- Solves using `om.solve()` and extracts solution components
- Constructs and returns a results structure with solution values

**Example:**
**Input Model Specification:**

```
{
  "base_system": "case39",
  "parameter_modifications": [
    {"component": "bus", "bus_id": 1, "target_parameter": "PD",
     "operation": "Scale", "value": 1.5},
    {"component": "gen", "bus_id": 32, "target_parameter": "PMAX",
     "operation": "Set", "value": 500},
    {"component": "branch", "fbus": 2, "tbus": 25,
     "target_parameter": "BR_X", "operation": "Set", "value": 0.01}
  ],
  "structural_modification": {
    "problem": "DCOPF",
    "objective_modification": {
      "op": "add",
      "name": "angle_difference_penalty",
      "beta": 10,
      "form": "beta * ||E*Va||_2^2"
    }
  },
  "solving_requirements": {"opf_violation": 1e-8}
}
```

**Expected Natural Language Output:** Formulate a DC optimal power flow (DCOPF) problem for the IEEE 39-bus system (case39). In addition to the default generation cost in the base case, add a quadratic penalty on phase-angle differences across all in-service transmission lines to discourage excessive angle separation (penalty weight beta = 10). Scale the active power demand at bus 1 by 1.5, set the maximum active power output of the generator(s) connected to bus 32 to 500 MW, and set the branch reactance of the line between bus 2 and bus 25 to 0.01. Set opf.violation to 1e-8, and write the corresponding MATPOWER code.

**Expected MATPOWER Code Output:**

```
1    define_constants;
2    mpc = loadcase('case39');
3
4    bus1_idx = find(mpc.bus(:, BUS_I) == 1);
5    mpc.bus(bus1_idx, PD) = mpc.bus(bus1_idx, PD) * 1.5;
6
7    gen32_idx = find(mpc.gen(:, GEN_BUS) == 32);
8    mpc.gen(gen32_idx, PMAX) = 500;
9
10   branch_idx_2_25 = find( ...
11       (mpc.branch(:, F_BUS) == 2  & mpc.branch(:, T_BUS) == 25) | ...
12       (mpc.branch(:, F_BUS) == 25 & mpc.branch(:, T_BUS) == 2) );
13   mpc.branch(branch_idx_2_25, BR_X) = 0.01;
14
15   mpopt = mpoption('opf.violation', 1e-8, 'model', 'DC');
16
17   beta = 10;
18
19   om = opf_setup(mpc, mpopt);
```

```
20
21    nb = size(mpc.bus, 1);
22    br_on = find(mpc.branch(:, BR_STATUS) == 1);
23    nl = length(br_on);
24
25    f = mpc.branch(br_on, F_BUS);
26    t = mpc.branch(br_on, T_BUS);
27
28    E = sparse(1:nl, f,  1, nl, nb) + sparse(1:nl, t, -1, nl, nb);
29
30    Q_va = 2 * beta * (E' * E);
31    om = om.add_quad_cost('ang_diff_pen', Q_va, [], 0, {'Va'});
32
33    idx_i1 = om.var.idx.i1;
34    idx_iN = om.var.idx.iN;
35
36    [x, fval, eflag, ~, ~] = om.solve();
37
38    Va_solution = x(idx_i1.Va:idx_iN.Va);
39    Pg_solution = x(idx_i1.Pg:idx_iN.Pg);
40
41    results = mpc;
42    results.bus(:, VA) = Va_solution * 180/pi;
43    results.gen(:, PG) = Pg_solution;
44    results.f = fval;
45    results.success = (eflag > 0);
46
47    printpf(results);
```

## D.4. Level 4: Structural Extensions with Semantic Parameters

### Part 1: Natural Language ($\mathcal{P}$)

The NL description combines scenario-based narratives with structural modification requirements, representing the most complex level.

A regional grid is modeled using the IEEE 39-bus system (case39). During an extreme summer heatwave, widespread air-conditioning usage sharply increases the demand around bus 1. High ambient temperature forces the generator(s) connected to bus 32 to operate in a derated mode, reducing their deliverable output capacity. In addition, a fault-triggered outage of local compensation equipment on the corridor between bus 2 and bus 25 degrades power transfer capability along that path. To discourage excessive phase separation under stressed operation, formulate a DC optimal power flow (DCOPF) and add a quadratic penalty on phase-angle differences across all in-service transmission lines with penalty weight beta = 10. Set opf.violation to 1e-8, use IPOPT as the DCOPF solver, and generate the corresponding MATPOWER code. All scenario-driven parameters are assigned using placeholder values of the form object_parameter_id, which represent post-modification values consistent with the specified scenario directions.

### Part 2: Model Specification ($\mathcal{M}$)

The structured specification combines semantic parameter specifications (direction constraints) with structural modifications (problem type and objective function extension).

```
    {
      "base_system": "case39",
      "parameter_modifications": [
        {"component": "bus", "bus_id": 1, "target_parameter": "PD", "direction": "
            Increase"},
        {"component": "gen", "bus_id": 32, "target_parameter": "PMAX", "direction": "
            Decrease"},
        {"component": "branch", "fbus": 2, "tbus": 25, "target_parameter": "BR_X",
            "direction": "Increase"}
      ],
       "structural_modification": {
        "problem": "DCOPF",
        "objective_modification": {
```

```
        "op": "add",
        "name": "angle_difference_penalty",
        "beta": 10,
        "form": "beta * ||E*Va||_2^2"
      }
    },
    "solving_requirements": {
      "opf_violation": 1e-8,
      "solver": "IPOPT"
    }
  }
```

**Part 3: MATPOWER Code**

The generated code is structured as a function that accepts placeholder variables as parameters, combines them with assertion checks, and implements advanced OPF setup for structural modifications.

```matlab
1   function results = opf_case39_sc(bus_PD_1, gen_PMAX_32, branch_BR_X_2_25)
2       define_constants;
3       mpc = loadcase('case39');
4
5       bus1_idx = find(mpc.bus(:, BUS_I) == 1);
6       pd0_bus1 = mpc.bus(bus1_idx, PD);
7       assert(bus_PD_1 >= pd0_bus1);
8       mpc.bus(bus1_idx, PD) = bus_PD_1;
9
10      gen32_idx = find(mpc.gen(:, GEN_BUS) == 32);
11      pmax0_gen32 = mpc.gen(gen32_idx, PMAX);
12      assert(all(gen_PMAX_32 <= pmax0_gen32));
13      mpc.gen(gen32_idx, PMAX) = gen_PMAX_32;
14
15      branch_idx_2_25 = find( ...
16          (mpc.branch(:, F_BUS) == 2  & mpc.branch(:, T_BUS) == 25) | ...
17          (mpc.branch(:, F_BUS) == 25 & mpc.branch(:, T_BUS) == 2) );
18      x0_2_25 = mpc.branch(branch_idx_2_25, BR_X);
19      assert(all(branch_BR_X_2_25 >= x0_2_25));
20      mpc.branch(branch_idx_2_25, BR_X) = branch_BR_X_2_25;
21
22      mpopt = mpoption('opf.violation', 1e-8, 'model', 'DC', 'opf.dc.solver', 'IPOPT');
23
24      beta = 10;
25
26      om = opf_setup(mpc, mpopt);
27
28      nb = size(mpc.bus, 1);
29      br_on = find(mpc.branch(:, BR_STATUS) == 1);
30      nl = length(br_on);
31
32      f = mpc.branch(br_on, F_BUS);
33      t = mpc.branch(br_on, T_BUS);
34
35      E = sparse(1:nl, f,  1, nl, nb) + sparse(1:nl, t, -1, nl, nb);
36
37      Q_va = 2 * beta * (E' * E);
38      om = om.add_quad_cost('ang_diff_pen', Q_va, [], 0, {'Va'});
39
40      idx_i1 = om.var.idx.i1;
41      idx_iN = om.var.idx.iN;
42
43      [x, fval, eflag, ~, ~] = om.solve();
44
45      Va_solution = x(idx_i1.Va:idx_iN.Va);
46      Pg_solution = x(idx_i1.Pg:idx_iN.Pg);
47
48      results = mpc;
```

```
49        results.bus(:, VA) = Va_solution * 180/pi;
50        results.gen(:, PG) = Pg_solution;
51        results.f = fval;
52        results.success = (eflag > 0);
53
54        printf(results);
55    end
```

## Part 4: Synthesis Instructions ($\tau^{\mathcal{L}_4}$)

The synthesis instruction $\tau^{\mathcal{L}_4}$ guides the LLM to generate scenario-based natural language descriptions $\mathcal{P}$ and parameterized MATPOWER implementations $\mathcal{I}$ that combine structural modifications with semantic parameter inference. This represents the most complex level, requiring both narrative composition from scenario fragments and advanced structural extension implementation with placeholder-based parameterization.

---

**Task:** You are an expert power systems engineer specializing in optimal power flow (OPF) modeling with advanced structural extensions and scenario-based parameter inference. Your task is to generate a scenario-based natural language description and a parameterized MATPOWER function for an OPF problem that includes both structural modifications and parameter modifications inferred from operational scenarios (without explicit numerical values).

**Input Format:** A JSON object containing the following components:

- **base_system:** The base power system specification (e.g., MATPOWER case file identifier or system data structure).

- **scenario_fragments:** A list of scenario descriptions that encode parameter modification directions without explicit numerical values. Each fragment describes an operational condition or event (e.g., weather conditions, equipment failures, demand changes) that implies specific parameter modifications (e.g., increased load, reduced generator capacity, degraded line parameters).

- **structural_modification:** A specification of structural changes to the OPF problem formulation, which may include:

  - **Problem Type Changes:** Modifications to the fundamental problem formulation that alter the decision variable space or reformulate the problem structure, such as switching between different OPF variants (e.g., `"ACOPF"`, `"DCOPF"`, `"ED"`, `"UC"`, `"OTS"`).
  - **Objective Function Extensions:** Additions or modifications to the objective function beyond the standard cost minimization, such as quadratic penalties on specific decision variables, linear terms, or custom cost functions that incorporate additional system considerations.
  - **Constraint Modifications:** Additions, modifications, or relaxations of constraints beyond the standard OPF constraints, including custom equality or inequality constraints that capture additional operational requirements or system characteristics.

  The structural_modification component should specify the type of structural change, the mathematical form of the modification (e.g., quadratic penalty terms, linear constraints), the decision variables involved, and any relevant parameters or coefficients.

- **solving_requirements:** Solver configuration and solution requirements (e.g., solver type, tolerance settings, output format specifications).

**Structural Modification Design Principles:**

- **Problem Type Changes:** Structural modifications that alter the fundamental mathematical model (e.g., linearization assumptions, variable elimination, constraint relaxation) require corresponding changes in the optimization framework configuration and variable space definition.

- **Objective Function Extensions:** Custom objective terms should be mathematically well-defined and compatible with the underlying optimization framework, maintaining convexity properties when required and ensuring proper integration with existing cost functions.

- **Constraint Extensions:** Additional constraints must be formulated in a manner consistent with the optimization model structure, properly indexing decision variables and maintaining feasibility of the solution space.

- **Implementation Considerations:** The implementation should follow the optimization framework's standard patterns for model construction, variable definition, constraint addition, and solution extraction, ensuring compatibility with the framework's internal mechanisms. Parameter modifications inferred from scenarios should be implemented using placeholder variables that can be instantiated with specific values during function invocation.

**Output Requirements:**

---

1. **Natural Language Description:** Compose the scenario fragments into a coherent, professional narrative that:
    - Integrates all scenario fragments into a unified operational scenario
    - Uses natural, scenario-based language without explicitly stating parameter names or modification directions
    - Explicitly describes the structural modification (e.g., "formulate a DC optimal power flow...", "add a quadratic penalty on...")
    - Includes a statement that scenario-driven parameters use placeholder values of the form object_parameter_id
    - Specifies solver requirements and requests MATPOWER code generation

2. **MATPOWER Code:** Generate a MATLAB function that:
    - Accepts placeholder variables as function parameters (naming convention: object_parameter_id)
    - Loads the base system
    - For each parameter modification, retrieves the original value, adds an assertion to validate the modification direction, and applies the placeholder parameter value
    - Configures solver options including model type if structural modification specifies a problem type change
    - Uses `opf_setup()` to create an optimization model object
    - Implements structural modifications using `om.add_quad_cost()` or other appropriate methods
    - Solves using `om.solve()`, extracts solution components, and constructs the results structure

**Example:**
**Input Scenario Fragments and Structural Modification:**

```
{
   "base_system": "case39",
   "scenario_fragments": [
     "Extreme summer heatwave + Increased electrical demand",
     "High ambient temperature + Generator derating",
     "Fault-triggered outage + Degraded power transfer capability"
   ],
    "structural_modification": {
     "problem": "DCOPF",
     "objective_modification": {
       "op": "add",
       "name": "angle_difference_penalty",
       "beta": 10,
       "form": "beta * ||E*Va||_2^2"
     }
   },
   "solving_requirements": {
     "opf_violation": 1e-8,
     "solver": "IPOPT"
   }
}
```

**Expected Natural Language Output:** A regional grid is modeled using the IEEE 39-bus system (case39). During an extreme summer heatwave, widespread air-conditioning usage sharply increases the demand around bus 1. High ambient temperature forces the generator(s) connected to bus 32 to operate in a derated mode, reducing their deliverable output capacity. In addition, a fault-triggered outage of local compensation equipment on the corridor between bus 2 and bus 25 degrades power transfer capability along that path. To discourage excessive phase separation under stressed operation, formulate a DC optimal power flow (DCOPF) and add a quadratic penalty on phase-angle differences across all in-service transmission lines with penalty weight beta = 10. Set opf.violation to 1e-8, use IPOPT as the DCOPF solver, and generate the corresponding MATPOWER code. All scenario-driven parameters are assigned using placeholder values of the form object_parameter_id, which represent post-modification values consistent with the specified scenario directions.
**Expected MATPOWER Code Output:**

```
1  function results = opf_case39_sc(bus_PD_1, gen_PMAX_32, branch_BR_X_2_25)
2      define_constants;
3      mpc = loadcase('case39');
4
5      bus1_idx = find(mpc.bus(:, BUS_I) == 1);
6      pd0_bus1 = mpc.bus(bus1_idx, PD);
7      assert(bus_PD_1 >= pd0_bus1);
```

```
8        mpc.bus(bus1_idx, PD) = bus_PD_1;

10       gen32_idx = find(mpc.gen(:, GEN_BUS) == 32);
11       pmax0_gen32 = mpc.gen(gen32_idx, PMAX);
12       assert(all(gen_PMAX_32 <= pmax0_gen32));
13       mpc.gen(gen32_idx, PMAX) = gen_PMAX_32;

15       branch_idx_2_25 = find( ...
16           (mpc.branch(:, F_BUS) == 2  & mpc.branch(:, T_BUS) == 25) | ...
17           (mpc.branch(:, F_BUS) == 25 & mpc.branch(:, T_BUS) == 2) );
18       x0_2_25 = mpc.branch(branch_idx_2_25, BR_X);
19       assert(all(branch_BR_X_2_25 >= x0_2_25));
20       mpc.branch(branch_idx_2_25, BR_X) = branch_BR_X_2_25;

22       mpopt = mpoption('opf.violation', 1e-8, 'model', 'DC', 'opf.dc.solver', '
             IPOPT');

24       beta = 10;

26       om = opf_setup(mpc, mpopt);

28       nb = size(mpc.bus, 1);
29       br_on = find(mpc.branch(:, BR_STATUS) == 1);
30       nl = length(br_on);

32       f = mpc.branch(br_on, F_BUS);
33       t = mpc.branch(br_on, T_BUS);

35       E = sparse(1:nl, f,  1, nl, nb) + sparse(1:nl, t, -1, nl, nb);

37       Q_va = 2 * beta * (E' * E);
38       om = om.add_quad_cost('ang_diff_pen', Q_va, [], 0, {'Va'});

40       idx_i1 = om.var.idx.i1;
41       idx_iN = om.var.idx.iN;

43       [x, fval, eflag, ~, ~] = om.solve();

45       Va_solution = x(idx_i1.Va:idx_iN.Va);
46       Pg_solution = x(idx_i1.Pg:idx_iN.Pg);

48       results = mpc;
49       results.bus(:, VA) = Va_solution * 180/pi;
50       results.gen(:, PG) = Pg_solution;
51       results.f = fval;
52       results.success = (eflag > 0);

54       printpf(results);
55   end
```

# E. Sample Examples and Evaluation Prompts for ProOPF-B

## E.1. Evaluation Level 1 Sample in ProOPF-B

This section presents a complete Level 1 sample from ProOPF-B, including the natural language description, executable MATPOWER code, ground-truth execution results, and evaluation prompts for both zero-shot and few-shot settings.

**Part 1: Natural Language ($\mathcal{P}$)**

The NL description provides explicit instructions with specific numerical values for parameter modifications.

> Perform AC optimal power flow (ACOPF) on the IEEE 39-bus system (case39). Scale the active power demand at bus 1 by a factor of 1.5. Set the maximum active power output of all generators at bus 32 to 500 MW. Relax the minimum voltage magnitude constraint at bus 10 by setting VMIN to 0. Set the branch reactance for the line between bus 2 and bus 25 to 0.01. Set the OPF violation tolerance (opf.violation) to 1e-8, and write the corresponding MATPOWER code.

**Part 2: MATPOWER Code ($\mathcal{I}$)**

The generated code implements the specified modifications and solves the OPF problem.

```
1   define_constants;
2   mpc = loadcase('case39');
3
4   bus1_idx = find(mpc.bus(:, BUS_I) == 1);
5   mpc.bus(bus1_idx, PD) = mpc.bus(bus1_idx, PD) * 1.5;
6
7   gen32_idx = find(mpc.gen(:, GEN_BUS) == 32);
8   mpc.gen(gen32_idx, PMAX) = 500;
9
10  bus10_idx = find(mpc.bus(:, BUS_I) == 10);
11  mpc.bus(bus10_idx, VMIN) = 0;
12
13  branch_idx = find( ...
14      (mpc.branch(:, F_BUS) == 2  & mpc.branch(:, T_BUS) == 25) | ...
15      (mpc.branch(:, F_BUS) == 25 & mpc.branch(:, T_BUS) == 2) );
16  mpc.branch(branch_idx, BR_X) = 0.01;
17
18  mpopt = mpoption('opf.violation', 1e-8);
19  results = runopf(mpc, mpopt);
20  printf(results);
```

**Part 3: Ground-Truth Execution Results**

The ground-truth results are obtained by executing the MATPOWER code, providing reference values for correctness verification.

```
{
  "converged": true,
  "objective_value": 43009.10379295345,
  "execution_time": 1.973813,
  "error_message": null
}
```

**Part 4: Zero-Shot Evaluation Prompt**

The zero-shot prompt provides task instructions without examples, testing the LLM's ability to generate OPF code from natural language descriptions.

> **Task:** You are an expert power systems engineer specializing in optimal power flow (OPF) modeling. Your task is to generate executable MATPOWER code for an OPF problem based on the provided natural language description. The implementation must correctly distinguish explicit parameter modifications, implicit scenario-driven parameters, and structural modifications if they are present.
> **Input:** A natural language description specifying:
>
> - The base power system (e.g., IEEE case file)
>
> - Parameter modifications with explicit numerical values (e.g., scale demand at bus X by factor Y, set generator limit at bus Z to

W MW)

- Operational scenarios that imply parameter modification directions, if any (e.g., demand decreases, voltage rises, capacitor banks disconnected)

- Structural modifications (if any)

  – Problem type changes (e.g., DCOPF, ACOPF)
  – Objective function extensions (e.g., quadratic penalties on decision variables)
  – Constraint modifications (e.g., additional operational constraints)

- Solver configuration requirements (e.g., violation tolerance, solver selection)

**Output Requirements:**

1. Generate MATLAB code only

2. Load the base system using `loadcase()`

3. Directly implement all explicit parameter modifications using appropriate MATPOWER indexing

4. If implicit scenario-driven parameters are present, expose them as inputs using descriptive names of the form object_parameter_id, retrieve the original values, validate directions with assertions, and apply the input values

5. Configure solver options via `mpoption()`, including solver and model type if specified

6. Execute OPF without structural modifications via `runopf()`

7. For OPF with structural modifications, use `opf_setup()`, add objective extensions with `om.add_quad_cost()` or add constraints with appropriate optimization-model methods, solve with `om.solve()`, and extract solution components

8. Display results in a clear format, e.g., using `printpf()` for standard OPF results

**Example Input:** Perform AC optimal power flow (ACOPF) on the IEEE 39-bus system (case39). Scale the active power demand at bus 1 by a factor of 1.5. Set the maximum active power output of all generators at bus 32 to 500 MW. Relax the minimum voltage magnitude constraint at bus 10 by setting VMIN to 0. Set the branch reactance (BR_X) for the line between bus 2 and bus 25 to 0.01. Set the OPF violation tolerance (opf.violation) to 1e-8, and write the corresponding MATPOWER code.
**Expected Output Format:** Generate executable MATLAB code that implements the specified modifications and solves the OPF problem.

**Part 5: Few-Shot Evaluation Prompt**

The few-shot prompt includes in-context examples to guide the LLM's code generation, demonstrating the expected input-output mapping.

**Task:** You are an expert power systems engineer specializing in optimal power flow (OPF) modeling. Your task is to generate executable MATPOWER code for an OPF problem based on the provided natural language description. The implementation must correctly distinguish explicit parameter modifications, implicit scenario-driven parameters, and structural modifications if they are present.
**Input Format:** A natural language description specifying the base power system, explicit numerical parameter modifications, operational scenarios that may imply parameter modification directions, structural modifications if any, and solver configuration requirements.
**Output Requirements:**

1. Generate MATLAB code only

2. Load the base system using `loadcase()`

3. Directly implement all explicit parameter modifications

4. If implicit scenario-driven parameters are present, expose them as inputs, validate modification directions with assertions, and apply the input values

5. Configure solver options via `mpoption()`, including solver and model type if specified

6. Execute OPF without structural modifications via `runopf()`

7. For OPF with structural modifications, use `opf_setup()`, add objective or constraint extensions, solve with `om.solve()`, and extract solution components

8. Display results in a clear format

**Example 1:**
**Input:** Perform AC optimal power flow (ACOPF) on the IEEE 14-bus system (case14). Scale the active power demand at bus 2 by a factor of 1.2. Set the maximum active power output of all generators at bus 1 to 300 MW. Set the OPF violation tolerance (opf.violation) to 1e-6, and write the corresponding MATPOWER code.
**Output:**

```
 1   define_constants;
 2   mpc = loadcase('case14');
 3
 4   bus2_idx = find(mpc.bus(:, BUS_I) == 2);
 5   mpc.bus(bus2_idx, PD) = mpc.bus(bus2_idx, PD) * 1.2;
 6
 7   gen1_idx = find(mpc.gen(:, GEN_BUS) == 1);
 8   mpc.gen(gen1_idx, PMAX) = 300;
 9
10   mpopt = mpoption('opf.violation', 1e-6);
11   results = runopf(mpc, mpopt);
12   printpf(results);
```

**Example 2:**
**Input:** Perform AC optimal power flow (ACOPF) on the IEEE 30-bus system (case30). Set the active power demand at bus 5 to 100 MW. Set the minimum voltage magnitude constraint at bus 8 to 0.95. Set the branch reactance for the line between bus 1 and bus 2 to 0.05. Set the OPF violation tolerance (opf.violation) to 1e-8, and write the corresponding MATPOWER code.
**Output:**

```
 1   define_constants;
 2   mpc = loadcase('case30');
 3
 4   bus5_idx = find(mpc.bus(:, BUS_I) == 5);
 5   mpc.bus(bus5_idx, PD) = 100;
 6
 7   bus8_idx = find(mpc.bus(:, BUS_I) == 8);
 8   mpc.bus(bus8_idx, VMIN) = 0.95;
 9
10   branch_idx = find( ...
11       (mpc.branch(:, F_BUS) == 1  & mpc.branch(:, T_BUS) == 2) | ...
12       (mpc.branch(:, F_BUS) == 2 & mpc.branch(:, T_BUS) == 1) );
13   mpc.branch(branch_idx, BR_X) = 0.05;
14
15   mpopt = mpoption('opf.violation', 1e-8);
16   results = runopf(mpc, mpopt);
17   printpf(results);
```

**Now, generate the code for the following input:**
Perform AC optimal power flow (ACOPF) on the IEEE 39-bus system (case39). Scale the active power demand at bus 1 by a factor of 1.5. Set the maximum active power output of all generators at bus 32 to 500 MW. Relax the minimum voltage magnitude constraint at bus 10 by setting VMIN to 0. Set the branch reactance for the line between bus 2 and bus 25 to 0.01. Set the OPF violation tolerance (opf.violation) to 1e-8, and write the corresponding MATPOWER code.

## E.2. Evaluation Level 2 Sample in ProOPF-B

This section presents a complete Level 2 sample from ProOPF-B, including the scenario-based natural language description, parameterized MATPOWER function code, ground-truth execution results for multiple parameter instantiation strategies, and evaluation prompts for both zero-shot and few-shot settings.

**Part 1: Natural Language ($\mathcal{P}$)**

The NL description uses scenario-based narratives to describe parameter changes, requiring inference of modification directions rather than explicit values.

A regional grid is modeled using the IEEE 14-bus system (case14). During the late-night hours, industrial loads at bus 3 are significantly reduced, causing both active and reactive power demand to decrease substantially. The long-distance transmission lines exhibit pronounced capacitive charging effects, leading to elevated voltage levels at bus 8. To mitigate the risk of voltage exceeding the upper limit, shunt capacitor banks at bus 5 need to be disconnected. Meanwhile, the reduced reactive power demand at bus 3 further contributes to the voltage rise phenomenon. Set opf.violation to 1e-6, use MIPS as the ACOPF solver, and generate the corresponding MATPOWER code. All scenario-driven parameters are assigned using placeholder values of the form object_parameter_id, which represent the post-modification values consistent with the specified scenario directions.

### Part 2: MATPOWER Code ($\mathcal{I}$)

The generated code is structured as a function that accepts placeholder variables as parameters and includes assertion checks to validate direction constraints.

```
1   function results = opf_case14(bus_PD_3, bus_QD_3, bus_VMAX_8, bus_BS_5)
2       define_constants;
3       mpc = loadcase('case14');
4
5       bus3_idx = find(mpc.bus(:, BUS_I) == 3);
6       pd0_bus3 = mpc.bus(bus3_idx, PD);
7       assert(bus_PD_3 <= pd0_bus3);
8       mpc.bus(bus3_idx, PD) = bus_PD_3;
9
10      qd0_bus3 = mpc.bus(bus3_idx, QD);
11      assert(bus_QD_3 <= qd0_bus3);
12      mpc.bus(bus3_idx, QD) = bus_QD_3;
13
14      bus8_idx = find(mpc.bus(:, BUS_I) == 8);
15      vmax0_bus8 = mpc.bus(bus8_idx, VMAX);
16      assert(bus_VMAX_8 >= vmax0_bus8);
17      mpc.bus(bus8_idx, VMAX) = bus_VMAX_8;
18
19      bus5_idx = find(mpc.bus(:, BUS_I) == 5);
20      assert(bus_BS_5 == 0);
21      mpc.bus(bus5_idx, BS) = bus_BS_5;
22
23      mpopt = mpoption('opf.violation', 1e-6, 'opf.ac.solver', 'MIPS');
24      results = runopf(mpc, mpopt);
25      printpf(results);
26  end
```

### Part 3: Ground-Truth Execution Results

The ground-truth results are obtained by executing the parameterized function with different parameter instantiation strategies, providing reference values for correctness verification. Strategy 1 and Strategy 2 represent different parameter value assignments consistent with the scenario directions.

**Strategy 1 Parameter Values:**

```
{
  "bus_PD_3": 89.49,
  "bus_QD_3": 18.05,
  "bus_VMAX_8": 1.113,
  "bus_BS_5": 0.0
}
```

**Strategy 1 Execution Results:**

```
{
  "converged": true,
  "objective_value": 7889.236964788632,
  "execution_time": 0.915973,
  "error_message": null
}
```

**Strategy 2 Parameter Values:**

```
{
    "bus_PD_3": 84.78,
    "bus_QD_3": 17.1,
    "bus_VMAX_8": 1.166,
    "bus_BS_5": 0.0
}
```

**Strategy 2 Execution Results:**

```
{
    "converged": true,
    "objective_value": 7698.5847023253245,
    "execution_time": 0.737384,
    "error_message": null
}
```

**Part 4: Zero-Shot Evaluation Prompt**

The zero-shot prompt provides task instructions without examples, testing the LLM's ability to generate parameterized OPF functions from scenario-based natural language descriptions.

**Task:** You are an expert power systems engineer specializing in optimal power flow (OPF) modeling. Your task is to generate a parameterized MATPOWER function for an OPF problem based on the provided scenario-based natural language description. The function must distinguish explicit parameter modifications from implicit scenario-driven parameters and expose all implicit parameters as function input arguments.

**Input:** A scenario-based natural language description specifying:

- The base power system (e.g., IEEE case file)

- Parameter modifications with explicit numerical values, if any

- Operational scenarios that imply parameter modification directions (e.g., "demand decreases", "voltage rises", "capacitor banks disconnected")

- Structural modifications (if any)

- Solver configuration requirements (e.g., violation tolerance, solver selection)

- A statement indicating that scenario-driven parameters use placeholder values of the form object_parameter_id

**Output Requirements:**

1. Generate a MATLAB function only

2. Load the base system using `loadcase()`

3. Directly implement all explicit parameter modifications

4. All implicit scenario-driven parameters must be exposed as function input arguments using descriptive names of the form object_parameter_id, and for each one:
    - Retrieve the original value from the base system
    - Add an assertion to validate the modification direction:
        - For "decrease" scenarios: `assert(new_value <= original_value)`
        - For "increase" scenarios: `assert(new_value >= original_value)`
        - For "set zero" scenarios: `assert(new_value == 0)`
    - Apply the placeholder parameter value to the model

5. Configure solver options via `mpoption()` including solver specification if provided

6. Execute OPF without structural modifications via `runopf()`

7. Display results in a clear format, e.g., using `printpf()`

**Example Input:** A regional grid is modeled using the IEEE 14-bus system (case14). During the late-night hours, industrial loads at bus 3 are significantly reduced, causing both active and reactive power demand to decrease substantially. The long-distance transmission lines exhibit pronounced capacitive charging effects, leading to elevated voltage levels at bus 8. To mitigate the risk of voltage exceeding the upper limit, shunt capacitor banks at bus 5 need to be disconnected. Meanwhile, the reduced reactive power demand at bus 3 further contributes to the voltage rise phenomenon. Set opf.violation to 1e-6, use MIPS as the ACOPF solver, and generate the corresponding MATPOWER code. All scenario-driven parameters are assigned using placeholder values of the form object_parameter_id, which represent the post-modification values consistent with the specified scenario directions.

**Expected Output Format:** Generate a MATLAB function that implements the scenario-based modifications with appropriate assertion checks for direction validation.

**Part 5: Few-Shot Evaluation Prompt**

The few-shot prompt includes in-context examples to guide the LLM's function generation, demonstrating the expected input-output mapping for scenario-based parameter inference.

**Task:** You are an expert power systems engineer specializing in optimal power flow (OPF) modeling. Your task is to generate a parameterized MATPOWER function for an OPF problem based on the provided scenario-based natural language description. The function must distinguish explicit parameter modifications from implicit scenario-driven parameters and expose all implicit parameters as function input arguments.

**Input Format:** A scenario-based natural language description specifying the base power system, explicit parameter modifications if any, operational scenarios that imply parameter modification directions, structural modifications if any, solver configuration requirements, and a statement about placeholder parameter values.

**Output Requirements:**

1. Generate a MATLAB function only

2. Load the base system using `loadcase()`

3. Directly implement all explicit parameter modifications

4. For each implicit scenario-driven parameter, expose it as a function input argument, retrieve the original value, add direction validation assertions, and apply the input value

5. Configure solver options via `mpoption()` including solver specification if provided

6. Execute OPF without structural modifications via `runopf()`

7. Display results in a clear format

**Example 1:**

**Input:** A regional grid is modeled using the IEEE 14-bus system (case14). During peak demand hours, the electrical load at bus 2 increases significantly due to commercial activity. Meanwhile, generator maintenance at bus 1 reduces the available generation capacity. Set opf.violation to 1e-6, use MIPS as the ACOPF solver, and generate the corresponding MATPOWER code. All scenario-driven parameters are assigned using placeholder values of the form object_parameter_id, which represent the post-modification values consistent with the specified scenario directions.

**Output:**

```matlab
function results = opf_case14(bus_PD_2, gen_PMAX_1)
    define_constants;
    mpc = loadcase('case14');

    bus2_idx = find(mpc.bus(:, BUS_I) == 2);
    pd0_bus2 = mpc.bus(bus2_idx, PD);
    assert(bus_PD_2 >= pd0_bus2);
    mpc.bus(bus2_idx, PD) = bus_PD_2;

    gen1_idx = find(mpc.gen(:, GEN_BUS) == 1);
    pmax0_gen1 = mpc.gen(gen1_idx, PMAX);
    assert(all(gen_PMAX_1 <= pmax0_gen1));
    mpc.gen(gen1_idx, PMAX) = gen_PMAX_1;

    mpopt = mpoption('opf.violation', 1e-6, 'opf.ac.solver', 'MIPS');
    results = runopf(mpc, mpopt);
    printpf(results);
end
```

**Example 2:**
**Input:** A regional grid is modeled using the IEEE 30-bus system (case30). During a scheduled maintenance period, industrial facilities at bus 5 completely shut down, eliminating all power demand. High reactive power injection from renewable sources causes voltage to rise at bus 12, requiring adjustment of the maximum voltage limit. Set opf.violation to 1e-8, use IPOPT as the ACOPF solver, and generate the corresponding MATPOWER code. All scenario-driven parameters are assigned using placeholder values of the form object_parameter_id, which represent the post-modification values consistent with the specified scenario directions.
**Output:**

```
1    function results = opf_case30(bus_PD_5, bus_VMAX_12)
2        define_constants;
3        mpc = loadcase('case30');
4
5        bus5_idx = find(mpc.bus(:, BUS_I) == 5);
6        assert(bus_PD_5 == 0);
7        mpc.bus(bus5_idx, PD) = bus_PD_5;
8
9        bus12_idx = find(mpc.bus(:, BUS_I) == 12);
10       vmax0_bus12 = mpc.bus(bus12_idx, VMAX);
11       assert(bus_VMAX_12 >= vmax0_bus12);
12       mpc.bus(bus12_idx, VMAX) = bus_VMAX_12;
13
14       mpopt = mpoption('opf.violation', 1e-8, 'opf.ac.solver', 'IPOPT');
15       results = runopf(mpc, mpopt);
16       printpf(results);
17   end
```

**Now, generate the code for the following input:**
A regional grid is modeled using the IEEE 14-bus system (case14). During the late-night hours, industrial loads at bus 3 are significantly reduced, causing both active and reactive power demand to decrease substantially. The long-distance transmission lines exhibit pronounced capacitive charging effects, leading to elevated voltage levels at bus 8. To mitigate the risk of voltage exceeding the upper limit, shunt capacitor banks at bus 5 need to be disconnected. Meanwhile, the reduced reactive power demand at bus 3 further contributes to the voltage rise phenomenon. Set opf.violation to 1e-6, use MIPS as the ACOPF solver, and generate the corresponding MATPOWER code. All scenario-driven parameters are assigned using placeholder values of the form object_parameter_id, which represent the post-modification values consistent with the specified scenario directions.

## E.3. Evaluation Level 3 Sample in ProOPF-B

This section presents a complete Level 3 sample from ProOPF-B, including the natural language description with structural modifications, executable MATPOWER code implementing structural extensions, ground-truth execution results, and evaluation prompts for both zero-shot and few-shot settings.

### Part 1: Natural Language ($\mathcal{P}$)

The NL description includes structural modifications (problem type change and custom objective function) along with explicit parameter specifications.

Build a DC optimal power flow (DCOPF) optimization problem for the IEEE 39-bus system (case39). In addition to the default generation cost in the base case, add a quadratic penalty on phase-angle differences across all in-service transmission lines to discourage excessive angle separation (penalty weight beta = 10). Scale the active power demand at bus 1 by 1.5, set the maximum active power output of the generator(s) connected to bus 32 to 500 MW, and set the branch reactance of the line between bus 2 and bus 25 to 0.01. Assign opf.violation to 1e-8, and create the corresponding MATPOWER code.

### Part 2: MATPOWER Code ($\mathcal{I}$)

The generated code implements structural modifications using advanced OPF setup with custom objective function.

```
1    define_constants;
2    mpc = loadcase('case39');
3
4    bus1_idx = find(mpc.bus(:, BUS_I) == 1);
5    mpc.bus(bus1_idx, PD) = mpc.bus(bus1_idx, PD) * 1.5;
6
7    gen32_idx = find(mpc.gen(:, GEN_BUS) == 32);
8    mpc.gen(gen32_idx, PMAX) = 500;
```

```
9
10    branch_idx_2_25 = find( ...
11        (mpc.branch(:, F_BUS) == 2  & mpc.branch(:, T_BUS) == 25) | ...
12        (mpc.branch(:, F_BUS) == 25 & mpc.branch(:, T_BUS) == 2) );
13    mpc.branch(branch_idx_2_25, BR_X) = 0.01;
14
15    mpopt = mpoption('opf.violation', 1e-8, 'model', 'DC');
16
17    beta = 10;
18
19    om = opf_setup(mpc, mpopt);
20
21    nb = size(mpc.bus, 1);
22    br_on = find(mpc.branch(:, BR_STATUS) == 1);
23    nl = length(br_on);
24
25    f = mpc.branch(br_on, F_BUS);
26    t = mpc.branch(br_on, T_BUS);
27
28    E = sparse(1:nl, f,  1, nl, nb) + sparse(1:nl, t, -1, nl, nb);
29
30    Q_va = 2 * beta * (E' * E);
31    om = om.add_quad_cost('ang_diff_pen', Q_va, [], 0, {'Va'});
32
33    idx_i1 = om.var.idx.i1;
34    idx_iN = om.var.idx.iN;
35
36    [x, fval, eflag, ~, ~] = om.solve();
37
38    Va_solution = x(idx_i1.Va:idx_iN.Va);
39    Pg_solution = x(idx_i1.Pg:idx_iN.Pg);
40
41    results = mpc;
42    results.bus(:, VA) = Va_solution * 180/pi;
43    results.gen(:, PG) = Pg_solution;
44    results.f = fval;
45    results.success = (eflag > 0);
```

**Part 3: Ground-Truth Execution Results**

The ground-truth results are obtained by executing the MATPOWER code with structural modifications, providing reference values for correctness verification.

```
{
  "converged": true,
  "objective_value": 42306.19654969372,
  "execution_time": 17.461786,
  "error_message": null
}
```

**Part 4: Zero-Shot Evaluation Prompt**

The zero-shot prompt provides task instructions without examples, testing the LLM's ability to generate OPF code with structural modifications from natural language descriptions.

**Task:** You are an expert power systems engineer specializing in optimal power flow (OPF) modeling with advanced structural extensions. Your task is to generate executable MATPOWER code for an OPF problem that includes structural modifications and parameter modifications. The implementation must directly apply explicit numerical parameter changes and use MATPOWER's optimization model interface for structural extensions.
**Input:** A natural language description specifying:

- The base power system (e.g., IEEE case file)

- Parameter modifications with explicit numerical values (e.g., scale demand at bus X by factor Y, set generator limit at bus Z to W MW)

- Operational scenarios that imply parameter modification directions, if any

- Structural modifications:

  - Problem type changes (e.g., DCOPF, ACOPF)
  - Objective function extensions (e.g., quadratic penalties on decision variables)
  - Constraint modifications (e.g., additional operational constraints)

- Solver configuration requirements (e.g., violation tolerance, solver selection)

**Output Requirements:**

1. Generate MATLAB code only

2. Load the base system using `loadcase()`

3. Directly implement all explicit parameter modifications using appropriate MATPOWER indexing

4. If implicit scenario-driven parameters are present, expose them as inputs, validate their modification directions with assertions, and apply the input values

5. Configure solver options via `mpoption()` including model type if structural modification specifies a problem type change

6. Use `opf_setup()` to create an optimization model object

7. Implement structural modifications:

   - For objective extensions: Construct the quadratic cost matrix $Q$ based on the specified form, then use `om.add_quad_cost()` to add the term
   - For constraint extensions: Use appropriate constraint addition methods

8. Solve using `om.solve()` and extract solution components

9. Construct a results structure with solution values

10. Display results in a clear format

**Example Input:** Build a DC optimal power flow (DCOPF) optimization problem for the IEEE 39-bus system (case39). In addition to the default generation cost in the base case, add a quadratic penalty on phase-angle differences across all in-service transmission lines to discourage excessive angle separation (penalty weight beta = 10). Scale the active power demand at bus 1 by 1.5, set the maximum active power output of the generator(s) connected to bus 32 to 500 MW, and set the branch reactance of the line between bus 2 and bus 25 to 0.01. Assign opf.violation to 1e-8, and create the corresponding MATPOWER code.
**Expected Output Format:** Generate executable MATLAB code that implements the structural modifications and parameter modifications, using `opf_setup()` and `om.add_quad_cost()` for objective extensions.

**Part 5: Few-Shot Evaluation Prompt**

The few-shot prompt includes in-context examples to guide the LLM's code generation, demonstrating the expected input-output mapping for problems with structural modifications.

**Task:** You are an expert power systems engineer specializing in optimal power flow (OPF) modeling with advanced structural extensions. Your task is to generate executable MATPOWER code for an OPF problem that includes structural modifications and parameter modifications. The implementation must directly apply explicit numerical parameter changes and use MATPOWER's optimization model interface for structural extensions.
**Input Format:** A natural language description specifying the base power system, explicit numerical parameter modifications, operational scenarios that may imply parameter modification directions, structural modifications, and solver configuration requirements.
**Output Requirements:**

1. Generate MATLAB code only

2. Load the base system and directly apply explicit parameter modifications

3. If implicit scenario-driven parameters are present, expose them as inputs, validate their modification directions with assertions, and apply the input values

4. Configure solver options including model type if structural modification specifies a problem type change

5. Use `opf_setup()` to create an optimization model object

6. Implement structural modifications using appropriate methods (e.g., `om.add_quad_cost()` for objective extensions)

7. Solve using `om.solve()` and extract solution components

8. Construct a results structure with solution values

9. Display results in a clear format

**Example 1:**
**Input:** Formulate a DC optimal power flow (DCOPF) problem for the IEEE 14-bus system (case14). Add a quadratic penalty on phase-angle differences across all in-service transmission lines with penalty weight beta = 5. Scale the active power demand at bus 2 by 1.2. Set opf.violation to 1e-6, and write the corresponding MATPOWER code.
**Output:**

```
1    define_constants;
2    mpc = loadcase('case14');
3
4    bus2_idx = find(mpc.bus(:, BUS_I) == 2);
5    mpc.bus(bus2_idx, PD) = mpc.bus(bus2_idx, PD) * 1.2;
6
7    mpopt = mpoption('opf.violation', 1e-6, 'model', 'DC');
8
9    beta = 5;
10
11   om = opf_setup(mpc, mpopt);
12
13   nb = size(mpc.bus, 1);
14   br_on = find(mpc.branch(:, BR_STATUS) == 1);
15   nl = length(br_on);
16
17   f = mpc.branch(br_on, F_BUS);
18   t = mpc.branch(br_on, T_BUS);
19
20   E = sparse(1:nl, f,  1, nl, nb) + sparse(1:nl, t, -1, nl, nb);
21
22   Q_va = 2 * beta * (E' * E);
23   om = om.add_quad_cost('ang_diff_pen', Q_va, [], 0, {'Va'});
24
25   idx_i1 = om.var.idx.i1;
26   idx_iN = om.var.idx.iN;
27
28   [x, fval, eflag, ~, ~] = om.solve();
29
30   Va_solution = x(idx_i1.Va:idx_iN.Va);
31   Pg_solution = x(idx_i1.Pg:idx_iN.Pg);
32
33   results = mpc;
34   results.bus(:, VA) = Va_solution * 180/pi;
35   results.gen(:, PG) = Pg_solution;
36   results.f = fval;
37   results.success = (eflag > 0);
```

**Example 2:**
**Input:** Build an AC optimal power flow (ACOPF) problem for the IEEE 30-bus system (case30). Add a quadratic penalty on generator active power outputs with penalty weight gamma = 0.1. Set the active power demand at bus 5 to 100 MW. Set the minimum voltage magnitude constraint at bus 8 to 0.95. Set opf.violation to 1e-8, and write the corresponding MATPOWER code.
**Output:**

```
1    define_constants;
2    mpc = loadcase('case30');
3
4    bus5_idx = find(mpc.bus(:, BUS_I) == 5);
5    mpc.bus(bus5_idx, PD) = 100;
```

```
6
7    bus8_idx = find(mpc.bus(:, BUS_I) == 8);
8    mpc.bus(bus8_idx, VMIN) = 0.95;
9
10   mpopt = mpoption('opf.violation', 1e-8);
11
12   gamma = 0.1;
13
14   om = opf_setup(mpc, mpopt);
15
16   ng = size(mpc.gen, 1);
17   Q_pg = 2 * gamma * speye(ng);
18   om = om.add_quad_cost('pg_penalty', Q_pg, [], 0, {'Pg'});
19
20   idx_i1 = om.var.idx.i1;
21   idx_iN = om.var.idx.iN;
22
23   [x, fval, eflag, ~, ~] = om.solve();
24
25   Va_solution = x(idx_i1.Va:idx_iN.Va);
26   Vm_solution = x(idx_i1.Vm:idx_iN.Vm);
27   Pg_solution = x(idx_i1.Pg:idx_iN.Pg);
28   Qg_solution = x(idx_i1.Qg:idx_iN.Qg);
29
30   results = mpc;
31   results.bus(:, VA) = Va_solution * 180/pi;
32   results.bus(:, VM) = Vm_solution;
33   results.gen(:, PG) = Pg_solution;
34   results.gen(:, QG) = Qg_solution;
35   results.f = fval;
36   results.success = (eflag > 0);
```

**Now, generate the code for the following input:**
Build a DC optimal power flow (DCOPF) optimization problem for the IEEE 39-bus system (case39). In addition to the default generation cost in the base case, add a quadratic penalty on phase-angle differences across all in-service transmission lines to discourage excessive angle separation (penalty weight beta = 10). Scale the active power demand at bus 1 by 1.5, set the maximum active power output of the generator(s) connected to bus 32 to 500 MW, and set the branch reactance of the line between bus 2 and bus 25 to 0.01. Assign opf.violation to 1e-8, and create the corresponding MATPOWER code.

## E.4. Evaluation Level 4 Sample in ProOPF-B

This section presents a complete Level 4 sample from ProOPF-B, including a scenario-based natural language description with structural modifications, parameterized MATPOWER function code implementing structural extensions, ground-truth execution results for multiple parameter instantiation strategies, and evaluation prompts for both zero-shot and few-shot settings.

### Part 1: Natural Language ($\mathcal{P}$)

The NL description combines semantic parameter inference with structural modifications. The model must infer parameter modification directions from the scenario while also implementing a DCOPF formulation with an additional quadratic objective term.

Build a DC optimal power flow (DCOPF) optimization problem for the IEEE 39-bus system (case39). During a low-load operating period, the active power demand at bus 1 is expected to decrease because of reduced industrial consumption. A generator connected to bus 32 is undergoing partial maintenance, reducing its available maximum active power output. In addition, a transmission upgrade on the line between bus 2 and bus 25 lowers the branch reactance. To discourage excessive phase-angle separation under this stressed operating condition, add a quadratic penalty on phase-angle differences across all in-service transmission lines with penalty weight beta = 10. Assign opf.violation to 1e-8, and generate the corresponding MATPOWER code. All scenario-driven parameters are assigned using placeholder values of the form object_parameter_id, which represent the post-modification values consistent with the specified scenario directions.

### Part 2: MATPOWER Code ($\mathcal{I}$)

The generated code is structured as a parameterized function that accepts placeholder variables, validates the semantic modification directions, and implements the structural extension through a custom quadratic objective term.

```matlab
 1  function results = opf_case39_level4(bus_PD_1, gen_PMAX_32, branch_BR_X_2_25)
 2      define_constants;
 3      mpc = loadcase('case39');
 4
 5      bus1_idx = find(mpc.bus(:, BUS_I) == 1);
 6      pd0_bus1 = mpc.bus(bus1_idx, PD);
 7      assert(bus_PD_1 <= pd0_bus1);
 8      mpc.bus(bus1_idx, PD) = bus_PD_1;
 9
10      gen32_idx = find(mpc.gen(:, GEN_BUS) == 32);
11      pmax0_gen32 = mpc.gen(gen32_idx, PMAX);
12      assert(all(gen_PMAX_32 <= pmax0_gen32));
13      mpc.gen(gen32_idx, PMAX) = gen_PMAX_32;
14
15      branch_idx_2_25 = find( ...
16          (mpc.branch(:, F_BUS) == 2  & mpc.branch(:, T_BUS) == 25) | ...
17          (mpc.branch(:, F_BUS) == 25 & mpc.branch(:, T_BUS) == 2) );
18      brx0_2_25 = mpc.branch(branch_idx_2_25, BR_X);
19      assert(all(branch_BR_X_2_25 <= brx0_2_25));
20      mpc.branch(branch_idx_2_25, BR_X) = branch_BR_X_2_25;
21
22      mpopt = mpoption('opf.violation', 1e-8, 'model', 'DC');
23
24      beta = 10;
25      om = opf_setup(mpc, mpopt);
26
27      nb = size(mpc.bus, 1);
28      br_on = find(mpc.branch(:, BR_STATUS) == 1);
29      nl = length(br_on);
30
31      f = mpc.branch(br_on, F_BUS);
32      t = mpc.branch(br_on, T_BUS);
33
34      E = sparse(1:nl, f,  1, nl, nb) + sparse(1:nl, t, -1, nl, nb);
35
36      Q_va = 2 * beta * (E' * E);
37      om = om.add_quad_cost('ang_diff_pen', Q_va, [], 0, {'Va'});
38
39      idx_i1 = om.var.idx.i1;
40      idx_iN = om.var.idx.iN;
41
42      [x, fval, eflag, ~, ~] = om.solve();
43
44      Va_solution = x(idx_i1.Va:idx_iN.Va);
45      Pg_solution = x(idx_i1.Pg:idx_iN.Pg);
46
47      results = mpc;
48      results.bus(:, VA) = Va_solution * 180/pi;
49      results.gen(:, PG) = Pg_solution;
50      results.f = fval;
51      results.success = (eflag > 0);
52  end
```

**Part 3: Ground-Truth Execution Results**

The ground-truth results are obtained by executing the parameterized structural OPF function with different parameter instantiation strategies. Strategy 1 and Strategy 2 represent different post-modification values that satisfy the semantic direction constraints.

**Strategy 1 Parameter Values:**

```
    {
      "bus_PD_1": 87.96,
      "gen_PMAX_32": 500.0,
      "branch_BR_X_2_25": 0.01
    }
```

**Strategy 1 Execution Results:**

```
{
  "converged": true,
  "objective_value": 41572.88463126713,
  "execution_time": 18.284517,
  "error_message": null
}
```

**Strategy 2 Parameter Values:**

```
{
  "bus_PD_1": 76.23,
  "gen_PMAX_32": 450.0,
  "branch_BR_X_2_25": 0.008
}
```

**Strategy 2 Execution Results:**

```
{
  "converged": true,
  "objective_value": 40938.51790482646,
  "execution_time": 17.936204,
  "error_message": null
}
```

**Part 4: Zero-Shot Evaluation Prompt**

The zero-shot prompt provides task instructions without examples, testing the LLM's ability to generate parameterized OPF functions that combine semantic parameter inference with structural OPF extensions.

**Task:** You are an expert power systems engineer specializing in optimal power flow (OPF) modeling. Your task is to generate a parameterized MATPOWER function for an OPF problem that includes both structural modifications and parameter modifications based on the provided scenario-based natural language description.

**Input:** A scenario-based natural language description specifying:

- The base power system (e.g., IEEE case file)

- Parameter modifications with explicit numerical values (e.g., scale demand at bus X by factor Y, set generator limit at bus Z to W MW)

- Operational scenarios that imply parameter modification directions (e.g., demand decreases, generator availability decreases, branch reactance decreases)

- Structural modifications:

    – Problem type changes (e.g., DCOPF, ACOPF)
    – Objective function extensions (e.g., quadratic penalties on decision variables)
    – Constraint modifications (e.g., additional operational constraints)

- Solver configuration requirements (e.g., violation tolerance, solver selection)

- A statement indicating that scenario-driven parameters use placeholder values of the form object_parameter_id

**Output Requirements:**

1. Generate a MATLAB function only

2. Load the base system using `loadcase()`

3. Directly implement all explicit parameter modifications

4. All implicit scenario-driven parameters must be exposed as function input arguments using descriptive names of the form object_parameter_id, and for each one:

- Retrieve the original value from the base system
- Add an assertion to validate the modification direction:
  - For "decrease" scenarios: `assert(new_value <= original_value)`
  - For "increase" scenarios: `assert(new_value >= original_value)`
  - For "set zero" scenarios: `assert(new_value == 0)`
- Apply the input argument value to the model

5. Execute OPF without structural modifications via `runopf()`

6. For OPF with structural modifications:

   - Use `opf_setup()` to create an optimization model object
   - For objective extensions: construct the quadratic cost matrix $Q$ based on the specified form, then use `om.add_quad_cost()` to add the term
   - For constraint extensions: use appropriate constraint addition methods
   - Solve using `om.solve()` and extract solution components

7. Configure solver options via `mpoption()` including model type if structural modification specifies a problem type change (e.g., DCOPF, ACOPF)

8. Display results in a nice format

**Example Input:** Build a DC optimal power flow (DCOPF) optimization problem for the IEEE 39-bus system (case39). During a low-load operating period, the active power demand at bus 1 is expected to decrease because of reduced industrial consumption. A generator connected to bus 32 is undergoing partial maintenance, reducing its available maximum active power output. In addition, a transmission upgrade on the line between bus 2 and bus 25 lowers the branch reactance. To discourage excessive phase-angle separation under this stressed operating condition, add a quadratic penalty on phase-angle differences across all in-service transmission lines with penalty weight beta = 10. Assign opf.violation to 1e-8, and generate the corresponding MATPOWER code. All scenario-driven parameters are assigned using placeholder values of the form object_parameter_id, which represent the post-modification values consistent with the specified scenario directions.
**Expected Output Format:** Generate a MATLAB function that implements the scenario-based parameter modifications with direction validation and the structural OPF extension using `opf_setup()` and `om.add_quad_cost()`.

**Part 5: Few-Shot Evaluation Prompt**

The few-shot prompt includes in-context examples to guide the LLM's function generation, demonstrating the expected input-output mapping for scenario-based parameter inference combined with structural OPF extensions.

**Task:** You are an expert power systems engineer specializing in optimal power flow (OPF) modeling. Your task is to generate a parameterized MATPOWER function for an OPF problem that includes both structural modifications and parameter modifications based on the provided scenario-based natural language description.
**Input Format:** A scenario-based natural language description specifying the base power system, explicit parameter modifications if any, operational scenarios that imply parameter modification directions, structural modifications, solver configuration requirements, and a statement about placeholder parameter values.
**Output Requirements:**

1. Generate a MATLAB function only

2. Load the base system using `loadcase()`

3. Directly implement all explicit parameter modifications

4. For each implicit scenario-driven parameter, expose it as a function input argument using the object_parameter_id naming convention, retrieve the original value, validate the modification direction with assertions, and apply the input value

5. For OPF with structural modifications, use `opf_setup()`, add objective extensions with `om.add_quad_cost()` or add constraints with appropriate methods, solve with `om.solve()`, and extract solution components

6. Configure solver options via `mpoption()` including model type if required

7. Display results in a nice format

**Example 1:**
**Input:** Formulate a DC optimal power flow (DCOPF) problem for the IEEE 14-bus system (case14). During a conservation period, the active power demand at bus 2 decreases. To improve angular stability, add a quadratic penalty on phase-angle differences across

all in-service transmission lines with penalty weight beta = 5. Set opf.violation to 1e-6, and generate the corresponding MATPOWER code. All scenario-driven parameters are assigned using placeholder values of the form object_parameter_id, which represent the post-modification values consistent with the specified scenario directions.

**Output:**

```
1  function results = opf_case14_level4(bus_PD_2)
2      define_constants;
3      mpc = loadcase('case14');
4
5      bus2_idx = find(mpc.bus(:, BUS_I) == 2);
6      pd0_bus2 = mpc.bus(bus2_idx, PD);
7      assert(bus_PD_2 <= pd0_bus2);
8      mpc.bus(bus2_idx, PD) = bus_PD_2;
9
10     mpopt = mpoption('opf.violation', 1e-6, 'model', 'DC');
11
12     beta = 5;
13     om = opf_setup(mpc, mpopt);
14
15     nb = size(mpc.bus, 1);
16     br_on = find(mpc.branch(:, BR_STATUS) == 1);
17     nl = length(br_on);
18
19     f = mpc.branch(br_on, F_BUS);
20     t = mpc.branch(br_on, T_BUS);
21
22     E = sparse(1:nl, f,  1, nl, nb) + sparse(1:nl, t, -1, nl, nb);
23
24     Q_va = 2 * beta * (E' * E);
25     om = om.add_quad_cost('ang_diff_pen', Q_va, [], 0, {'Va'});
26
27     idx_i1 = om.var.idx.i1;
28     idx_iN = om.var.idx.iN;
29
30     [x, fval, eflag, ~, ~] = om.solve();
31
32     Va_solution = x(idx_i1.Va:idx_iN.Va);
33     Pg_solution = x(idx_i1.Pg:idx_iN.Pg);
34
35     results = mpc;
36     results.bus(:, VA) = Va_solution * 180/pi;
37     results.gen(:, PG) = Pg_solution;
38     results.f = fval;
39     results.success = (eflag > 0);
40  end
```

**Example 2:**

**Input:** Build an AC optimal power flow (ACOPF) problem for the IEEE 30-bus system (case30). During generator wear mitigation, the active power dispatch should be discouraged by adding a quadratic penalty on generator active power outputs with penalty weight gamma = 0.1. Meanwhile, industrial load at bus 5 decreases during an off-peak period, and voltage support requirements at bus 8 become more stringent, increasing the minimum voltage magnitude. Set opf.violation to 1e-8, and generate the corresponding MATPOWER code. All scenario-driven parameters are assigned using placeholder values of the form object_parameter_id, which represent the post-modification values consistent with the specified scenario directions.

**Output:**

```
1  function results = opf_case30_level4(bus_PD_5, bus_VMIN_8)
2      define_constants;
3      mpc = loadcase('case30');
4
5      bus5_idx = find(mpc.bus(:, BUS_I) == 5);
6      pd0_bus5 = mpc.bus(bus5_idx, PD);
7      assert(bus_PD_5 <= pd0_bus5);
8      mpc.bus(bus5_idx, PD) = bus_PD_5;
9
```

```
10        bus8_idx = find(mpc.bus(:, BUS_I) == 8);
11        vmin0_bus8 = mpc.bus(bus8_idx, VMIN);
12        assert(bus_VMIN_8 >= vmin0_bus8);
13        mpc.bus(bus8_idx, VMIN) = bus_VMIN_8;
14
15        mpopt = mpoption('opf.violation', 1e-8);
16
17        gamma = 0.1;
18        om = opf_setup(mpc, mpopt);
19
20        ng = size(mpc.gen, 1);
21        Q_pg = 2 * gamma * speye(ng);
22        om = om.add_quad_cost('pg_penalty', Q_pg, [], 0, {'Pg'});
23
24        idx_i1 = om.var.idx.i1;
25        idx_iN = om.var.idx.iN;
26
27        [x, fval, eflag, ~, ~] = om.solve();
28
29        Va_solution = x(idx_i1.Va:idx_iN.Va);
30        Vm_solution = x(idx_i1.Vm:idx_iN.Vm);
31        Pg_solution = x(idx_i1.Pg:idx_iN.Pg);
32        Qg_solution = x(idx_i1.Qg:idx_iN.Qg);
33
34        results = mpc;
35        results.bus(:, VA) = Va_solution * 180/pi;
36        results.bus(:, VM) = Vm_solution;
37        results.gen(:, PG) = Pg_solution;
38        results.gen(:, QG) = Qg_solution;
39        results.f = fval;
40        results.success = (eflag > 0);
41    end
```

**Now, generate the code for the following input:**

Build a DC optimal power flow (DCOPF) optimization problem for the IEEE 39-bus system (case39). During a low-load operating period, the active power demand at bus 1 is expected to decrease because of reduced industrial consumption. A generator connected to bus 32 is undergoing partial maintenance, reducing its available maximum active power output. In addition, a transmission upgrade on the line between bus 2 and bus 25 lowers the branch reactance. To discourage excessive phase-angle separation under this stressed operating condition, add a quadratic penalty on phase-angle differences across all in-service transmission lines with penalty weight beta = 10. Assign opf.violation to 1e-8, and generate the corresponding MATPOWER code. All scenario-driven parameters are assigned using placeholder values of the form object_parameter_id, which represent the post-modification values consistent with the specified scenario directions.

## F. Evaluation Workflow for ProOPF-B

### F.1. Evaluation Workflow

**Concrete OPF Modeling (Levels 1/3).** For concrete modeling, the LLM generates a complete implementation $\widehat{\mathcal{I}}$ that can be directly executed to obtain $\widehat{f^*}$, which is compared against the ground-truth $f^*$ for correctness verification.

**Abstract OPF Modeling (Levels 2/4).** For abstract modeling, the LLM generates a parameterized function $\widehat{\mathcal{I}}(\cdot)$ that requires parameter instantiation $\pi$ before execution. The generated implementation is validated by executing $\widehat{\mathcal{I}}(\pi)$ and comparing $\widehat{f^*}(\pi)$ against the ground-truth $f^*(\pi)$ obtained from the expert-annotated implementation $\mathcal{I}(\pi)$.

This distinction reflects the fundamental difference between explicit parameter specification and semantic parameter inference in OPF modeling tasks, as illustrated in Figure 5.

---

**Algorithm 6** Evaluation for Concrete OPF Modeling (Levels 1/3)

---

**Require:** Sample set $\mathcal{Z} = \{z_i\}_{i=1}^N$ where $z_i = \{\mathcal{M}_i, \mathcal{P}_i, \mathcal{I}_i\}$, LLM model $p_{\text{LLM}}$, instruction specification $\tau$, tolerance threshold $\epsilon > 0$ (a small positive number)
**Ensure:** Correctness rate $r \in [0, 1]$
 1: Initialize correct count: $c_{\text{total}} \leftarrow 0$
 2: **for** $i = 1$ to $N$ **do**
 3:      Generate implementation from NL description: $\widehat{\mathcal{I}}_i \sim p_{\text{LLM}}(\cdot \mid \mathcal{P}_i, \tau)$
 4:      Execute generated implementation: $\widehat{f^*}_i \leftarrow \widehat{\mathcal{I}}_i()$
 5:      Execute ground-truth implementation: $f_i^* \leftarrow \mathcal{I}_i()$
 6:      Check correctness: $c_i \leftarrow \mathbf{1}[\|\widehat{f^*}_i - f_i^*\| \leq \epsilon]$
 7:      Update count: $c_{\text{total}} \leftarrow c_{\text{total}} + c_i$
 8: **end for**
 9: Compute correctness rate: $r \leftarrow c_{\text{total}}/N$
10: **return** $r$

---

**Algorithm 7** Evaluation for Abstract OPF Modeling (Levels 2/4)

---

**Require:** Sample set $\mathcal{Z} = \{z_i\}_{i=1}^N$ where $z_i = \{\mathcal{M}_i, \mathcal{P}_i, \mathcal{I}_i(\cdot), \pi_i\}$, LLM model $p_{\text{LLM}}$, instruction specification $\tau$, tolerance threshold $\epsilon > 0$ (a small positive number)
**Ensure:** Correctness rate $r \in [0, 1]$
 1: Initialize correct count: $c_{\text{total}} \leftarrow 0$
 2: **for** $i = 1$ to $N$ **do**
 3:      Generate parameterized function from NL description: $\widehat{\mathcal{I}}_i(\cdot) \sim p_{\text{LLM}}(\cdot \mid \mathcal{P}_i, \tau)$
 4:      Execute generated function with parameters: $\widehat{f^*}_i(\pi_i) \leftarrow \widehat{\mathcal{I}}_i(\pi_i)$
 5:      Execute ground-truth function with parameters: $f_i^*(\pi_i) \leftarrow \mathcal{I}_i(\pi_i)$
 6:      Check correctness: $c_i \leftarrow \mathbf{1}[\|\widehat{f^*}_i(\pi_i) - f_i^*(\pi_i)\| \leq \epsilon]$
 7:      Update count: $c_{\text{total}} \leftarrow c_{\text{total}} + c_i$
 8: **end for**
 9: Compute correctness rate: $r \leftarrow c_{\text{total}}/N$
10: **return** $r$

---

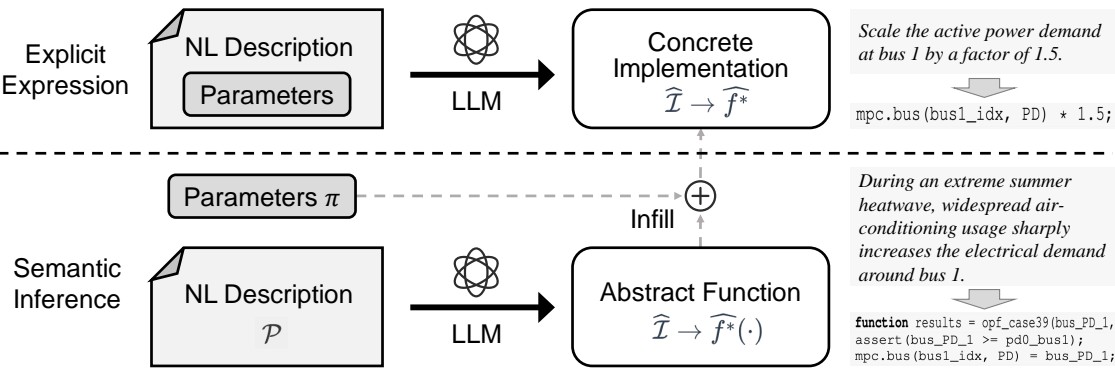

*Figure 5.* Schematic comparison of evaluation workflows for concrete (Levels 1/3) and abstract (Levels 2/4) OPF modeling in ProOPF-B, illustrating the key difference in validation procedures.

# G. Six-Dimensional Capability Analysis Framework

To provide deeper insights into the performance gaps among different models on ProOPF-B, we introduce a six-dimensional capability analysis framework that decomposes model performance into fundamental competencies. These dimensions are designed to capture distinct aspects of the OPF modeling task and explain why certain models excel or struggle on specific difficulty levels. The six dimensions are defined as follows:

### G.1. Dimension 1: Executable Rate

**Definition:** The average executable rate across all difficulty levels (Levels 1–4).

**Calculation:** For each level $\ell \in \{1, 2, 3, 4\}$, let $E_\ell$ denote the number of test cases that produce executable code (no syntax or runtime errors before solver invocation), and $T_\ell$ denote the total number of test cases at level $\ell$. The executable rate is:

$$\text{Executable Rate} = \frac{1}{4} \sum_{\ell=1}^{4} \frac{E_\ell}{T_\ell} \times 100\% \tag{24}$$

**Interpretation:** This dimension measures the model's ability to generate syntactically and semantically valid MATPOWER code that can be executed without errors. A low executable rate indicates fundamental coding proficiency issues, such as incorrect API usage, invalid syntax, or misunderstanding of the MATPOWER toolchain.

### G.2. Dimension 2: Explicit Parameter Modification Correctness

**Definition:** The correctness rate on tasks requiring explicit parameter modifications (Levels 1 and 3).

**Calculation:** Let $C_1$ and $C_3$ denote the number of correct implementations at Levels 1 and 3, respectively. The explicit parameter modification correctness is:

$$\text{Explicit Param Mod} = \frac{1}{2} \left( \frac{C_1}{T_1} + \frac{C_3}{T_3} \right) \times 100\% \tag{25}$$

**Interpretation:** This dimension evaluates the model's ability to follow explicit numerical parameter specifications and apply them correctly to the OPF model. High performance on this dimension indicates strong instruction-following capability for concrete parameter assignments.

### G.3. Dimension 3: Semantic Parameter Inference Correctness

**Definition:** The correctness rate on tasks requiring semantic parameter inference from natural language descriptions (Levels 2 and 4).

**Calculation:** Let $C_2$ and $C_4$ denote the number of correct implementations at Levels 2 and 4, respectively. The semantic parameter inference correctness is:

$$\text{Semantic Param Inf} = \frac{1}{2} \left( \frac{C_2}{T_2} + \frac{C_4}{T_4} \right) \times 100\% \tag{26}$$

**Interpretation:** This dimension measures the model's ability to infer appropriate numerical parameters from semantic descriptions (e.g., "reduce line capacity by 20%" or "increase load by a moderate amount"). This requires both natural language understanding and domain knowledge about reasonable parameter ranges in power systems.

### G.4. Dimension 4: Structural Modification Identification

**Definition:** The rate at which models correctly identify that structural modifications to the OPF formulation are required (Levels 3 and 4).

**Calculation:** Let $S_3$ and $S_4$ denote the number of implementations at Levels 3 and 4 where the model recognizes and attempts to implement structural changes (e.g., adding new variables, modifying constraints, or introducing new objective terms), regardless of execution success. The structural modification identification rate is:

$$\text{Structural Mod} = \frac{1}{2} \left( \frac{S_3}{T_3} + \frac{S_4}{T_4} \right) \times 100\% \tag{27}$$

where $T_3$ and $T_4$ are the total number of test cases at Levels 3 and 4.

**Interpretation:** This dimension measures the model's ability to recognize when a task requires non-trivial structural extensions to the base OPF formulation, such as adding new decision variables, modifying constraints, or introducing new objective terms, distinguishing it from simple parameter modifications in Levels 1–2. A high score indicates the model understands *what* structural changes are needed, even if implementation challenges prevent successful execution. This separates conceptual understanding from coding proficiency.

### G.5. Dimension 5: Resolution Specification Correctness

**Definition:** The rate at which models correctly configure the resolution specification $\mathcal{R}$, including solver selection (e.g., MIPS, IPOPT, KNITRO), convergence criteria (e.g., constraint violation tolerance `opf_violation`), and output configuration settings.

**Calculation:** Let $R_\ell$ denote the number of implementations at level $\ell$ where the resolution specification $\mathcal{R}$ is correctly configured (i.e., the solver, convergence tolerance, and output settings match the requirements specified in the task). The resolution specification correctness is:

$$\text{Resolution Spec} = \frac{1}{4} \sum_{\ell=1}^{4} \frac{R_\ell}{T_\ell} \times 100\% \tag{28}$$

where $T_\ell$ denotes the total number of test cases at level $\ell$.

**Interpretation:** This dimension measures the model's ability to correctly interpret and implement the resolution specification $\mathcal{R}$ across all difficulty levels. The resolution specification controls how the OPF problem is solved, including which optimization solver to use, what convergence tolerance to apply, and how results should be reported. Since $\mathcal{R}$ is specified in every task regardless of difficulty level, this dimension evaluates fundamental proficiency in translating solver requirements into correct MATPOWER API calls. A model may generate executable code but still fail this dimension if it uses the wrong solver, sets incorrect tolerance values, or misconfigures output options.

### G.6. Dimension 6: Executable Correctness Rate

**Definition:** The average correctness rate among executable code across all levels.

**Calculation:** The executable correctness rate is:

$$\text{Exec Correct Rate} = \frac{\sum_{\ell=1}^{4} E_\ell^c}{\sum_{\ell=1}^{4} E_\ell} \times 100\% \tag{29}$$

where $E_\ell^c$ denotes the number of executable and correct implementations at level $\ell$.

**Interpretation:** This dimension measures the model's *conditional* correctness given that it produces executable code. A high executable correctness rate but low overall performance suggests that the model understands the task conceptually but struggles with implementation details. Conversely, a low executable correctness rate indicates that even when the code runs, it often produces incorrect results due to logical errors or misunderstanding of the problem specification.

### G.7. Summary and Usage

The six-dimensional framework provides a comprehensive view of model capabilities across different aspects of OPF modeling. By analyzing these dimensions jointly (e.g., via radar charts as in Table 4), we can identify specific bottlenecks:

- Low **Executable Rate** $\Rightarrow$ basic coding/API proficiency issues

- Low **Semantic Param Inf** $\Rightarrow$ difficulty in natural language understanding and domain knowledge

- Low **Structural Mod** $\Rightarrow$ failure to recognize when structural modifications are needed

- High **Structural Mod** but low Level 3/4 performance $\Rightarrow$ conceptual understanding exists but implementation barriers prevent success

- High **Exec Correct Rate** but low overall performance $\Rightarrow$ implementation barriers rather than conceptual misunderstanding

This diagnostic framework enables targeted improvements for future model development and training strategies.

# H. Additional Experiments Results

## H.1. Out-of-Distribution Generalization

To evaluate the robustness of the fine-tuned model, we conduct comprehensive out-of-distribution (OOD) generalization analyses across cross-system and cross-scenario dimensions.

**Cross-System Generalization.** We evaluate the model on 36 additional Level-1 samples derived from unseen power system architectures. As shown in Table 12, the model exhibits strong cross-system generalization, maintaining comparable performance on OOD systems (72.22% few-shot) relative to in-distribution (ID) systems (75.00% few-shot). This stability suggests that the model effectively learns the underlying OPF implementation workflow, which remains largely consistent across varying grid topologies.

**Cross-Scenario Generalization.** To assess generalization to novel operational conditions, we introduce 30 Level-2 and 26 Level-4 samples generated from extended scenario trees. Table 12 demonstrates that the model experiences only marginal degradation on OOD scenarios. Notably, it maintains nontrivial performance on unseen scenario branches, indicating the acquisition of transferable, scenario-conditioned parameter inference capabilities rather than mere memorization of the training scenario trees.

*Table 12.* OOD generalization performance (%) on unseen power systems and operational scenarios.

| Generalization Type | Setting | Zero-shot | Few-shot |
|---|---|---|---|
| Cross-System (Level 1) | ID Systems | 63.90 | 75.00 |
| | OOD Systems | 66.67 | 72.22 |
| Cross-Scenario (Level 2) | ID Scenarios | 23.30 | 33.30 |
| | OOD Scenarios | 16.67 | 30.00 |
| Cross-Scenario (Level 4) | ID Scenarios | 7.69 | 11.54 |
| | OOD Scenarios | 7.69 | 7.69 |

## H.2. Extended Model Performance Comparison

Table 13 presents extended experimental results with additional model variants. The findings are consistent with the main results: concrete modeling (Levels 1/3) significantly outperforms abstract modeling (Levels 2/4), with Level 4 remaining at 0% across all models. Among the additional models, DeepSeek-r1 achieves the highest average performance (31%) under few-shot prompting, primarily due to its relatively strong performance on Level 3 (16%) compared to other models. However, all models struggle with abstract modeling tasks, reinforcing the conclusion that semantic parameter inference and MATPOWER implementation remain fundamental challenges for current LLMs.

## H.3. Cross-Backbone Fine-tuning

To examine whether the benefits of ProOPF-D generalize beyond the Qwen model family, we fine-tune Llama-3.1-8B on ProOPF-D and evaluate it on ProOPF-B under the few-shot setting. As shown in Table 14, SFT improves performance from 0.00% to nontrivial accuracy across all four levels, with the largest gain on Level 2. This result indicates that ProOPF-D provides useful supervision for OPF modeling across different open-source backbones. The modest improvement on Level 4 (3.84%) suggests that mastering the most difficult setting, which combines semantic parameter inference with structural reformulation, still requires stronger base-model capability and further training data.

*Table 14.* Few-shot performance (%) of Llama-3.1-8B before and after SFT on ProOPF-D.

| Model | Level 1 | Level 2 | Level 3 | Level 4 |
|---|---|---|---|---|
| Llama-3.1-8B (w/o SFT) | 0.00 | 0.00 | 0.00 | 0.00 |
| Llama-3.1-8B (w/ SFT) | 22.20 | 36.70 | 10.30 | 3.84 |

*Table 13.* Model Performance on Different Levels

| Model | Concrete Modeling | | Abstract Modeling | | Average |
|---|---|---|---|---|---|
| | Level 1 | Level 3 | Level 2 | Level 4 | |
| *Few-shot Prompt* | | | | | |
| GPT-5.2 | 33.33% | 10.34% | 6.67% | 0.00% | 14.05% |
| GPT-5.1 | 61.11% | 6.90% | 10.00% | 0.00% | 22.31% |
| Claude 4.5 Sonnet | 77.78% | 37.93% | 6.67% | 0.00% | 33.88% |
| Claude 3.5 Sonnet | 58.33% | 0.00% | 10.00% | 0.00% | 19.83% |
| DeepSeek-V3.2 | 94.44% | 6.90% | 0.00% | 0.00% | 29.75% |
| DeepSeek-r1 | 94.44% | 17.24% | 16.67% | 0.00% | 36.36% |
| Gemini 3.0 Pro | 94.44% | 31.03% | 6.67% | 0.00% | 37.19% |
| Qwen3-Coder | 80.56% | 0.00% | 0.00% | 0.00% | 23.97% |
| Qwen3-30B-A3B | 50.00% | 0.00% | 0.00% | 0.00% | 14.88% |
| *Zero-shot Prompt* | | | | | |
| GPT-5.2 | 25.00% | 6.90% | 6.67% | 0.00% | 10.74% |
| GPT-5.1 | 22.22% | 0.00% | 6.67% | 0.00% | 8.26% |
| Claude 4.5 Sonnet | 66.67% | 6.90% | 6.67% | 0.00% | 23.14% |
| Claude 3.5 Sonnet | 16.67% | 6.90% | 0.00% | 0.00% | 6.61% |
| DeepSeek-v3.2 | 61.11% | 0.00% | 0.00% | 0.00% | 18.18% |
| DeepSeek-r1 | 5.56% | 0.00% | 0.00% | 0.00% | 1.65% |
| Gemini 3.0 Pro | 77.78% | 13.79% | 0.00% | 0.00% | 26.45% |
| Qwen3-Coder | 13.89% | 6.90% | 0.00% | 0.00% | 5.79% |
| Qwen3-30B-A3B | 0.00% | 0.00% | 0.00% | 0.00% | 0.00% |

