# OpenReview forum: "ProOPF: Benchmarking and Improving LLMs for Professional-Grade Power Systems Optimization Modeling"
_ICML.cc/2026/Conference — ICML 2026 regular_

### Official Review · Reviewer_iz8C · 2026-03-02

**Soundness:** 3
**Presentation:** 3
**Significance:** 3
**Originality:** 3
**Overall Recommendation:** 5
**Confidence:** 4

**Summary:**

This paper presents a dataset/benchmark for optimal power flow (OPF) related operations. The proposed approach splits the dataset four types of scenarios based on the OPF-related task (adding more components, changing values, etc.).

**Compliance With Llm Reviewing Policy:**

Affirmed.

**Final Justification:**

I have updated my score accordingly, as the authors answered my questions sufficiently.

**Key Questions For Authors:**

- You mention you have a dataset of 12K (4K samples per level), and also that you have a pool of 62 power systems + many different operational scenarios. Are 4K samples per case enough to create a good enough training set for such a wide number of cases? Would having more training samples during fine-tuning improve performance further?

- It would be interesting to evaluate the generalization capabilities to different power system architectures with this benchmark as well (given that every region has very different power grid archetypes). For example, what if in the training set you only had scenarios using grid X, and then the benchmark also had scenarios with grid Y? What do you think would happen then?

- What is the level of human validation of the 12K samples? Was human validation of the scenarios required, and what part was automated?

- Power systems are complex and have very specific mathematical formulations (compared to text datasets), hence it is not easy to evaluate whether an LLM answer is correct or not. How do you validate the LLM's answers in each case?

- You mention that a big part of the performance drop (on Level 2 and 4) is because of the lack of coding proficiency of the LLMs. Is it possible to get better performance if better tools are provided to the LLMs? Maybe a better prompt that helps in creating a valid model for the OPF solver?

**Limitations:**

The authors do not explicitly mention the weaknesses of this dataset (as they do for related work datasets), e.g., the number of samples, difficulty in validating outputs, etc. Maybe a short discussion/sentence could be added to emphasize this aspect as well.

**Strengths And Weaknesses:**

### Strengths
- LLM usage for power systems is very important for accelerating many tasks, from hosting capacity to future infrastructure planning. However, prior to this work, no datasets or benchmarks exist to validate the performance of existing LLM-based approaches for power systems.
- The dataset creation is well-documented, and useful insights are provided for the benchmark results.
- The four levels of "difficulty" make sense and show the progressively higher level of reasoning and expert knowledge required, making this dataset useful for evaluation of future, more powerful LLMs.

###  Weaknesses:
- Power systems are complex and have very specific mathematical formulations (compared to text datasets), hence it is not easy to evaluate whether an LLM answer is correct or not.

---

> ### Author Rebuttal · Authors · 2026-03-31
>
> We thank Reviewer iz8C for the constructive comments. Below we clarify the dataset design, validation process, and additional analyses, which will be reflected in the revised manuscript.
>
> > **Are 12K training samples, including 4K per level, sufficient to cover the diversity of power systems and operational scenarios considered, and would more fine-tuning data further improve performance?**
>
> We thank the reviewer for this important question. While ProOPF-D is constructed to cover diverse systems, operational scenarios, and parameter or structural modifications, it does not exhaust the full combinatorial space induced by these factors. **Instead, ProOPF-D prioritizes coverage across difficulty levels, since within-level samples often exhibit similar NL-to-code translation patterns, while cross-level capability gaps are more substantial.**
>
> To examine the effect of training data size, we add a scaling analysis with 4K, 8K, and 12K training samples:
> |Training data|L1 (%)|L2 (%)|L3 (%)|L4 (%)|
> |-|-|-|-|-|
> |4K|22.20|0.00|10.30|0.00|
> |8K|50.00|16.67|10.30|0.00|
> |12K|75.00|20.70|33.30|11.54|
>
> Due to time constraints, we did not further expand the dataset in the current revision. Nevertheless, the results show a clear positive scaling trend: gains on Levels 1-2 begin to taper, whereas Levels 3-4 improve markedly from 8K to 12K, suggesting that further expansion of Level-3/4 data is likely beneficial.
>
> > **How well does the model generalize to unseen power-system architectures?**
>
> We thank the reviewer for raising this important point. To evaluate cross-system generalization, we evaluate the ProOPF-D-trained model on 36 additional Level-1 samples from unseen power systems and compare ID and OOD performance. The new benchmark samples will be updated in the GitHub repository.
>
> |Setting|Zero-shot (%)|Few-shot (%)|
> |---|---:|---:|
> |ID systems|63.90|75.00|
> |OOD systems|66.67|72.22|
>
> **The results show good generalization to unseen systems**, which can be attributed to the largely consistent implementation workflow of OPF and its parameter modifications across different systems. ProOPF-D helps the LLM learn this workflow, leading to stable performance.
>
>
> > **What is the level of human validation of the 12K samples, and what parts are automated?**
>
> We thank the reviewer for this important question. **The 12K samples in ProOPF-D are constructed through expert-designed rules and an automated synthesis pipeline, whereas ProOPF-B is expert-annotated and manually verified.** For ProOPF-D, human effort is mainly concentrated on designing the synthesis pipeline, including the scenario trees, admissible parameter/structural modification space, prompt templates, and filtering rules for removing conflicting samples. Once these are specified, large-scale sample generation is carried out automatically.
>
> > **How do we validate the LLM's answers in each case?**
>
> During testing, we validate the generated executable MATPOWER code via execution-based evaluation (see Sec. 5 and Fig. 5 in the appendix).
> 1. For Levels 1/3, we execute the generated code directly and compare its objective value with that of the ground-truth implementation.
> 2. For Levels 2/4, we execute the generated parameterized MATPOWER function under predefined parameter instantiations and compare its objective value with that of the ground-truth implementation.
>
> > **Could performance on Levels 2 and 4 be improved by providing LLMs with better tools or prompts for generating valid OPF solver models?**
>
> We thank the reviewer for this valuable suggestion, and to assess whether stronger tool/prompt support can improve performance on Levels 2 and 4, we conduct additional tests on GPT and Claude with substantial additional reference information. Specifically, the prompt provides:
> 1. instructions for modifying objectives and constraints in MATPOWER code;
> 2. reference implementations of representative OPF variants;
> 3. instructions for translating scenario descriptions into parameter changes.
>
> |Model|Orig. L2(%)|Orig. L4(%)|Enh. L2(%)|Enh. L4(%)|
> |-|-|-|-|-|
> |GPT-5.2|6.67|0.00|26.67|26.92|
> |Claude-4.5|6.67|0.00|23.33|23.08|
>
> The results show that more solver-oriented support can substantially improve performance on Levels 2/4, suggesting a promising future work for tool-augmented OPF modeling.
>
> > **The paper should include a discussion of the dataset's limitations, particularly regarding sample size and the difficulty of output validation.**
>
> We thank the reviewer for this valuable suggestion. ProOPF currently focuses on transmission-level OPF modeling, and future work will expand Level-3/4 data to further improve performance on harder cases, while extending the ProOPF to broader power system optimization settings such as distribution network scheduling. Beyond execution-based evaluation, future work may also consider finer-grained validation of intermediate modeling steps. We will clarify these limitations in the revised manuscript.

---

> > ### Author Rebuttal · Reviewer_iz8C · 2026-04-02
> >
> > I thank the authors for addressing all my concerns.

---

> > > ### Author Response · Authors · 2026-04-03
> > >
> > > Thank you very much for your recognition and your positive feedback! We will further refine and polish the manuscript according to your suggestions to make it as strong as possible.

---

### Official Review · Reviewer_xXpe · 2026-03-04

**Soundness:** 2
**Presentation:** 2
**Significance:** 1
**Originality:** 2
**Overall Recommendation:** 2
**Confidence:** 4

**Summary:**

This paper introduces ProOPF-D (12K synthetic training instances) and ProOPF-B (121 expert-annotated test cases) to benchmark and improve LLMs in OPF modeling, where natural-language operational requests are translated into executable OPF code by treating each task as parameter patches and/or structural extensions to a canonical OPF; the benchmark evaluates end-to-end correctness via solver-executed objective values and finds that while SOTA models handle explicit parameter edits, they largely fail at semantic parameter inference and complex structural variants, with supervised fine-tuning on ProOPF-D yielding notable gains.

**Compliance With Llm Reviewing Policy:**

Affirmed.

**Final Justification:**

I appreciate the author's rebuttal, while I still have the following major concerns unresolved.

> Average solving time on IEEE300

I replicated this experiment using Pypower (a python port of the Matpower used in the paper) and found the reported results unreasonable:
|  IEEE300 | author reported  | reviewer reproduced (5 runs, Pypower) |
|---|---|---|
| ACOPF | 0.42s | 2.563568 s (std=0.037928)  |
| DCOPF  | 0.24s  |  0.092906 s (std=0.001593) |

To my knowledge, it is unreasonable for the speedup ratio of DCOPF be merely two times, especially given the transformation from nonlinear, nonconvex problem to linear programming problem and using MATLAB, which is much faster than python package in scientific computation.

> ProOPF is not a general framework for all power system tasks. & RL-based OPF methods are developed for a different setting.

Sorry, I cannot agree with this. Considering the application objective, such claims would restrain this paper in a problem-specific method and weaken its strength. RL-based methods can also deal with the varying operational conditions. To my knowledge, RL-based methods can be orders of magnitude faster than ProOPF since ProOPF still requires optimizer in the loop; unless ProOPF can significantly outperform RL methods in terms of accuracy—otherswise, with all due respect, I struggle to see the significance of this research. The authors have failed to provide this crucial comparison.

In addition, there are already agentic research for power flow analysis [1][2], enabling cross-task and cross-tool capabilities, rather than restraining the capability within a single task (OPF) and a single tool (MATPOWER, more for research rather than production tool) in this paper, which greatly diminishes its potential for application in real-world scenarios.

> Checking for catastrophic forgetting and Assessing broader optimization modeling ability.

I don't think these should be major concerns of this paper, as its application nature and targeted for power system operation.

[1] Chen X. X-GridAgent: An LLM-Powered Agentic AI System for Assisting Power Grid Analysis[J]. arXiv preprint arXiv:2512.20789, 2025.

[2] Zhang Q, Xie L. PowerAgent: A road map toward agentic intelligence in power systems: Foundation model, model context protocol, and workflow[J]. IEEE Power and Energy Magazine, 2025, 23(5): 93-101.

**Key Questions For Authors:**

See above weaknesses.

**Limitations:**

See above.

**Strengths And Weaknesses:**

### Strengths:
**Well-motivated problem setting**: The paper aims to automate professional power system optimization using LLMs, with clear practical importance in real-world operations since OPF models are required to adapt to changing operation conditions.

**Domain-specific benchmark**: The paper introduces ProOPF-D (dataset) and ProOPF-B (benchmark), shifting evaluation from coarse cross-domain benchmarks to OPF-centric modeling.


### Weaknesses:
**Misalignment with practically critical problem**: Generating codes for conducting optimization didn't resolve the bottleneck of optimization itself. Power system researchers are typically struggling with how to adjust parameters or relax constraints to get an approximately optimal solution when a nonlinear AC-OPF does not converge in scaling systems with thousands of buses, which in my perspective, a Reinforcement Learning formulation that teaches the coding agent to reason given the optimizer returns could be much more interesting. Or, how to accelerate the solving process when it requires tens of minutes to solve. Unfortunately, this paper included scaled systems, however, it didn't consider these critical challenges at all.

**Limited scope and general impact**: Restricting considered tasks only within OPF and limiting the dataset scale to 12K diminishes the contribution, making it difficult to produce broader impacts on tasks other than OPF and out of power system domain, which to my knowledge, more attractive in a power system focused conference/journal.

**Trivial conclusion & incomplete ablations**: The conclusion of that SFT on domain-specific data improves domain-specific performance is trivial. The authors did not provide results from open-source models other than Qwen for ablation study, nor did they conduct data scaling experiments.

---

> ### Author Rebuttal · Authors · 2026-03-31
>
> We thank Reviewer xXpe for the comments on the practical relevance, scope, and experimental design of our work. Below, we clarify the contribution of ProOPF and summarize the additional analyses included in the revision.
>
> > **The paper's focus on optimization code generation is not sufficiently aligned with practical challenges in large-scale power system optimization, such as non-convergence handling and solver efficiency.**
>
> We agree that non-convergence handling and solver efficiency are important challenges in large-scale power system optimization. However, under growing renewable and inverter-based resource penetration, real-time dispatch increasingly requires finer-grained and more adaptive optimization models that reflect changing operating conditions and accommodate structural modifications, including stability-related constraints and probabilistic formulations [B2]. While experts can revise such OPF models manually, doing so remains labor-intensive and difficult to scale in near-real-time settings [B1]. This motivates LLM-assisted optimization modeling. **In this context, we introduce ProOPF-D/B, the first dataset and benchmark for evaluating LLM competence in power system optimization modeling.** We will clarify this motivation more explicitly in the revised manuscript and discuss as future work modeling methods that better balance formulation fidelity and solution efficiency.
>
> > **The paper's scope and broader impact appear limited because it focuses on OPF-specific tasks and a 12K-instance dataset, raising concerns about its generalizability beyond OPF and the power systems domain.**
>
> We thank the reviewer for this comment and clarify ProOPF's broader impact from three perspectives.
>
> 1) **ProOPF covers a family of OPF-related formulations rather than a single task.** Beyond a canonical OPF backbone, it includes structurally related variants such as OTS and SCUC, as detailed in Appendix B.
> 2) **ProOPF offers a novel benchmark perspective by prioritizing depth over breadth, while remaining readily scalable.** Rather than spanning many heterogeneous tasks, ProOPF enables fine-grained evaluation within a professional domain, and the largely automated synthesis pipeline allows the dataset to be expanded when needed.
> 3) **Cross-benchmark transfer remains broadly stable after ProOPF fine-tuning.** We further evaluate the ProOPF-D-trained model on non-OPF benchmarks as shown below. The results indicate that ProOPF-D improves professional power-system optimization modeling while largely preserving general optimization-modeling capability.
>
> |Benchmark|Before SFT(%)|After SFT(%)|
> |-|-|-|
> |NL4Opt|71.74|73.91|
> |MAMO-EasyLP|90.03|90.34|
> |MAMO-ComplexLP|24.17|21.80|
> |OptiBench|55.37|47.93|
>
> > **The conclusion of that SFT on domain-specific data improves domain-specific performance is trivial. The authors did not provide results from open-source models other than Qwen for ablation study, nor did they conduct data scaling experiments.**
>
> We thank the reviewer for this important comment. To test whether ProOPF-D yields only in-domain gains, we also evaluate the fine-tuned model on non-OPF benchmarks. As shown above, performance remains broadly stable after SFT, suggesting improved power-system modeling with little loss of general optimization-modeling capability.
>
> To strengthen the empirical analysis, we add two more experiments. First, we evaluate Llama 3.1 8B before and after SFT on ProOPF-D under the few-shot setting. As shown below, SFT improves performance on all four levels, with the largest gain on Level 2, showing that ProOPF-D is effective beyond the Qwen family. However, likely due to its smaller scale, Llama 3.1 8B still trails Qwen3-30B in overall accuracy.
>
> |Model|L1(%)|L2(%)|L3(%)|L4(%)|
> |-|-|-|-|-|
> |Llama-3.1-8B (w/o SFT)|0.00|0.00|0.00|0.00|
> |Llama-3.1-8B (w/ SFT)|22.20|36.70|10.30|3.84|
>
> Second, we conduct a data-scaling study by SFT Qwen3-30B on 4K, 8K, and 12K subsets of ProOPF-D and evaluating the resulting models on ProOPF-B under the few-shot setting. Overall performance improves with more training data, indicating a clear positive scaling trend. However, gains on Levels 1-2 begin to taper, whereas Levels 3-4 improve markedly from 8K to 12K, suggesting that structurally variant cases require more samples for the model to learn typical code patterns. Future work will further expand Level-3/4 data to test whether larger gains can be achieved on these harder settings.
>
> |Data|L1(%)|L2(%)|L3(%)|L4(%)|
> |-|-|-|-|-|
> |4K|22.20|0.00|10.30|0.00|
> |8K|50.00|16.67|10.30|0.00|
> |12K|75.00|20.70|33.30|11.54|
>
> [B1] Jia et al. Enhancing LLMs for Power System Simulations: A Feedback-Driven Multi-Agent Framework. IEEE TSG.
>
> [B2] Lorca et al. Adaptive Robust Optimization With Dynamic Uncertainty Sets for Multi-Period Economic Dispatch Under Significant Wind. IEEE TPS.

---

> > ### Author Rebuttal · Reviewer_xXpe · 2026-04-03
> >
> > > The paper's focus on optimization code generation is not sufficiently aligned with practical challenges in large-scale power system optimization, such as non-convergence handling and solver efficiency.
> >
> > I think the rebuttal didn't directly answer my concern, e.g., the non-convergence issue. To my knowledge, the efficiency issue cannot be resolved in the proposed framework. The core idea of ProOPF is to automatically adjust OPF model, still requesting optimizer in the loop, where the main overhead comes from.
> >
> > > ProOPF covers a family of OPF-related formulations rather than a single task.
> >
> > My concern is whether this framework could be extended to tasks out of OPF-related formulations, e.g. fault detection. There are many task-specific AI methods that solve OPF well using reinforcement learning with better efficiency (since no optimizer in the loop).
> >
> > > Cross-benchmark transfer remains broadly stable after ProOPF fine-tuning.
> >
> > Power system model should focus on power system tasks. It is not reasonable to evaluate power system model on general benchmarks after finetuning on power system corpus.

---

> > > ### Author Response · Authors · 2026-04-06
> > >
> > > We thank the reviewer xXpe for the follow-up comments. Below, we clarify the scope, applicability, and evaluation rationale of ProOPF.
> > >
> > > > The rebuttal does not directly address non-convergence and solver efficiency.
> > >
> > > We thank the reviewer for the feedback. Non-convergence handling and solver efficiency are important challenges in large-scale OPF. However, these issues are **largely orthogonal** to the focus of ProOPF, which concerns the automation of optimization model formulation/code generation. This focus is motivated by:
> > >
> > > 1. **Practical need in power systems**: In dynamic industrial contexts, optimization models require frequent revision, yet such updates remain costly because high-quality modeling demands substantial expertise and labor [4]. This challenge is particularly acute in power systems, where increasing renewable penetration makes near-real-time dispatch more dependent on frequent OPF revisions to reflect changing operating requirements [1,2].
> > > 2. **Benchmark gap in optimization LLMs**: Although LLM-based optimization modeling has emerged as a recognized research problem [4,6], existing LLM optimization benchmarks (like NL4OPT and Optibench) focus on cross-task generalization and fail to evaluate fine-grained expert modeling for power systems.
> > >
> > > While task-specific AI methods without optimizers in loop may offer faster inference, reliability and interpretability requirements still necessitate solving explicitly formulated optimization models with numerical optimizers in practical power system settings. Within this paradigm, **solver efficiency is often improved through adjustments such as model reformulation, better initial points, or solver parameter tuning** [2]. ProOPF **can support such adjustments** by translating these instructions into executable solver code. Specifically, we add an IEEE 300-bus ACOPF test with poor initialization to increase solve time. For each instruction type, we sample 20 outputs from the ProOPF-D-trained model and report code pass rate and average solve time. As shown below, the model reliably converts such instructions into executable code and achieves clear runtime reduction.
> > >
> > > |Speedup instr.|Code pass (/20)|Avg. time (s)|
> > > |-|-|-|
> > > |Original ACOPF|--|0.42|
> > > |+ AC to DC reformulation|20|0.24|
> > > |+ Improved initialization|19|0.21|
> > > |+ Solver parameter tuning|20|0.35|
> > >
> > > Based on these results, we will further explore solver-efficiency-oriented extensions of ProOPF in future work.
> > >
> > > > The extensibility of ProOPF beyond OPF-related optimization tasks remains unclear.
> > >
> > > Thank you for the comment. We clarify this as follows:
> > >
> > > 1. **ProOPF is not a general framework for all power system tasks**. it is designed for tasks whose core difficulty lies in **optimization model construction and code generation**. Accordingly, it may extend to other optimization structured problems, such as active distribution network dispatch, but not to fault detection, which is a diagnosis task rather than an optimization-modeling task.
> > > 2. **RL-based OPF methods are developed for a different setting**. They aim to solve a fixed OPF formulation efficiently [6], whereas ProOPF addresses formulation revision under changing operational requirements. As a result, the two are not directly comparable by solver or inference efficiency alone.
> > >
> > > > General-benchmark evaluation after power-system fine-tuning seems insufficiently motivated.
> > >
> > > We agree that evaluation should focus primarily on power-system tasks. Accordingly, ProOPF-B is the principal benchmark in this work. The cross-domain benchmarks are included only as supplementary evidence, for two reasons:
> > >
> > > 1. **Checking for catastrophic forgetting.** It is standard practice to verify that a model, after domain-specific SFT, retains its foundational abilities rather than overfitting to the target corpus [3, 4]—similar to validating math-trained models on coding benchmarks [5].
> > >
> > > 2. **Assessing broader optimization modeling ability.** Since these benchmarks evaluate general NL-to-optimization translation, they help verify that power system SFT does not come at the expense of general optimization modeling capability.
> > >
> > > We will explicitly clarify this motivation in the revision.
> > >
> > >
> > > [1] Jia et al. Enhancing LLMs for Power System Simulations: A Feedback-Driven Multi-Agent Framework. IEEE TSG.
> > >
> > > [2] Park et al. Compact optimization learning for AC optimal power flow. IEEE TPS.
> > >
> > > [3] Li et al. Revisiting Catastrophic Forgetting in Large Language Model Tuning. Findings of EMNLP 2024.
> > >
> > > [4] Huang et al. Orlm: A customizable framework in training large models for automated optimization modeling. Operations Research.
> > >
> > > [5] Shao et al. DeepSeekMath: Pushing the Limits of Mathematical Reasoning in Open Language Models. arXiv.
> > >
> > > [6] Jiang et al. LLMOPT: Learning to Define and Solve General Optimization Problems from Scratch. ICLR.
> > >
> > > [7] Yan et al. Real-time optimal power flow: A lagrangian based deep reinforcement learning approach. IEEE TPS.

---

### Official Review · Reviewer_ZXyg · 2026-03-12

**Soundness:** 3
**Presentation:** 3
**Significance:** 3
**Originality:** 3
**Overall Recommendation:** 5
**Confidence:** 4

**Summary:**

This is a dataset & benchmarks paper, introducing ProOPF, consisting of a dataset and a benchmark designed to eval/improve LLMs in generating optimization models, specifically in the specialized field of optimal power flow modeling.

**Compliance With Llm Reviewing Policy:**

Affirmed.

**Key Questions For Authors:**

See above.

**Limitations:**

Yes

**Strengths And Weaknesses:**

## 1. Novelty and utility:

Unlike NLP4OPT, MAMO, and existing optimization modeling benchmarks; which are general and canonical problems, the authors focus on OPF problem specifically. On one hand, this limits its generality, but on the other hand; OPF problems seem to be a good problem to benchmark on due to the high number of variables and coupled physical constraints. The authors also introduce a fairly reproducible recipe of synthesizing new instances following a structured progression.

## 2. Method:

The main 'method' the authors propose is a way to synthesize progressively more challenging problems by starting from a given problem then introducing parametric or structural modifications. My main question here is that this approach seems generalizable to a lot of different problem classes e.g., vehicle routing, scheduling-type problems where there are also clear parameters that can be explicit or inferred and there are fairly clear structure modifications to progressively increase complexity (e.g., new constraints). Have the authors tried this, or do they think this is feasible?

## 3. ProOPF-B vs ProOPF-D:

I like that the lower-fidelity/quality ProOPF-D is mainly targeted at model training/finetuning. I have a concern that the -B and -D subsets are in-distribution i.e., the -D is generated specifically to appear similar in problem mathematical structure and description. Have the authors considered this, is it by-design or unintentional?

## 4. Semantic 'diversity' in NL descriptions:

The 'scenario fragments' used to generate more diverse and natural sounding problem descriptions are fairly rigid. I mean this in the sense that it all has to come from a predefined tree of possible events. This does not appear scalable with respect to diversity and 'coverage'.


Overall, I think this is a useful contribution; its narrow focus is also its strength; allowing the benchmark to be designed thoughtfully while also enabling scaling. One final suggestion I have for the authors is to show some correlation between ProOPF and other optimization modeling benchmarks; it would be very useful if e.g., a moel finetuned on ProOPF or simply evaled on the benchmark is strongly likely to perform well on non-OPF benchmarks too.

---

> ### Author Rebuttal · Authors · 2026-03-31
>
> We thank the Reviewer ZXyg for the constructive comments. In response, we add evaluations on broader optimization benchmarks and out-of-distribution generalization, and will reflect these changes in the revised manuscript.
>
> > **Could the progressive synthesis framework underlying ProOPF be extended to other problem classes beyond OPF?**
>
> We agree that the proposed synthesis framework is not limited to OPF. More broadly, it provides a general recipe for constructing multi-level datasets from a canonical optimization model via parameter and structural modifications. As future work, we plan to extend this synthesis paradigm to additional domains and build similarly structured datasets with multiple difficulty levels.
>
> > **Does the current design of ProOPF-D/B primarily support in-distribution (ID) evaluation?**
>
> We thank the reviewer for raising this important concern. The ProOPF-D/B mainly supports ID evaluation, as is common in optimization-modeling benchmarks such as IndustryOR [A1], LLMOPT [A2], and ReSocratic [A3]. Following the reviewer's suggestion, we add three OOD evaluations: unseen systems on Level 1, unseen scenarios on Levels 2 and 4, and transfer from Levels 1--3 to Level 4. The new benchmark samples will also be released on GitHub.
>
> **1. Level-1 cross-system generalization.** We evaluate the ProOPF-D-trained model on 36 additional Level-1 samples from unseen power systems and compare its ID and OOD performance.
>
> |Setting|Zero-shot (%)|Few-shot (%)|
> |-|-|-|
> |ID systems|63.90|75.00|
> |OOD systems|66.67|72.22|
>
> **The results show good generalization to unseen systems**, which can be attributed to the largely consistent implementation workflow of OPF and its parameter modifications across different systems. ProOPF-D helps the LLM learn this workflow, leading to stable performance.
>
> **2. Scenario generalization on Levels 2 and 4.** We add 30 Level-2 and 26 Level-4 samples from extended scenario trees to test unseen scenario compositions.
>
> |Setting|Zero-shot (%)|Few-shot (%)|
> |-|-|-|
> |ID Level-2|23.30|33.30|
> |OOD Level-2|16.67|30.00|
> |ID Level-4|7.69|11.54|
> |OOD Level-4|7.69|7.69|
>
> **The results show limited degradation under OOD scenarios**, while maintaining nontrivial performance on unseen scenario branches. This suggests that ProOPF-D helps the model acquire transferable scenario-conditioned parameter inference capabilities beyond the original scenario-tree structure.
>
> **3. Generalization from Levels 1--3 to Level 4.** We further evaluate whether the capabilities learned from the first three levels transfer to Level 4. Specifically, we compare Level-4 performance under full training and under training restricted to Levels 1--3.
>
> |Training split|Zero-shot (%)|Few-shot (%)|
> |-|-|-|
> |Levels 1--4|7.69|11.54|
> |Levels 1--3 only|3.84|3.84|
>
> **The results show nonzero Level-4 performance even when training excludes Level-4 data**, indicating partial transfer to this held-out setting. At the same time, the gap relative to full training on Levels 1--4 highlights the additional difficulty of Level 4 and supports the need for its explicit inclusion in ProOPF-D.
>
>
>
> > **Can the predefined scenario tree in ProOPF be extended to provide sufficient semantic diversity and coverage in NL descriptions?**
>
> We thank the reviewer for this important question. **The scenario tree is not a fixed template set, but an extensible semantic framework that maps operational narratives to valid OPF modifications**. New event types and scenario-to-parameter mappings can be added incrementally, enabling broader diversity and coverage. This extensibility is further supported by the OOD results on Levels 2 and 4, where the model is evaluated on newly introduced scenario branches beyond the original tree.
>
> > **How well does performance on ProOPF transfer to other non-OPF optimization benchmarks?**
>
> We thank the reviewer for this valuable suggestion. To assess transfer beyond ProOPF-B, we evaluate the model on NL4Opt, MAMO-Easy/ComplexLP, and OptiBench before/after SFT on ProOPF-D.
>
> |Benchmark|Before SFT (%)|After SFT (%)|
> |-|-|-|
> |NL4Opt|71.74|73.91|
> |MAMO-EasyLP|90.03|90.34|
> |MAMO-ComplexLP|24.17|21.80|
> |OptiBench|55.37|47.93|
>
> Overall, performance on non-OPF benchmarks remains broadly stable after SFT: NL4Opt and MAMO-EasyLP improve, while the other two decline only modestly. These results suggest that ProOPF-D strengthens professional power-system modeling while largely preserving general optimization-modeling capability. As future work, we will explore combining ProOPF-D with general optimization datasets to further improve both specialization and generality.
>
> [A1] Huang et al. Orlm: A customizable framework in training large models for automated optimization modeling. Operations Research.
>
> [A2] Jiang et al. LLMOPT: Learning to Define and Solve General Optimization Problems from Scratch. ICLR.
>
> [A3] Yang et al. OptiBench Meets ReSocratic: Measure and Improve LLMs for Optimization Modeling. ICLR

---

> > ### Author Rebuttal · Reviewer_ZXyg · 2026-04-04
> >
> > The author's rebuttal has resolved my main concerns; and I maintain my positive view on the paper.

---

> > > ### Author Response · Authors · 2026-04-06
> > >
> > > Thank you very much for your recognition and your positive feedback! We will further refine and polish the manuscript according to your suggestions to make it as strong as possible.

---

### Decision · Program_Chairs · 2026-04-30

**Decision:**

Accept (regular)

**Comment:**

Reviewers praised the strength and thoughtful design of this paper's dataset/benchmark, which enables in-depth assessment of methods to translate natural language prompts into executable optimization models for optimal power flow (OPF)-style problems in power grids. Reviewers appreciated the focus on OPF, enabling the evaluation of methods in settings with a large number of variables and coupled physical constraints, as well as careful construction of multiple levels of difficulty. Comments regarding out-of-distribution evaluation and extensibility of insights to non-OPF problems were satisfactorily addressed during the rebuttal. I therefore recommend acceptance.

One major point for improvement, however, is that the motivation of the work could be better framed. Other work on LLMs for optimization modeling has often been developed to aid cases where non-experts need to form optimization problems, but this motivation does not necessarily hold for power systems, where modeling is often being done by power systems experts who understand the underlying optimization formulations. While the authors, in the rebuttal, mention labor savings and faster formulation in real-time settings, the justifying citation is to another paper developing LLM-based methods, rather than a primary power systems domain source identifying the core challenge. To avoid the perception of a "hammer finding a nail," the authors should revise the paper ahead of the camera ready to strengthen the discussion of the domain-related gap this work addresses, in addition to addressing other reviewer feedback.